

# Assessing the impact of shipping emissions on air pollution in the Canadian Arctic and northern regions: current and future modelled scenarios

Wanmin Gong[1], Stephen R. Beagley[1], Sophie Cousineau[2], Mourad Sassi[2], Rodrigo Munoz-Alpizar[2], Sylvain Ménard[2], Jacinthe Racine[2], Junhua Zhang[1], Jack Chen[3], Heather Morrison[1], Sangeeta Sharma[1], Lin Huang[1], Pascal Bellavance[4], Jim Ly[4], Paul Izdebski[5], Lynn Lyons[4], Richard Holt[6]

[1]Science and Technology Branch, Environment and Climate Change Canada, Toronto, Ontario, M3H 5T4, Canada
[2]Meteorological Service of Canada, Environment and Climate Change Canada, Montreal, Quebec, H9P 1J3, Canada
[3]Science and Technology Branch, Environment and Climate Change Canada, Ottawa, Ontario, K1V 1C7, Canada
[4]Environmental Protection Branch, Environment and Climate Change Canada, Gatineau, Quebec, K1A 0H3, Canada
[5]Environmental Protection Branch, Environment and Climate Change Canada, Toronto, Ontario, M3H 5T4, Canada
[6]Environmental Protection Branch, Environment and Climate Change Canada, Vancouver, British Columbia, V6C 3S5, Canada

*Correspondence to*: Wanmin Gong (Wanmin.gong@canada.ca)

**Abstract.** A first regional assessment of the impact of shipping emissions on air pollution in the Canadian Arctic and northern regions was conducted in this study. Model simulations were carried out on a limited-area domain (at 15-km horizontal resolution) centred over the Canadian Arctic, using the Environment and Climate Change Canada's on-line air quality forecast model (GEM-MACH), to investigate the contribution from the marine shipping emissions over the Canadian Arctic waters (at both present and projected future levels) to ambient concentrations of criteria pollutants ($O_3$, $PM_{2.5}$, $NO_2$, and $SO_2$), atmospheric deposition of sulphur and nitrogen, atmospheric loading and deposition of black carbon in the Arctic. Several model upgrades were introduced for this study, including the treatment of sea-ice in the dry deposition parameterization, chemical lateral boundary conditions, and the inclusion of North American wildfire emissions. The model is shown to have similar skills in predicting ambient $O_3$ and $PM_{2.5}$ concentrations in the Canadian Arctic and northern regions as the current operational air quality forecast models in North America and Europe. In particular, the model is able to simulate well the observed $O_3$ and PM components at the Canadian high Arctic site, Alert. The model assessment shows that, at the current (2010) level, Arctic shipping emissions contribute to less than 1% of ambient $O_3$ concentration over the eastern Canadian Arctic and between 1 and 5% of ambient $PM_{2.5}$ concentration over the shipping channels. Arctic shipping emissions make a much greater contributions to the ambient $NO_2$ and $SO_2$ concentrations, at 10 – 50% and 20 – 100 %, respectively. At the projected 2030 business-as-usual (BAU) level, the impact of Arctic shipping emissions is predicted to increase to up to 5% in ambient $O_3$ concentration over a broad region of the Canadian Arctic and to 5 – 20% in ambient $PM_{2.5}$ concentration over the shipping channels. In contrast, if emission controls such as the ones implemented in the current North American Emission Control Area (NA ECA) are to be put in place over the Canadian Arctic waters, the impact of shipping to ambient criteria pollutants would be significantly reduced. For example, with NA-ECA-like controls, the



shipping contributions to population-weighted concentration of $SO_2$ and $PM_{2.5}$ would be brought down to below the current level. The contribution of Canadian Arctic shipping to the atmospheric deposition of sulphur and nitrogen is small at the current level, $< 5\%$, but is expected to increase to up to 20% for sulphur and 50% for nitrogen under the 2030 BAU scenario. At the current level, Canadian Arctic shipping also makes only small contributions to BC column loading and BC deposition, $< 0.1\%$ on average and up to 2% locally over eastern Canadian Arctic for the former, and between 0.1 and 0.5% over the shipping channels for the latter. The impacts are again predicted to increase at the projected 2030 BAU level particularly over the Baffin Island and Baffin Bay area in response to the projected increase in ship traffic there, e.g., up to 15% on BC column loading and locally exceeding 30% on BC deposition. Overall, the study indicates that shipping induced changes in atmospheric composition and deposition are at regional to local scales (particularly in the Arctic). Climate feedbacks are thus likely to act at these scales so climate impact assessments will require modelling undertaken at much finer resolutions than those used in the existing radiative forcing and climate impact assessments.

# 1 Introduction

Unprecedented rates of warming are increasing the navigability of the Arctic Ocean and, subsequently, rendering this region accessible to increasing resource exploitation and the development that goes along with this. Over the past several decades, the extent of Arctic sea ice has declined. The rate of decline of late summer sea-ice cover has been particularly rapid since the beginning of this century (e.g., Serreze et al., 2007). The latest climate model simulations predict that the retreat of Arctic sea ice will continue throughout the $21^{st}$ century, and that an ice-free Arctic ocean in late summertime may be realised by mid-to-late this century (Wang and Overland, 2009; Boé et al., 2009). The decline in Arctic sea ice has raised the prospect of increased Arctic shipping activities and the potential use of new transit routes, such as the Northern Sea Route, the Northwest Passage, and the Trans-Polar Route (e.g., Stephenson et al., 2013; Melia et al., 2016). Pizzolato et al. (2016) conducted a coupled spatial analysis between shipping activity and sea ice using observations in the Canadian Arctic over the 1990-2015 period, and found that there has been an increase in shipping activities in Hudson Strait, Beaufort Sea, Baffin Bay, and regions in the southern route of the Northwest Passage, and that the increases in shipping activity are significantly correlated with the reductions in sea ice concentration in these regions.

Shipping is an important source of air pollutants. Emissions of exhaust gases and particles from ocean-going ships contains carbon dioxide ($CO_2$), nitrogen oxides ($NO_x$), carbon monoxide (CO), volatile organic compounds (VOC), sulfur dioxide ($SO_2$), black carbon (BC), and particulate organic matter (OM). These pollutants lead to the production of ozone ($O_3$) and fine particulate matter (e.g., $PM_{2.5}$), the latter primarily through oxidation of $SO_2$ and formation/production of sulphate ($SO_4$) particles, which degrade air quality. At the same time, $O_3$ and $SO_4$ resulting from ship emissions, along with $CO_2$ and BC directly emitted from shipping, are also climate forcing agents which can impact the radiative balance through either direct or indirect effect. Shipping emissions also contribute to the deposition of nitrogen (N) and sulphur (S) which can impact ecosystems through acidification and eutrophication. Recent studies have suggested that around 15% and 4-9% of all global





anthropogenic emissions of $NO_x$ and $SO_2$, respectively, are from ocean-going ships (e.g., Corbett and Köhler, 2003; Eyring et al., 2005a). As most of the ship emissions occur within 400 km of coastlines, this primarily contributes to air pollution in coastal areas (e.g., Eyring et al., 2010; Viana et al., 2014; Aulinger et al., 2016; Aksoyoglu et al., 2016). However, theses emissions can be transported hundreds of kilometers downwind and impact a much broader region (e.g., Eyring et al., 2010;

Aulinger et al., 2016). Although Arctic marine shipping currently accounts for a small percentage of global shipping emissions, it makes a proportionally bigger impact on the environment than does shipping at lower latitudes due to the generally pristine Arctic background. Furthermore, the lower troposphere in the Arctic is more isolated during summer, which is also the Arctic shipping season, due to the retreating Arctic dome, giving rise to much slower transport of pollutants from lower latitudes, and more efficient removal processes (Law and Stohl, 2007; Stohl, 2006). Local sources of air

pollution, such as shipping, play a more important role in determining air quality in this region during this time.

A number of studies assessing the impact of Arctic shipping emissions have been conducted in recent years. Based on the high-growth scenario projection of Eyring et al. (2005b) on future international shipping emissions (to year 2050), assuming a fraction of the increase would occur in the Arctic, Granier et al. (2006) predicted an increase in Arctic surface $O_3$ concentration, by a factor of 2 to 3, due to the increase in ship $NO_x$ emission. Ødemark et al. (2012) looked into short-lived

climate forcers from current shipping and petroleum activities in the Arctic based on inventories developed by Peters et al. (2011), and found that radiative forcing from shipping emissions is dominated by the direct and indirect effects of sulphate from $SO_2$ emissions during shipping season. The overall effect from shipping on radiative forcing is negative. Dalsøren et al. (2013) assessed the changes in surface concentrations of $NO_2$, $O_3$, $SO_4$, BC, and organic carbon (OC) between year 2004 and 2030, based on the Arctic shipping inventories developed by Corbett et al. (2010), which take into account Arctic

shipping growth, possible emission control measures, and the opening of diversion routes for shipping in the Arctic due to the expected melting of sea ice. Based on the same inventories of Corbett et al (2010), Browse et al. (2013) investigated the impact of Arctic shipping on BC deposition at high latitudes, and found that the overall impact from Arctic shipping to total BC deposition remains low. Their results show that Arctic shipping contributes a maximum of 1.9% to the total annual BC deposition north of 60ºN at present levels and a maximum of 5% at 2050 levels under a high-growth scenario. Most of these

assessments were conducted using global models at coarse resolutions (e.g., 2.8º x 2.8º). In a recent study on cross-polar transport and scavenging of Siberian aerosols, Raut et al. (2017) found that the model simulation at a coarser horizontal resolution (i.e., 100 km instead of 40 km) was unable to resolve plume structures transported across the polar region in summer. The model performed much better at simulating the cross-polar transport and processing using a finer horizontal resolution (40 km). At a regional scale, Marelle et al. (2016) used model simulations at 15-km resolution to estimate the

regional impacts of shipping pollution in northern Norway during a 15-day period in July 2012 when an aircraft measurement campaign was conducted to characterize pollution originating from shipping and other local sources. Their estimate of the impact of shipping emissions on $O_3$ production over the Norwegian coast was considerably lower than the estimate of Ødemark et al. (2012) which was based on model simulation at a much coarser resolution (2.8º x 2.8º). The authors attributed the difference in estimated impact, at least in part, to the non-linear effects associated with the unrealistic



instant dilution of ship $NO_x$ emissions in global models run at coarse resolutions, particularly under pristine background conditions, as found in Vinken et al. (2011).

In this study, we assess the impact of emissions from marine shipping on the Canadian Arctic using an on-line comprehensive air quality forecast model (GEM-MACH) configured for the Arctic at 15-km resolution. A detailed baseline

emission inventory for ships sailing in Canadian waters was developed utilizing vessel movement data for 2010 supplied by the Canadian Coast Guard (CCG) and activity-based emissions factors. Projections of Canadian Arctic marine shipping emissions to a future year (2030) were made based on two scenarios, business-as-usual and emission controls (a.k.a., controlled). Model simulations for the Arctic shipping season were carried out, with and without the marine shipping emissions over the Canadian Arctic waters, at both the current (2010 baseline) and future (projected) levels. The

contributions from Canadian Arctic shipping emissions to ambient concentrations of criteria pollutants ($O_3$, $PM_{2.5}$, $NO_2$, and $SO_2$), total S and N deposition, and BC loading and deposition were assessed in the context of their relevance to air quality, local ecosystems, and climate. In the following sections, we will describe the Canadian shipping emission inventories (Section 2) and the modelling system and simulation setup (Section 3). An evaluation of the 2010 baseline simulation against available observations is presented in Section 4, and the assessment of the impact of the Arctic shipping emissions in Section

5. We will end with conclusions in Section 6.

## 2 The 2010 Canadian national marine shipping inventory, Arctic shipping activities (current and projections)

The 2010 Canadian national marine shipping emission inventory used for this study was generated by using a Marine Emission Inventory Tool (MEIT) developed for Environment and Climate Change Canada (ECCC) (SNC-Lavalin Environment, 2012). The inventory includes all commercial marine vessel classes tracked by the Canadian Coast Guard

(CCG) within Canadian waters, as well as small commercial craft such as ferries, tugboats and fishing vessels. All coastal area as well as inland rivers and lakes are included in the inventory. The basis for the inventory is movement data as logged in the Information System on Marine Navigation (INNAV) for eastern Canada and the Arctic and the Vessel Traffic Operator Support System (VTOSS) through CG Vessel Traffic Services (VTS) for the west coast. INNAV data for 2010 is representative of all oceangoing vessel (OGV) movements, whereas data gaps exist in the 2010 VTOSS dataset. In addition,

Pacific Pilotage Authority movement data and port-level data are also used to supplement VTOSS data as needed (SLE, 2012). The activity-based emission factors used in MEIT for processing the 2010 national inventory were specific factors appropriate for engine size (based on US EPA engine classification), speed, and fuel type (Weir Marine Engineering, 2008; SLE, 2012). Emissions were calculated on a voyage-by-voyage basis, and vessel speed and implied load on the main and auxiliary engines were evaluated by each segment of a voyage. Temporal resolution of the 2010 national marine inventory

includes emissions by hour, day, and month of the year, and spatial resolution includes emissions allocated to regions of Canada (by province and many sub-regions defined in previous marine emission inventory analysis work). The Arctic portion of the 2010 national marine emission inventory was further updated including revised main engine load factors



(Innovation Maritime and SNC-Lavalin Environment, 2013). The emission inventory covers criteria air contaminants (CACs), such as nitrogen oxides ($NO_x$), sulphur oxides ($SO_x$), carbon monoxide (CO), volatile organic compounds (VOCs) (including VOCs from combustion and fugitive VOCs from crude oil tankers but not fugitive VOC emissions from oil barges and other petroleum tankers), particulate matter (PM, as total PM, $PM_{10}$ and $PM_{2.5}$, as well as elemental, organic, and sulphate fractions), and ammonia ($NH_3$), greenhouse gases (GHGs), and air toxics.

The Canadian Arctic waters defined in this study are the portion of Canadian waters excluded from the North American Emission Control Area (ECA), which include both coastal and inland waters north of 60ºN, Hudson Bay, and James Bay (see Figure 1). Canada's Arctic waters (particularly in the high Arctic) are characterized by variable ice conditions and extreme weather. The vastness and remoteness of the region further contribute to the challenges that shippers are faced with when sailing through these waters. Even during the summer months when ice levels are at their lowest, ships must ensure that they have ice-strengthened hulls or be escorted by a CCG icebreaker to ensure a safe and manageable transit. Current marine traffic in Canada's Arctic is primarily comprised of vessels heading to specific Northern destinations. These vessels function as a vital link between remote Northern communities and the essential supplies they need, typically from Southern Canada. In addition to these vital community resupply sea-lifts, ships transiting Canada's Arctic are also engaged in hydrocarbon and mineral exploration (i.e., seismic exploration) and extraction, eco-tourism and activities of the CCG, including ship-escorts and research missions.

Figure 1 shows 2010 vessel movements in Canada's Arctic waters from 120 active vessels and 978 total voyages (based on CCG data). The majority of these trips were made by merchant vessels (348), followed by tug boats engaged in community resupply (300) and tankers (169). Table 1 shows the emission estimates from these activities. The majority of emissions come from large commercial/merchant vessels such as general cargo vessels, bulkers, and tankers, collectively. Table 2 compares the Arctic portion of the marine shipping emission estimates to the other two Canadian regions: the west coast and eastern Canada (including east coast, the Great Lakes, and St. Lawrence Seaway), by activities. The Canadian Arctic marine shipping emissions currently count for less than 2% of the national marine emission totals. Compared to existing pan-Arctic estimates, e.g., Corbett et al (2010) for 2004 and Winther et al. (2014) for 2012, the Canadian portion of Arctic shipping emissions contributes to about 1% of current pan-Arctic shipping emissions.

To project future shipping emissions in Canadian Arctic waters, a number of factors were considered. Marine traffic is expected to increase in Canada's Arctic as both current and planned resource development projects come online. There are several operating and planned resource development projects in Canada's North that will require regular servicing by ships, including product transport, resupply vessels, drilling ships, and platforms. In addition, it is expected that Arctic tourism, also known as "eco-tourism", will increase in popularity as destinations become more accessible with thinning levels of ice as a result of a changing climate, and activities of the Canadian Coast Guard (CCG) will also likely increase.

An extensive review of ship traffic projections was conducted, utilizing environmental assessment reports for resource development and other projects in the Canadian Arctic that would be serviced by ships. As well, expected increases in other sectors, as noted above, were taken into account. Based on this information, a projection of the types and number of sailings



of vessels and their expected emissions in the future was developed (Environment and Climate Change Canada, 2015). To validate the forecast, the growth rates were compared with published data from companies and published studies related to shipping forecasts in the Arctic (e.g., Corbett et al. 2010). In predicting future shipping traffic, a limited number of transits via the Northwest Passage were assumed. Despite predictions of an ice-free Arctic, sea-ice variability, navigability and

dangerous weather remain constant challenges for Arctic shipping (Haas and Howell, 2015). Combined, these factors present an inherent degree of uncertainty in predicting future shipping levels in the Canadian Arctic.

Also included in Table 1 are the projected trips and emissions in Canadian Arctic water in 2030 by vessel classes. The largest anticipated increases in marine activities are from merchant vessels, particularly merchant bulk and passenger vessels. In estimating emissions related to projected shipping activities, the MARPOL Annex VI global cap on the sulphur

content of 0.5% for fuel oil used on board ships is assumed to be in place in the business-as-usual (BAU) scenario. For the controlled scenario, it is assumed that the Canadian Arctic is designated as an Emission Control Area (ECA) for $SO_x$, PM and $NO_x$ and therefore ships are subject to comply with the 0.1% sulphur in fuel limit, as well as the IMO Tier III $NO_x$ standards for new vessels. Under the BAU scenario, a nearly three-fold increase in total $NO_x$ shipping emissions is expected by 2030, mostly from merchant bulk vessel activities. The increases in $SO_x$ and PM emissions (compared to the present

levels) are moderate due to the global cap on sulphur content in fuel. In comparison, under the ECA scenario, the projected $NO_x$ emissions would be considerably reduced (from BAU levels) – to about two folds of the current (2010) level in total amount, while the $SO_x$ emissions would be reduced to below the current (2010) level by the more stringent regulation in sulphur content (0.1%).

## 3 Modelling system and simulation setup

The base model used for this study, GEM-MACH (**G**lobal **E**nvironmental **M**ulti-scale model – **M**odelling **A**ir quality and **CH**emistry), is an on-line chemistry transport model (CTM) embedded within Environment and Climate Change Canada's (ECCC) numerical weather forecast model GEM (Côté et al. 1998a,b; Charron et al., 2012). A limited area version of GEM-MACH has been in use as ECCC's operational air quality prediction model since 2009 (Moran et al., 2010). The representations of many atmospheric processes in GEM-MACH are the same as in the ECCC's AURAMS (A Unified

Regional Air-quality Modelling System) off-line CTM (Gong et al., 2006), including gas-phase, aqueous-phase, heterogeneous chemistry (inorganic gas-particle partitioning), secondary organic aerosol formation, aerosol microphysics (nucleation, condensation, coagulation, activation), sedimentation of particles, and dry deposition and wet removal (in-cloud and below-cloud scavenging) of gases and particles. Specifically, the gas-phase chemistry mechanism in GEM-MACH is a modified version of the ADOM-II mechanism (Stockwell and Lurmann, 1989) with 47 gas-phase species and 114 reactions;

aerosol chemical composition is represented by nine components: sulfate ($SO_4$), nitrate ($NO_3$), ammonium ($NH_4$), elemental carbon (EC), primary organic matter (POA), secondary organic matter (SOA), crustal material (CM), sea salt, and particle-bound water; aerosol particles are assumed to be internally mixed. The operational version of GEM-MACH uses a 2-bin





sectional representation of aerosol size distribution (Moran et al., 2010), i.e., 0 – 2.5 μm and 2.5 – 10 μm. The 2-bin configuration was also used for this study.

In this study, model simulations were conducted over a domain with a rotated latitudinal-longitudinal grid projection at a 15-km horizontal resolution. The domain is centered over the Canadian Arctic with its southern boundary extending south of the Canada-US border (see Figure 3). Eighty vertical, unevenly spaced, hybrid coordinate levels were used to cover between the surface and 0.1 hPa, with the lowest terrain-following model layer of about 20 m (GEM-MACH version 1.5). Several model upgrades and special considerations were made for this study:

1. *Representation of sea ice and snow cover in dry deposition.* Sea-ice cover from the Canadian Meteorological Centre's regional ice analysis system (Buehner et al., 2012) and snow cover and depth based on surface diagnostics were introduced to the dry deposition module to account for ice-snow cover conditions. In contrast, the base model (GEM-MACH v1.5) only takes into account permanent ice (glacier) cover in the dry deposition module. In addition, a different (lower) dry deposition velocity for $O_3$ over snow and ice was introduced following the recommendation of Helmig et al. (2007a).

2. *Chemical lateral boundary conditions.* Instead of using climatology-based lateral boundary conditions as is done in the operational GEM-MACH (see Pavlovic et al., 2016), the MACC-IFS (Monitoring Atmospheric Composition and Climate; Integrated Forecast System) chemical reanalysis for 2010 (Inness et al., 2013), available every 3-hours, was used to build daily chemical boundary condition files for the GEM-MACH Arctic domain. In addition, the southern boundary condition was enhanced by using the operational GEM-MACH forecast archives for the simulation time period in order to better represent transport of pollutants from North American continent.

3. *North American wildfire emissions.* Wildfire emissions were included in this study as it has been shown that northern boreal forest fires can be an important pollution source for the Arctic in summertime (Law and Stohl, 2007). Retrospective daily wildfire emissions per fire hotspot for the 2010 North American fire season were generated using the same methodology as in the ECCC's FireWork system; an air quality forecast system with representation of near-real-time biomass burning emissions (Pavlovic et al., 2016). The fire emission processing relies on the fire activity data from NASA's Moderate Resolution Imaging Spectroradiometer (MODIS) and NOAA's Advanced Very High Resolution Radiometer (NOAA/AVHRR), a fire behaviour prediction system - the Canadian Wildland Fire Information System (CWFIS; Lee et al., 2002), and the Fire Emission Production Simulator (FEPS) - a component of the BlueSky Modeling Framework (Larkin et al., 2009) to determine the daily total emission per fire hotspot. The per-fire-hotspot daily total emissions were then converted to hourly, chemically speciated, and grid-cell-specific emissions using the SMOKE emission processing system for use in GEM-MACH (see Pavlovic et al., 2016, for details). The fire emissions are treated as major point-source emissions in the model using the same Briggs plume-rise algorithm (Briggs 1975) as anthropogenic point-source emissions, with assigned "stack" parameters: 3 m, 773ºK, and 1 m s$^{-1}$ for stack height, exit temperature and velocity, respectively. Other fire plume injection schemes were tested in this study including one designed using satellite derived plume statistics. In



this scheme, the vegetation (biome) type based statistics for plume height and depth derived from 5-year satellite observations over North America (Val Martin et al., 2010) were used to determine plume centre height and vertical spread for flaming portion, taking into consideration of atmospheric stability, while the smoldering portion of the emission is evenly spread within the modelled planetary boundary layer (PBL). The test results showed that, while

the different plume injection schemes strongly impacted the modelled pollutant concentrations over the fire source region, the differences were considerably reduced at longer transport distances. As a result, the Briggs plume-rise algorithm was used in the final simulations for this study, as is used in the current FireWork system (Pavlovic et al., 2016), for distributing fire emissions.

4.  *Canadian marine shipping emissions*. The Canadian marine shipping emission inventories described in Section 2

above were further processed into model-ready point-source emissions. The MEIT database provides ship route polygons, vessel activities information associated with each route polygon, and link-based monthly emissions, by ship track, ship types, and fuel type. The database also includes stack parameters by ship type allowing plume-rise calculations in GEM-MACH. Table 3 shows the averaged stack parameters assigned to each fuel type.  To reduce data size and processing time, the more detailed ship types in the original MEIT database were aggregated, based on

vessel activities, into four classes: merchant passenger, merchant commercial, fishing, and other (as indicated in Table 1). The monthly emissions for the four classes were mapped onto model grids, along ship tracks, in a form of aggregated point sources (by class) and then further allocated to hourly emissions, by applying uniform temporal profiles for day-of-week and hour-of-day in the SMOKE emission processing system (http://www.cmascenter.org/smoke/). Figure 2 shows an example of the final processed model-ready marine

shipping emissions over Canadian waters used in this study. For assessing the impact of shipping emissions over the Canadian Arctic waters, the shipping emissions outlined by the red line in Figure 2 are turned on or off in the model simulations as discussed in Section 5.

Other anthropogenic emissions included in the model simulations are based on the 2010 Canadian Air Pollutant Emission Inventory (APEI) and the 2008 U.S. national emission inventories (NEI; https://www.epa.gov/air-emissions-

inventories/2008-national-emissions-inventory-nei-data), processed to hourly area and major point source emissions using SMOKE. Supplementary anthropogenic emissions from Emissions Database for Global Atmospheric Research-Hemispheric Transport of Air Pollutants (EDGAR-HTAP) v2 (see http://edgar.jrc.ec.europa.eu/htap_v2/; Janssens-Maenhout et al., 2012) were used for areas outside the North American continent. Biogenic emissions were estimated on-line using the BEIS v3.09 algorithms. Sea salt emissions were computed online within GEM-MACH based on Gong et al. (2003).

The simulations were carried out for the time period of March to October, 2010; the first month of the simulation is counted as spin-up and not included in the analysis. The eight-month simulation was conducted by a series of staggered 30-hour runs with a 6-hour (meteorology-only) "jump-back", starting at 00 Z daily, to allow meteorological spin-up; the meteorology is initialized at 00 Z using the Canadian Meteorological Centre's regional objective analyses while chemistry (delayed for 6-hours from run start time) continues from the preceding run.



## 4 Model evaluation - 2010 base case

The performance of GEM-MACH over the North America domain has been evaluated in a number of existing studies (e.g. Im et al, 2015a,b; Moran et al., 2011). As this is a first adaptation of the model for the Canadian Arctic domain, evaluation of model performance against available observations was carried out for criteria pollutants $O_3$, $PM_{2.5}$, $NO_2$, and $SO_2$, focused on the July-to-September period (the peak Arctic shipping season). The hourly observational data used for the evaluation were obtained from the Canadian National Atmospheric Chemistry (NAtChem; https://www.ec.gc.ca/natchem/) database which contains monitoring data from the National Air Pollution Surveillance (NAPS) network in Canada (http://www.ec.gc.ca/rnspa-naps/) and the U.S. Environment Protection Agency's Air Quality System (AQS) database for U.S. air quality data (https://aqs.epa.gov/aqsweb/documents/data_mart_welcome.html). For $O_3$, additional data from the World Data Centre for Greenhouse Gases (WDCGG: https://ds.data.jma.go.jp/gmd/wdcgg/) were also used. Data completeness criteria of 75% for daily data and 66% for the full period were used to screen the data. Figure 3 indicates the monitoring sites after the data completeness screening process was completed for the 4 criteria pollutants. Overall, most of the monitoring sites within the model domain are located over southeastern Canada (Ontario, Quebec and the Maritime Provinces) and southwestern Canada (British Columbia and Alberta). There are very few sites in central Canada and north of 55ºN. For this study, which focuses on the Canadian Arctic and northern regions, a significant challenge is the data sparsity over the region of interest: for the year 2010 (the base year for the study), Alert, on the northern tip of Ellesmere Island (82.45ºN, 62.51ºW), is the only air monitoring site in the entire eastern Canadian Arctic. For the purpose of model evaluation, the model domain is divided into three geographical sub regions based on general climatological and source characteristics: south-western Canada (49º – 55º N, west of 100º W), south-eastern Canada ([49º – 55º N, 75º – 100º W] and [44º – 53º N, 50º – 75º W]), and northern region ([55º – 90º N, 75º – 160º W] and [53º – 90º N, 50º – 75º W]) covering both the Canadian North and Alaska (U.S.). The division of the sub-regions is indicated in Figure 3.

### 4.1 Statistical scores

Various statistical measures were computed to evaluate model performance both at individual monitoring sites and as a group in the three geographical sub-regions. Table 4 presents the results of the regional (sector) statistical analysis using a few selected evaluation metrics chosen to characterise overall model performance for each of the criteria pollutants. Three sets of the statistical metrics are shown, based on hourly data ("Hly"), daily averages ("Dly"), and averages over the full July-to-September period ("Snl"). The statistical evaluation metrics are defined in Appendix A.

*$O_3$*

As shown in Table 4, for ambient $O_3$ concentrations, the model performs the best for the northern region in terms of model bias and error (e.g., MB, NMB, RMSE, and NMSE). There is an overall over-prediction of ambient $O_3$ concentrations by ~3 ppbv on average for the northern region, ~4 ppbv for the southwestern region, and ~6 ppbv for the southeastern region. The model's predictive skill increases with increased time scale as indicated by RMSE (or NMSE) with smallest errors for



seasonal averaged concentrations compared to daily and hourly concentrations. The Pearson correlation coefficient (r) for hourly $O_3$ is highest for the southeastern region (0.66) and lowest for the southwestern region (0.54); the correlation coefficients for daily concentrations are in similar range. As for the correlation coefficient for seasonal concentrations, a measure of spatial association between model and observation, the model performed the best for the Northern region (r =

0.66) but rather poorly for the two southern regions (r ≤ 0.30). The low spatial correlation between model and observation over the two southern regions may at least be partly attributable to the close proximity to the upstream boundaries (western and southern), where the model predictions are more influenced by the lateral boundary conditions. Overall, the model showed similar skill for modelling $O_3$ in the northern domain as the operational regional air quality models included in Im et al. (2015a) for modelling the North America domain in terms of NMSE, RMS, and r. Note that the statistical scores in Im et

al. (2015a,b) were based on domain-mean hourly data. The equivalent statistical scores were computed for this study and shown in Table 4 (in brackets). The averaging essentially minimizes spatial variability amongst the sites within the domain (or geographical sub-regions) and hence the statistical scores on the regional-averaged hourly data are much higher (in terms of RMSE, NMSE, and r) than the regional statistical scores based on hourly data at individual sites.

*PM$_{2.5}$*

The regional statistical scores for $PM_{2.5}$ show that the model performed best over the southeastern region with lowest NMB and NMSE and highest correlation. The model under-predicted $PM_{2.5}$ for the northern region with an overall negative bias of ~ -14% and the correlation is poor. It is worth noting, however, that there were very few sites with data available for evaluating model prediction of $PM_{2.5}$ in the northern and southwestern regions, 9 in each, compared to 36 in the southeastern region. In particular, of the nine northern sites, five are located in Alaska - 4 in Anchorage and surrounding area, and 1 in

Juneau, with the other four in Northwest Territories (NT). There were no $PM_{2.5}$ monitoring sites available over the entire eastern Canada North region. The four sites in NT include one located in Yellowknife, the only city (and the largest community) in NT, while the others are located in smaller communities (Inuvik, Norman Wells, and Fort Liard). As $PM_{2.5}$ contains both primary and secondary components, the ambient concentration at these northern sites is influenced by both long-range transport and local emissions. There are large uncertainties in both emission estimates and the spatial surrogates

used for distributing estimated emissions in the northern region (note that the Canadian emission inventory is at Provincial/Territorial level). These uncertainties contribute to the poor model performance at these northern sites. For example, as shown in supplementary material, the model over-predicted $PM_{2.5}$ at the Yellowknife site while under-predicting at the other NT sites (see Table S.1b). Furthermore, the modelled $PM_{2.5}$ at Yellowknife site is dominated by "crustal material" (see Figure S.1) which is a major component of primary PM emissions in NT. The spatial surrogates used for

crustal material are paved roads and mine locations. The paved road network in NT used in processing the 2010 emission inventory was very limited, mainly concentrated in Yellowknife and its surroundings. As for mine locations, the surrogate was based on place-of-work data from the 2006 Canadian census for mining industry (http://www12.statcan.ca/census-recensement/2006/rt-td/pow-ltd-eng.cfm), which can lead to allocating  mining related emissions to cities rather than actual mining operation sites as many mining company employees work at headquarters which tend to be located in cities (e.g.,



Moran et al., 2015). For the Inuvik site on the east channel of the Mackenzie Delta, the model under-prediction may be partially attributable to an under-estimation of emissions from the oil fields in Prudhoe Bay on Alaska's North Slope in the U.S. 2008 emission inventory (https://www.epa.gov/air-emissions-modeling/20072008-version-5-air-emissions-modeling-platforms ).

### $NO_2$

For predicting $NO_2$, the model performed the best, comparatively, for the Northern sites overall with the lowest NMB (8.3%), RMS (5.6 ppb), and highest R (0.56), based on hourly data (Table 4). However, the relatively small overall bias may be misleading as there are large positive and negative model biases at the individual northern sites (Table S.1c). This is indicated by the large NMSE value (104%). The 10 northern sites here include 4 in NT, where the model generally under-

10 predicted, and 6 in the lower Athabasca oil sands region in Alberta, where the site-specific model biases, in terms of NMB, varied between -64% (at Fort Chipewyan) and 143% (at Syncrude UE1), indicating significant heterogeneity. Again the model performance at these sites is influenced by the uncertainties (challenges) in estimating and representing emissions in these regions of Canada (ECCC & AEP, 2016, Zhang, et al., 2017). Also note that the $NO_2$ observations from the NAPS network were reported in an increment of one ppb which will have a considerable impact on the statistical scores particularly

at more remote sites where $NO_2$ concentrations are low and of the order of < 1 ppbv. The high correlation between the modelled and observed seasonal averaged concentrations indicates however that the model captured the geographical distribution of the regional $NO_x$ sources and plumes reasonably well.

### $SO_2$

The statistical scores for model prediction of $SO_2$ are considerably poorer than those for the other criteria pollutants

discussed above, with large biases (in terms of NMB) and errors (in terms of NMSE). Note that the reference unit for $SO_2$ in this comparison is $\mu g\ m^{-3}$ at standard atmosphere (0ºC) because the reported $SO_2$ concentrations were converted to this unit in the NAtChem database. There are several factors to be considered when interpreting these statistical scores. Firstly, the group statistical scores for the northern sites are largely influenced by the sites located in the lower Athabasca oil sands region in Alberta and the Peace region of northeastern British Columbia (see Table S.1d) with considerable oil and gas

industries there. The monitoring sties in these regions are located at or near industrial facilities. The modelled $SO_2$ at these locations are primarily driven by the model emission inputs. There are large model biases at these locations indicating again potential deficiencies in emission estimates and processing in these regions (e.g., spatial and temporal allocation of the annual emissions; e.g., ECCC & AEP, 2016; Zhang et al., 2017; Gordon et al., 2017). Secondly, similar to the case of $NO_2$ discussed above, there is also a precision issue with monitoring data reporting: $SO_2$ concentrations are reported at one ppb

(or ~ 2.86 $\mu g\ m^{-3}$) increment. This is particularly problematic for model evaluation at more remotes sites (such as those in the Northwest Territories) where $SO_2$ concentrations are generally below 1 or 2 ppb and the reported concentration values toggle between 0, 1, and 2 ppb (or 0, 2.86, and 5.72 $\mu g\ m^{-3}$ after conversion in the NAtChem database). Again, despite the large mean bias (~ 10 $\mu g\ m^{-3}$) and RMSE (seasonal, ~ 16 $\mu g\ m^{-3}$), the correlation between the modelled and observed seasonal





averaged $SO_2$ concentrations in the northern region is high (r = 0.90), indicating that the model was able to capture the spatial distribution/structure of the observed concentrations (averaged over the July-to-September period).

**4.2 Time series**

Figures 4 to 7 show the model-observation comparison of the regional averaged hourly time series of $O_3$, $PM_{2.5}$, $NO_2$, and

$SO_2$ for the three sub regions. Given the monitoring site locations, the "Northern" regional average really represents only northwestern Canada (and Alaska in the case of $O_3$ and $PM_{2.5}$). The regional $O_3$ time series show that the overall temporal variability is smallest at the northern sites and greatest at the southeastern sites. The southwestern sites are strongly dominated by the diurnal variations particularly during mid-summer period (July, August) indicating the influence of local photochemistry, while the southeastern sites are strongly influenced by regional/synoptic events. These characteristics are

generally captured well by the model. A positive bias in model prediction is evident, particularly as seen in the $O_3$ nighttime minima which the model consistently over-predicted. There were also more pronounced over-prediction events during the month of August at the northern and southwestern sites. This may be associated with large wild fire events in British Columbia during that period. The model tends to over-predict $O_3$ in fire plumes (Pavlovic et al., 2016; Gong et al., 2016) particularly within a short transport time. A number of factors may be contributing to the over-prediction, including

uncertainties in emission factors and the lack of representation of aerosol shading in the model which may lead to an overestimation of photolysis rates in fire plumes. The possible causes are currently under investigation.

The northern regional averaged $PM_{2.5}$ time series is dominated by variations at small scales implying a strong influence of primary components from local sources at these northern sites, while the southeastern regional $PM_{2.5}$ time series is more controlled by variations at larger scales, or regional events, implying the dominance of secondary components and/or

regional sources. The southwestern time series contains the signature of both local and regional influences with the main regional events in August coinciding with the major wild fire events in BC at that time. The model captured the general trends well particularly for the regional events, while it had difficulty tracking the local-scale variations, which is not unexpected given the model resolution.

The regional averaged $NO_2$ time series show rather distinct diurnal fluctuations on top of a nearly 7-day cycle particularly for

the southwestern sites. The northern averaged time series has a similar pattern but with smaller amplitudes. The model predictions compare well for the northern and southwestern sites. For the southeastern sites the model captured the general trend well but there is a tendency for more significant over-prediction particularly at the beginning of July. Significant over-predictions of $NO_2$ over eastern Canada during this time period from the operational GEM-MACH forecast were also shown in the evaluation of Moran et al. (2011). It should be noted that the southeastern sites in this study are in close proximity to

the southern boundary and are more likely to be influenced by the model southern boundary condition which comes from the operational GEM-MACH forecast archives.

As a reflection of the $SO_2$ regional statistical scores discussed above, the comparison of regional averaged time series of the observed and modelled $SO_2$ for the northern region is strongly influenced by the sites located near oil and gas facilities,



particularly the two sites in northeastern BC (Taylor Town site and Pine River Gas Plant). The observations show occasional spikes of very high concentrations (200 – 300 μg m$^{-3}$) while the model simulation shows more regular daily peaks in a similar range, indicative of possible deficiency in the emission inventory and the emission processing for these facilities[1]. The comparison for the southwestern region is considerably better where model simulation tracks the observed general trend

at the regional scale very well. The comparison for the southeastern region shows a general over-prediction by the model. In particular, the modelled group-averaged time series shows a higher regional baseline level than indicated by the observations. As shown in Figure 3d, these southeastern sites are situated under the influence of the model's southern boundary and the modelled average $SO_2$ concentration over the July-August-September period shows a regional plume originating from the southern boundary reflecting the influence of major $SO_2$ source area in the Ohio River Valley. Note that

the emission inputs used by the operational GEM-MACH forecast in 2010, the basis for the model southern chemical boundary condition for the current study, were based on the 2006 Canadian, 2005 U.S., and 1999 Mexican national emission inventories (Moran et al., 2011). Due to the various U.S. EPA's emission control programs in recent years (e.g., Acid Rain Program, NOx Budget Trading Program, Clean Air Interstate Rule, see https://www.epa.gov/airmarkets), $SO_2$ emissions over eastern U.S. have reduced considerably between 2005 and 2010. The model over-prediction of ambient $SO_2$ in the

southeastern region in this study can therefore be, at least in part, attributed to the possible over-prediction of $SO_2$ from the operational GEM-MACH over the U.S. Northeast.

*Canadian high Arctic site, Alert*

Several long-term monitoring measurements of atmospheric constituents have been carried out at ECCC's Alert baseline observatory located at the northern tip of the Ellesmere Island (82.45ºN, 62.51ºW); one of the Global Atmosphere Watch

global network stations. For the year 2010 the measurements included, in addition to $O_3$ (continuous, hourly), inorganic aerosol components from weekly high-volume sampler (Sirois and Barrie, 1999; Sharma et al., 2004), organic carbon (OC) and elemental carbon (EC) using a thermal method from bi-weekly quartz filter samples (Huang et al., 2006), and equivalent black carbon (EBC) from aerosol light absorption measurement using an Aethalometer (Shama et al., 2017). These data are all used for evaluating model prediction at this high Arctic location. The comparisons of the modelled and observed time

series of $O_3$, sulfate, EC and OC (OM – organic matter) over the June-to-September period are shown in Figure 8.

The model is seen to predict $O_3$ very well at this high Arctic site: the modelled $O_3$ time series tracks closely with the observations reaching a minimum at the end of July and the beginning of August and then rising steadily throughout late August and September. The model did not predict the low ozone event observed at the beginning of June. The low ozone event may be the result of ozone depletion involving bromine chemistry within the Arctic marine boundary layer (Barrie and

30 Platt, 1997) which is not represented in this version of the model. The modelled sulfate also compared well, particularly in terms of general trend and magnitudes, with the non-sea-salt sulfate measurements based on weekly samples. The modelled

---

[1] These two sites are collocated with two upstream oil and gas (UOG) facilities with large $SO_2$ emissions from flaring stacks. However, due to the lack of stack information in the UOG inventory, the $SO_2$ emissions from these facilities were allocated at the ground level in emission processing.



EC is compared with both EBC derived from the continuous Aethalometer measurement and EC measurement using a thermal desorption method from quartz filter sampling (bi-weekly in 2010). It can be seen that while the modelled EC is overall biased low compared to the EBC from the Aethalometer measurement, and biased lower still compared to the bi-weekly EC measurement, the model however captured the general trends shown in both observation sets. In particular, the

event in early July was captured well by the model, which is attributable to biomass burning emissions from northern Canada. Sharma et al., (2017) discussed in depth the various techniques for measuring black carbon mass at the Alert observatory. In particular, comparisons amongst the BC mass measurements using the different techniques showed that, based on the data collected from March 2011 to December 2013 at Alert, EC mass based on the thermal method is highest over summer months followed by the EBC mass estimate from the Aethalometer measurement; both are significantly greater

than the refractory BC (rBC) mass measurement using a Single Particle Soot Photometer (SP2). The ambient concentration of BC at Alert is nominally lowest during summer months due to reduced long-range transport from mid and low latitudes and more efficient removal due to wet deposition during this season (e.g., Sharma et al., 2004). The EC measurement at Alert is most uncertain during summer months due to very low EC concentrations and is likely biased high due to the possible presence of residue pyrolyzed OC and carbonate carbon (Sharma et al., 2017). A combination of EC and rBC (i.e.,

an arithmetic average of the two) was recommended as a best estimate of BC mass at Alert for comparison with chemical transport models. rBC measurements were not available in 2010 but, based on the EC-to-rBC ratio given by Sharma et al. (2017), of 3.5 for summer, the equivalent "best estimate" can be obtained from the EC measurement with a scaling factor of $0.5(1 + \alpha)/\alpha$ (where $\alpha$ is 3.5 here). The scaled EC is indicated in Figure 8(c) with solid dots connected by the dashed line. However, one needs to be careful in comparing the modelled aerosol EC component with BC measurements as they may not

be strictly comparable, depending on the measurement techniques (e.g., Petzold et al., 2013; Sharma et al., 2017) and how EC (or BC) is modelled (including emission input). The modelled organic aerosol component (POA+SOA) is compared with the bi-weekly measurement of OC from the thermal desorption method. For this comparison the measured OC is converted to OM by applying an OM/OC ratio of 1.8. The total OC (TOC) from the OC/EC analysis includes OC released at 550°C and pyrolyzed OC (POC) plus inorganic carbonate carbon (CC) released at 850°C. The estimate of CC fraction of POC+CC is

40% at Alert in summer time. The CC fraction was removed from the TOC measurement for the comparison in Figure 8(d) based on the CC-to-POC+CC fraction. The measured OC component (at 550°C) is also shown in Figure 8(d), indicating that this is the dominant component of measured TOC at this site. Overall the model under-predicted organic aerosol component at this site compared to the measurement based on the OC/EC analysis but again captured the event in the beginning of July (as in the case of EC comparison above) associated with long-range transport of biomass burning pollutants. Recent

observations conducted in the Canadian Arctic have suggested possible marine secondary organic aerosol production over the Arctic Ocean during summer time from oceanic/biological sources (e.g., Willis et al., 2016), which may explain at least in part the model under-prediction of organic aerosols (Gong et al., 2017).





The evaluation results presented in this section demonstrate that GEM-MACH's skill in predicting ambient $O_3$ and $PM_{2.5}$ in the Canadian northern and Arctic region is comparable to the skill level of the current operational air quality forecast models in North America and Europe. The model has reasonable skill in predicting $NO_2$ and $SO_2$ in the north at a regional scale; at local scales the model prediction is strongly influenced by emission inputs. The evaluation indicates a deficiency in representing local emissions in the remote north and the need for improved emission estimates and representation for the oil and gas facilities in northeastern British Columbia and the Athabasca oil sand region in northern Alberta. There is also a significant data gap in northern Canada, particularly the eastern Arctic, for air quality monitoring and for model evaluation. The model however is able to simulate well the observed ambient $O_3$, and some of the PM components at Alert, the only air quality monitoring site in the eastern high Arctic.

## 5 Impact of shipping emissions on Arctic air pollution

The impact of shipping emissions in the Canadian Arctic is assessed by comparing pairs of model simulations, with and without the Canadian portion of the Arctic shipping emissions, under three scenarios: current (2010), projected 2030 BAU, and 2030 with ECA (see Section 2 above). To isolate the impact of shipping emissions, only shipping emissions were changed between the different scenarios, while meteorology, land-use, and other emissions (such as non-shipping anthropogenic emissions and wild fire emissions) remained the same for all scenario simulations. The analysis is focused on the July-August-September (JAS) peak Arctic shipping period. It should also be stated that the impact is mostly assessed in relative terms in this study for these considerations: 1) since the modelled future scenarios do not reflect changes in forcing factors other than shipping emissions, it is more meaningful to assess the modelled relative response to the emission changes; 2) robustness in using a model to assess relative changes – past studies involving multi-models have shown that, despite the large difference in performance amongst models, only relatively minor differences were found in the relative response of concentrations to emission changes (Jones et al., 2005; Hogrefe et al., 2008).

### 5.1 On ambient air concentration of criteria pollutants

The modelled JAS averaged ambient concentrations of $O_3$, $PM_{2.5}$, $NO_2$, and $SO_2$ and the corresponding contributions from Arctic shipping are shown in Figures 9 – 12, with a focus on the Canadian northern and Arctic regions. The percentage ship contributions shown were computed as

$$\frac{conc(with\ arctic\ shipping)_{i,j} - conc(without\ arctic\ shipping)_{i,j}}{conc(with\ arctic\ shipping)_{i,j}} \times 100\ (\%),$$

where $i$ and $j$ denote pollutants (e.g., $O_3$ $PM_{2.5}$, $NO_2$ and $SO_2$) and scenarios (i.e., 2010 base-case, 2030-BAU, and 2030-ECA), respectively.

The modelled ambient $O_3$ concentrations averaged over the JAS period range between 20 and 25 ppbv over most of the eastern Arctic (Figure 9(a)). The relatively high ambient concentrations over Greenland are due to the high elevation. The



Arctic shipping emissions contribute to less than 1% of the JAS averaged $O_3$ concentration at the present level (or 2010 base-case); the impact is mostly felt between 50 W and 100 W (Figure 9(b)) and Mackenzie Bay in the west. At the projected 2030 BAU level, the model predicted considerably greater shipping contributions, showing up to 5% of the JAS averaged ambient $O_3$ concentration (Figure 9(c)); the area where shipping emissions contribute greater than 0.5% extends to almost all

of the eastern Canadian Arctic (or Nunavut Territories, NU). This is in response to the projected increase in $NO_x$ emissions from Arctic shipping in the 2030 BAU scenario. For the 2030 ECA scenario, the model predicted shipping contributions to $O_3$ concentrations are reduced compared to the 2030 BAU scenario but are still greater than the present 2010 base-case level (Figure 9(d)), particularly along Davis Strait and Baffin Bay. This is consistent with the fact that projected $NO_x$ emissions from Arctic shipping in 2030 under ECA are intermediate between current 2010 and 2030 BAU levels (see Table 1).

The modelled JAS averaged ambient $PM_{2.5}$ concentrations show a general south-to-north decreasing gradient, from a few micrograms per a cubic-metre in the sub-Arctic regions to below 0.1 $\mu g\ m^{-3}$ in the High Arctic (Figure 10(a)). As $PM_{2.5}$ consists of both primary and secondary components, the impact of shipping emissions accentuates the shipping channels (Figure 10(b), (c), and (d)) more than is the case for $O_3$. The contributions from Arctic shipping emissions to the JAS averaged $PM_{2.5}$ concentrations are in the range of 1-5% along eastern Perry Channel, Pond Inlet, and north of Baffin Island

and generally < 0.5% over land, at the present level (2010 base-case; Figure 10(b)). At the projected 2030 BAU level, the contributions from Arctic shipping emissions to ambient $PM_{2.5}$ concentrations are predicted to increase to 5 – 20% over the main shipping channels, particularly along the east coast of Baffin Island and Lancaster Sound area (Figure 10(c)). The greater contribution in this case is due to the projected increase in both primary PM emissions and PM precursor emissions (of $SO_2$, $NO_x$, and VOCs) from shipping; this is evident from examining the shipping contributions to individual PM

components. The components contributing to the increase in total PM due to shipping include primary PM, such as elemental carbon, primary organics, and crustal material, and secondary PM, e.g., sulfate, ammonium, and nitrate, (see S.2 – 7 in supplementary materials). Again, for the 2030 ECA scenario, the model predicted a considerably reduced contribution from shipping in comparison with the 2030 BAU scenario (Figure 10(d)), primarily resulting from the drastic reduction in sulfur emissions if ECA is in effect over the Arctic waters.

For $NO_2$ and $SO_2$, both primary pollutants, the model shows that Arctic shipping emissions make major contributions to ambient concentrations over and near the Arctic waterways. The modelled JAS averaged ambient concentrations of $NO_2$ and $SO_2$ are 0.02 – 0.1 ppbv and 0.001 – 0.01 ppbv, respectively, over the eastern low- and sub-Arctic, and generally below 0.02 ppbv and 0.001 ppbv, respectively, over the High Arctic (Figure 11(a) and Figure 12(a)). The relatively elevated concentrations around the lower east coast of Greenland primarily reflect shipping emissions based on the 2010 HTAP

inventory (used in this study for areas outside North America, see section 3 above). At current (2010) levels, based on the model simulations, the Arctic shipping emissions contribute to 10 – 50% (Figure 11(b)) and 20 – 100% (Figure 12(b)) of the ambient $NO_2$ and $SO_2$ concentrations, respectively, over the Arctic shipping channels. The contributions are greatly increased at the projected 2030 BAU level in the case of $NO_2$, to > 50% over most of the shipping channels (Figure 11(c)), in response to a nearly three-fold increase in $NO_x$ emissions from Arctic shipping. In contrast, the contributions from Arctic





shipping to ambient $SO_2$ concentrations are only moderately higher at the projected 2030 BAU level compared to the present 2010 level (Figure 12(c) vs. Figure 12(b)). This is in response to a more moderate (~ 32%) increase in $SO_2$ emissions over the 2010 level (assuming the global cap of 0.5% on sulfur content in fuels used onboard ships is in effect, i.e., MARPOL Annex VI Regulation 14.8). Under the 2030 ECA scenario, there is a moderate decrease in the Arctic shipping contribution

5  to ambient $NO_x$ concentration (Figure 11(d) vs. Figure 11(c)), while there is a drastic decrease in the Arctic shipping contribution to the ambient $SO_2$ concentration (Figure 12(d) vs. Figure 12(c)). This is in accordance with the reductions of 35% and 79% in $NO_x$ and $SO_2$ emissions from the 2030 BAU level when assuming the NA ECA controls are in effect over the Canadian Arctic waters. In fact, the ECA control on sulfur emissions would bring down the shipping contribution to the ambient $SO_2$ concentration to below the current 2010 base-case level.

### *Statistical assessment by geographical sectors*

A more quantified (and area-specific) assessment of the impact of ship emissions was carried out by dividing the area of interest into 9 geographical sectors (see Table 5; also indicated on Figure 9(b)) and shipping contribution statistics were computed for each of the geographical sectors. Table 6 summaries the mean, median, and maximum percentage Arctic

shipping contributions to the JAS average ambient concentrations of criteria pollutants for each of the 9 sectors. Generally speaking, the shipping impact is greater over the eastern Canadian Arctic than the western Canadian Arctic, due to the proximity of the area to the Arctic shipping channels. In addition, the western region of the Canadian Arctic is more strongly impacted by North American boreal forest fire plumes during the summer season with relatively higher background concentrations of these criteria pollutants than the eastern region (e.g., Gong et al., 2016).

At the current level (2010), the contribution statistics for $O_3$ show that both mean and median percentage contributions from Arctic shipping are relatively uniform over the eastern sectors, with slightly higher contributions over sectors E3 and E4 at around 0.3% and the rest of the eastern sectors at around 0.2%. As for $PM_{2.5}$, the shipping contributions are higher over the north eastern sectors (north of 60N) and highest (> 0.5% in mean value) over sectors E3 and E6, both of which are in close proximity to the Arctic shipping routes (see Figure 1). Shipping contributions to ambient concentrations of $NO_2$ and $SO_2$ are

much higher in comparison to $O_3$ and $PM_{2.5}$, and are again highest over sectors E3 and E6 (with mean percentage contributions: > 10% for $NO_2$ and ~ 20% and higher for $SO_2$). Shipping contributions over E4 (in close proximity to ship traffic over Hudson Bay) and W6 (in close proximity to the Beaufort Sea) are also pronounced in this case. Sector E6 has the highest relative contribution from Arctic shipping emissions which is attributed to its proximity to northern Arctic shipping routes and, as well, being the most remote region with lowest background concentrations and hence most sensitive area to

local emissions. Notice that the statistics shown in Table 6 imply that the probability distribution functions (PDFs) of the percentage shipping contributions for pollutants $PM_{2.5}$, $NO_2$, and $SO_2$ are highly skewed (i.e., large differences between means and medians, and confirmed by further statistical analyses undertaken but not shown here), while the percentage contributions for $O_3$ are relatively normally distributed (i.e., small differences between mean and median values). This is consistent with $O_3$ being a secondary pollutants, and, with its relatively long atmospheric lifetime, $O_3$ has much higher



background ambient concentrations (and hence smaller relative contribution from shipping emissions) compared to the other pollutants assessed in this study.

At the projected 2030 BAU level, there is an overall increase in the shipping contributions to ambient concentrations of the criteria pollutants over all sectors (with the exception of sector W1 which is far away from Arctic shipping routes). The average contribution from shipping to ambient $O_3$ concentrations increases to about 1% or higher over the north eastern sectors (from < 0.4% currently). The average shipping contribution to the ambient $PM_{2.5}$ concentration increases more significantly over sectors E3, E5, and E6, e.g., 2% over E3 compared to 0.6% at the current level. The most significant contribution of ship emissions to ambient levels of pollutants is for $NO_2$ for which average contributions are over 30% in sector E6 and reaching 20% in sector W3. The increase in shipping contribution to ambient $SO_2$ concentrations at the projected 2030 BAU level is overall predicted to be more moderate, compared to the case of $NO_2$ for most of the sectors, except for sector W3 where the average shipping contributions increase to nearly 30% from just over 10% at the current level. As mentioned above, for $SO_2$, the projected increase in shipping activity is partly offset by the global sulfur cap coming into effect in 2020 (or by 2025 with a five year delay, i.e., MARPOL Annex VI Regulation 14.8). If the same North American ECA regulations were to be applied within the Arctic waters in 2030 (i.e., with 0.1% sulfur cap and IMO Tier III $NO_x$ standard for new vessels, the 2030 ECA scenario), the shipping contribution to ambient $SO_2$ concentrations would be well below the current (2010) level, and the shipping contribution to ambient $PM_{2.5}$ would be brought roughly back to the current level. There would be reductions in shipping contributions to the ambient $NO_2$ and $O_3$ concentrations compared to the 2030 BAU scenario but the contributions would still be greater than the current level. This is in line with the less stringent regulation (in comparison to sulfur) on $NO_x$ under the NA ECA.

*Population-weighted concentrations*

Since criteria pollutants are closely related to health effects, it is pertinent to look at the impact of Arctic shipping emissions in terms of population-weighted concentration. Population-weighted concentration is often used in population exposure and health effect analysis (e.g., Ivy et al., 2008, Mahmud et al., 2012). It is calculated as:

$$\frac{\sum_{i=1}^{n} pop_i \times conc_i}{\sum_{i=1}^{n} pop_i}$$

where $i$ designates each computational grid cell and $pop_i$ and $conc_i$ denote population and concentration, respectively, at grid cell $i$. Here population weighted concentrations of the criteria pollutants are calculated for Canada's eastern and western Arctic, defined as north of 60º N, (60º W – 100º W) and (100º W – 140º W), respectively.

Figure 13 shows the gridded population density over the model domain based on the 2010 US and 2011 Canadian population data. As shown, over the eastern Arctic, the populations are mostly distributed along coastlines in small isolated communities and are thus more directly subjected to the impact from shipping emissions than over the western Arctic. The time series of the population weighted concentrations of $O_3$, $PM_{2.5}$, $NO_2$, and $SO_2$ and the corresponding shipping contributions over the June-September period are plotted in Figure 14 (a-d). Overall the population-weighted concentrations



are higher in the western Canadian Arctic than in the east. The communities and population centres are larger in the west and, as well, the western Arctic is more affected by North American boreal forest fire emissions in the summer months (e.g., Alaska, northern British Columbia, and northern Prairies; Gong et al., 2016). Conversely, the relative contributions from ship emissions are higher in the east than in the west, due to the closeness of the eastern communities to the shipping channels

and cleaner background air. The population weighted $O_3$ concentration over the eastern Arctic shows an overall summer minimum in July and a slow recovery during late summer and early fall, which is consistent with the general $O_3$ seasonal trend observed at the Arctic sites (Helmig et al., 2007b). In contrast, the time series for the western Arctic shows higher values in mid-July and early August likely due to biomass burning impact in the region. The shipping contribution is relatively uniform over the peak shipping season (JAS) over the eastern Arctic, whereas over the western Arctic the shipping

contribution is greater over the later part of the shipping season (September) than the early part (i.e. July-August) when the region is impacted by biomass burning plumes (Gong et al., 2016). Table 7 shows the statistics of ship contributions to population-weighted concentrations over the eastern Arctic (i.e., mean, median, maximum). When compared to the geographically based sectoral statistics above, the ship impacts on population-weighted pollutant concentrations are larger particularly over the eastern Arctic (in terms of relevance to health impact). Similar to the sectoral statistical assessment

above, the application of ECA-like controls over Arctic waters (in the projected 2030 emission scenario) would result in an important reduction in shipping contributions to the ambient air pollution. In the case of $PM_{2.5}$ and $SO_2$, the ECA-like control would bring the projected 2030 shipping contributions down to, or well below, current (2010) levels, respectively.

It is interesting to compare the above model based assessment of Arctic shipping emissions on air quality with measurement

based analysis. Aliabadi et al. (2015) conducted an analysis on the air quality measurements collected during the 2013 shipping season from two monitoring stations in the eastern Canadian Arctic: Cape Dorset (on Foxe Peninsula at the southern end of Baffin Island) and Resolute, in Nunavut, both located near Arctic shipping channels. Using back trajectories and high-resolution ship position data, they estimated that ship emissions contributed to cumulated concentrations (equivalent to dosage) of $NO_x$, $O_3$, $SO_2$, and $PM_{2.5}$ of: 12.9 – 17.5%, 16.2 – 18.1%, 16.9 – 18.3%, and 19.5 – 31.7%,

respectively, at Cape Dorset (southern site), and 1.0 – 7.2%, 2.9 – 4.8%, 5.5 – 10.0%, and 6.5 – 7.2%, respectively, at Resolute (northern site). This may be loosely compared to the model assessment based on population-weighted concentration above (Table 7) bearing in mind the difference in metrics, as it also is weighted towards small coastal communities. Ship contributions to $O_3$ and $PM_{2.5}$ concentrations were estimated to be higher based on the measurements than from the model assessment. This may be due in part to the methodology used in Aliabadi et al., where the concentrations exceeding the

deemed "background level" was attributed entirely to ship influence whenever a back-trajectory crossed a ship location. In the case of $O_3$ and $PM_{2.5}$, which are either purely or partly secondary pollutants with relative long lifetimes, this is likely to over attribute ship influence as the air parcel could well be influenced by other sources as well as ship plumes. In contrast, the ship contributions to $NO_2$ (or $NO_x$ in the case of measurement based analysis) and $SO_2$ were estimated lower from the measurements than from the model assessment. This can also be expected as the measurement sites were often influenced by



local sources (e.g., garbage burn, off-road use of diesel, aeroplane landings and take-offs) which are not represented well in the model simulations. Combined with instrument lower detection limits (LDLs), the background levels in the measurement analysis for $NO_x$ and $SO_2$ are much greater than the corresponding modelled background levels, which leads to greater ship contribution (in relative sense) from the model assessment than from the measurements.

## 5.2 On deposition of S and N

The impacts of Arctic shipping on the deposition of sulfur and nitrogen at current 2010 and projected 2030 levels were also examined in this study. The model computes both dry and wet deposition fluxes of various sulfur- and nitrogen-containing species. They include, for dry deposition, $SO_2$, p-$SO_4$, NO, $NO_2$, $HNO_3$, $NH_3$, HONO, $RNO_3$ (organic nitrate), PAN (peroxyacetyl-nitrate), p-$NO_3$, p-$NH_4$, and, for wet deposition, $HSO_3^-$, $SO_4^=$, $NO_3^-$, $NH_4^+$. The modelled wet deposition includes both "rain-out", i.e., tracer transfer from cloud water to rain water due to precipitation production (autoconversion/collision/coalescence), and "wash-out", i.e., below cloud scavenging of aerosol particles and soluble gases by falling hydrometeors, as described in Gong et al. (2006).

Shown in Figure 15 and 16 are the modelled total sulfur and nitrogen deposition fluxes accumulated over the JAS period and the contributions from Arctic shipping emissions. The deposition fluxes are shown here for the 2010 base case only, due to the similarity in the geographical distribution patterns between different scenarios, while the shipping contributions are shown for all three scenarios. Overall the deposition fluxes are much lower over the Arctic region compared to lower latitudes. The total sulfur deposition (over the three-month period) ranges from 0.2 – 0.5 kg of S per hectare over the Canadian sub-Arctic to 0.02 – 0.05 kg of S per hectare over the Canadian high-Arctic; the corresponding ranges for total nitrogen deposition are 0.1 – 0.5 and 0.01 – 0.05 kg of N per hectare, respectively. These levels are in general accordance with previous model estimates (e.g., Hole et al., 2009, Vet et al., 2014). The contribution to total sulfur deposition from Arctic shipping is relatively small, below 5%, at the 2010 base level, however the contribution from shipping increases to up to 20% along the coast of Baffin Bay in the 2030 BAU scenario. The 2030 ECA scenario brings down the shipping contribution to generally below the current 2010 level except for along the coast of Baffin Bay where a major increase in shipping activity from increased economic development is projected. The shipping contribution to total N deposition is comparable to the case of S deposition at the current 2010 level, but it increases substantially under the 2030 BAU scenario, up to 50%. With assumed ECA-like regulation, the shipping contribution is slightly reduced but is still much greater than at the current 2010 level.

The statistics of shipping contributions to total deposition of S and N by the 9 geographical sectors are shown in Table 8. Similar to the cases of ambient $SO_2$ and $NO_2$, the sectors most affected by Arctic shipping emissions are the four northern-most sectors in the east (E3 – E6). However, in contrast to the cases of ambient $SO_2$ and $NO_2$ where Arctic shipping contributions are much more important, the contributions to total depositions of S and N from Arctic shipping are much less substantial. This is in part due to the dominance of wet deposition in the total deposition of S and N (as is discussed later) over the region of interest. The dominance of wet deposition over dry deposition over northern Canada is also found in a





recent global assessment study of Vet et al. (2014), and it is consistent with the fact that the area has relatively low emissions and moderate precipitation amounts (particularly during the summer months). While dry deposition is more associated with ambient (or near-surface) concentrations, wet deposition is more associated with concentrations aloft (i.e., at cloud levels and through the vertical column) and hence is more affected by long-range transport and distant sources. Due to its moderate

solubility and fast oxidation pathways in the aqueous phase, $SO_2$ can be efficiently scavenged into cloud droplets, oxidized into sulfate, and be transported and deposited (through rain-out) long distances from its sources. Similarly, both $NH_3$ and $HNO_3$ can be readily scavenged by cloud water and both contribute significantly to the wet deposition of N: gaseous $NH_3$ is highly soluble and, once absorbed by cloud water, will mostly be in the form of ammonium ion ($NH_4^+$); $HNO_3$ is extremely soluble and will quickly dissociate into nitrate ions ($NO_3^-$) once dissolved in cloud water (Seinfeld and Pandis, 1996).

The deposition of S and N is of importance in considering ecosystem impacts, e.g., acidification and eutrophication of terrestrial and aquatic systems (Reuss and Johnson, 1986; Bouwman et al., 2002). To this end, land-cover weighted deposition fluxes of S and N for three primary land-cover types found in the Canadian Arctic, namely lakes, tundra, and barren/desert, were computed and the contributions to the land-cover weighted deposition from Arctic shipping are examined. Figure 17 shows the gridded land-cover fractions for the three land-cover types based on the U.S. Geological

Survey's (USGS) global land cover characteristics (GLCC) database at 1-km resolution (see https://lta.cr.usgs.gov/glcc/globdoc2_0). Similar to the population-weighted concentration, the land-cover-weighted deposition is calculated as:

$$\frac{\sum_{i=1}^{n} frac_i \times A_i \times depo_i}{\sum_{i=1}^{n} frac_i \times A_i}$$

where $frac_i$, $A_i$, and $depo_i$ are gridded land-cover fraction (for a given land-cover type), grid area, and deposition flux,

respectively, at grid cell $i$.

Forsius et al. (2010) estimated critical loads of acidity (S and N) for terrestrial ecosystems north of 60° latitude using a Simple Mass Balance (SMB) model, and found that in northern North America the lowest critical loads (or most sensitive regions) occur in eastern Canada. Table 9 shows the land-cover-weighted depositions of S and N (total, as well as, separately, dry and wet) for eastern Canadian Arctic (60° W – 100° W, 60° N – 90° N) over the JAS period and the respective

contributions from Arctic shipping. At the current level, land-cover-weighted total S deposition over the eastern Canadian Arctic varies from 73 g ha$^{-1}$ over barren land to 143 g ha$^{-1}$ over lakes for the three-month period. This would, crudely, translate to an annual deposition of 18 – 36 eq ha$^{-1}$ (assuming that 1 mole of S is equivalent to 2 acid equivalents, Bouwman et al., 2002), which is well below the lowest critical load of acidity (based on 5[th] percentile of maximum critical load of S) estimated by Forsius et al. (2010) for the area: 200 eq ha$^{-1}$ a$^{-1}$ (using an aluminum-to-base cations ratio criteria) or 100 eq ha$^{-1}$

a$^{-1}$ (using an acid neutralizing capacity criteria, a more stringent measure). Note that caution needs to be taken in interpreting the corresponding deposition values for the 2030 scenarios as there was no projection done for the anthropogenic emissions other than for the marine shipping emissions over Canadian waters for these model runs. The





shipping contributions to the total deposition of S to the three land cover types are small (below 1%), while the contributions to dry deposition (which is more heavily tied to ambient concentrations) are noticeably greater. As shown in Table 9, the total deposition of S (and N) is dominated by wet deposition in this region. The land-cover-weighted N deposition ranges between 36 g ha$^{-1}$ (over barren land) to 84 g ha$^{-1}$ (over lakes) over the JAS period at the present level. This again translates to (approximately) 0.144 – 0.336 kg of N ha$^{-1}$ a$^{-1}$, or 10 – 24 eq ha$^{-1}$ a$^{-1}$, which is also below the critical load for acidification currently estimated for the region in Forsius et al. (2010) and the empirical critical loads for nutrient N of 1 – 3 kg of N ha$^{-1}$ a$^{-1}$ for North America ecoregion of tundra (Linder et al., 2013; Pardo et al., 2011).

The contributions from Arctic shipping to total N deposition for the three land-cover types are simulated as small at the current level, but are predicted to increase significantly under the 2030 scenarios. It should be noted that, although the current deposition of S and N over the Arctic region are low and generally below the existing critical load estimates, with the projected increase in global production of nitrogen expected to be needed to meet the growing demand for food and energy, atmospheric emissions and depositions of nitrogen are expected to increase (Galloway et al., 2004; Dentener et al., 2006); this situation combined with the expected increase in shipping activities in Arctic waters could raise the level of deposition to above the critical loads for the region. Furthermore, it is recognized that the current estimates of critical loads for North American Arctic ecosystems are highly uncertain due to a number of factors including limitations in methodology and lack of data (Forsius et al., 2010; Pardo et al., 2011; Linder et al., 2013). Given these considerations, a careful assessment of potential ecosystem impacts from Arctic shipping emissions, particularly in the future context, is warranted.

## 5.3 On black carbon

Black carbon (BC), formally defined as an ideally light-absorbing substance composed of carbon (Petzold et al., 2013), is a short-lived climate forcer (SLCF): it absorbs solar radiation, influences cloud processes, and alters the melting of snow and ice and, hence, surface albedo (Bond et al., 2013; Flanner et al., 2007). BC is emitted into the atmosphere from a variety of combustion processes including shipping activities. Although shipping contributes only up to about 2% of global BC emissions, it may constitute a larger fraction of direct BC emissions in remote regions such as the Arctic, an area with higher sensitivity to carbonaceous emissions due to snow albedo effects (Bond et al., 2013). In our model, BC is represented by the "elemental carbon" (EC) component of the internally-mixed aerosols. By its sources and chemical/physical properties represented in the model, the modelled EC is equivalent to BC. In the context of the important radiative effect of BC, the impact of Arctic shipping emissions on both column loading and deposition of BC (or modelled EC) will be assessed here.

Figure 18 shows the modelled EC (or modelled BC, hereafter) column loadings averaged over the JAS period (2010 base case) and the percentage contributions from Arctic shipping for the 2010 base case, 2030 BAU and 2030 ECA scenarios. The contribution statistics by geographical sectors are included in Table 8 (last column). The modelled averaged BC column loading over the Canadian Arctic (north of 60 N) ranges between 20 and 200 μg m$^{-2}$ (Figure 18(a)), higher over the western Canadian Arctic than the east, where the region is strongly impacted by northern boreal-forest fires over western Canada and Alaska during the summer months. A similar range of modelled BC loading over the Arctic is also reported by Eckhardt et



al. (2015) in a recent multi-model assessment for simulating BC and sulfate in the Arctic atmosphere. The contribution to BC loading from Canadian Arctic shipping emissions at the 2010 baseline level is limited and localized, generally below 0.1% on average and up to 2% over localized areas in the eastern Canadian Arctic (Figure 18(b)). In absolute terms, the shipping contribution to BC loading is below 0.1 µg m$^{-2}$ over most parts of the Canadian Arctic. This is somewhat smaller than the estimate of Ødemark et al. (2012), where the Arctic shipping contribution to the tropospheric BC column is estimated at 0.38 µg m$^{-2}$ averaged over 60º N – 90º N. Noting that the present assessment focuses on the impact of shipping over the Canadian Arctic waters only as opposed to shipping over the entire Arctic waters (as in the case of Ødemark et al., 2012), the smaller contribution from this assessment is expected, as shipping activities within Canadian Arctic waters constitute only a small portion of overall Arctic shipping activities, e.g., compared to the activities over the Barents Sea, Norwegian Sea, and along southwest coast of Greenland (Arctic Council, 2009; Winther et al., 2014). There is a considerable increase in the contribution to BC loadings from Canadian Arctic shipping emissions in the 2030 BAU scenario, as seen in Figure 18(c) and Table 8, particularly over Baffin Bay, of up to 15% locally, in response to projected increases in ship traffic there. Under the 2030 ECA scenario, the modelled shipping contribution to BC loading is slightly reduced from the 2030 BAU level but it is still significantly greater than that at the current 2010 level (Figure 18(d) and Table 8).

The model simulated total (dry + wet) deposition of BC accumulated for the JAS period at the current (2010) level is shown in Figure 19 along with the percentage contribution from shipping over Canadian Arctic waters under all three scenarios. The contribution statistics by geographical sectors are included in Table 8 (2$^{nd}$ last column). Modelled BC deposition over the Canadian Arctic ranges from up to 50 mg m$^{-2}$ in the southwest to around 0.5 mg m$^{-2}$ in the northeast over the three month period. The modelled area-averaged BC deposition flux for 60º N – 90º N between 50º W and 140º W is 2.3 mg m$^{-2}$ over the 3 month period, or 9.2 mg m$^{-2}$ yr$^{-1}$, which is within the range of modelled BC deposition fluxes averaged over the Arctic (60º – 90º N) from a multi-model assessment of Jiao et al. (2014; see their Figure 9). The contribution from Canadian Arctic shipping at current levels is mostly between 0.1% and 0.5% over the shipping channels, and locally up to 5% (Figure 19(b)). Similar to the case of BC column loading discussed above, there is an important increase in the shipping contribution to BC deposition in the 2030 BAU scenario over the east coast of Baffin Island (Figure 19(c)). The shipping contribution to BC deposition averaged over the northeast sector E6 increases to 1.5%, exceeding 30% locally, under the 2030 BAU scenario (Table 8).

Since BC deposition to ice and snow is of most interest when considering the potential albedo effect, averaged BC deposition fluxes to ice and snow, defined as

$$\sum_{i=1}^{n} F_{ice/snow}(i) \times A(i) \times depo(i) \Big/ \sum_{i=1}^{n} F_{ice/snow}(i) \times A(i)$$

(where $\boldsymbol{F_{ice/snow}}$ is the grid fraction of ice/snow cover), have been computed and the respective contributions from shipping within Canadian Arctic waters are examined here. Table 10 shows average monthly BC deposition fluxes to ice/snow (total,





as well as dry and wet, separately) over the Canadian Arctic region (60º N – 90º N, 50º W – 140º W) for the three peak shipping months, July – September, and the corresponding shipping contributions. Modelled monthly mean ice and snow cover fields (shown in Figure 20) are used for this calculation. As shown, the Arctic ice/snow cover recedes progressively through the summer months. The monthly BC deposition to ice and snow is highest in August due to higher precipitation and

wet deposition. There is a sharp reduction in September as a result of the combination of a reduction in column BC loading (see supplementary material, Figure S.8) due to the reduced wildfire events in western Canada in late summer and receding ice/snow cover further to the north (Figure 20). Again total deposition is largely dominated by the wet component. In general, shipping over Canadian Arctic waters makes only a small contribution to the total BC deposition on Arctic ice/snow; the relative contribution is larger in September due to the reduced impact from wildfire emissions. Proportionally,

Arctic shipping makes a greater contribution through dry deposition than through wet deposition over northern regions as the emissions are more likely to be trapped within the stable marine boundary layer and hence have a greater impact on the near-surface atmospheric concentration. Table 10 also includes the shipping contribution to BC deposition to ice/snow in absolute terms. It shows that the shipping contributions are roughly double in the 2030 BAU scenario from present levels. It is interesting to see that dry deposition is playing a bigger role in this increase, particularly for the month of July, reflecting a

significant increase in near-surface atmospheric concentration of BC in this scenario.

It is seen from this assessment that current shipping emissions over Canadian Arctic waters make relatively small contributions to both BC loading and deposition in the Arctic. However the contributions are expected to increase in the 2030 scenarios. Assessing the radiative effect from BC loading and deposition on snow attributable to the shipping emissions over the Canadian Arctic waters is beyond the scope of this study. There are existing efforts to assess radiative forcing from

specific forcing agents and/or emission sectors mostly using global models with relatively coarse resolutions. For example, a global BC radiative forcing of ~ 2 mW m$^{-2}$ attributable to current international shipping (without the consideration for BC snow albedo effect) was estimated by Eyring et al. (2010); Ødemark et al. (2012) estimated annual mean BC relative forcing attributable to Arctic shipping activities at the present (2004) level to be 0.60 mW m$^{-2}$ (due to BC in air) and 0.47 mW m$^{-2}$ (due to BC in snow) averaged over 60º N – 90º N. The current understanding is that overall net forcing from the present-day

ship emissions of SLCF pollutants is negative due to higher emission of sulfur (Fuglestvedt et al., 2008; Eyring et al., 2010; Ødemark et al., 2012). As seen from this assessment, shipping induced changes in atmospheric composition and deposition are occurring at regional to local scales (particularly in the Arctic). Climate feedbacks are therefore likely to act at these scales and hence climate forcing impact assessments will require modelling undertaken at much finer resolutions than those used in the existing relative forcing and climate impact assessments.

**6 Summary and conclusions**

In this study, an on-line air quality forecast model (GEM-MACH) was used for a first regional assessment of the impact of Arctic shipping emissions on air pollution in the Canadian Arctic and northern regions. First, the model's ability to simulate





ambient atmospheric compositions in the region of interest was evaluated with available observations. The impacts of Arctic shipping emissions at both present and projected future levels were then assessed based on model sensitivity runs using a detailed marine emission inventory for ships sailing in Canadian waters developed specially for this study.

The adapted GEM-MACH for Arctic is shown to have similar skill in predicting ambient $O_3$ and $PM_{2.5}$ in the Canadian northern and Arctic region as the current operational air quality forecast models in North America and Europe. The model is able to simulate well the observed ambient $O_3$, and some of the PM components at the Canadian high Arctic site, Alert. The model has reasonable skill in predicting $NO_2$ and $SO_2$ in the north at a regional scale; at local scales the model prediction depends heavily on emission inputs. The evaluation results indicate large uncertainties in the representation of local emissions in the remote north and the need for improved emission estimates and representation for the oil and gas facilities in northeastern British Columbia and northern Alberta. There is a significant data gap in northern Canada, particularly the eastern Arctic, for air quality monitoring and model evaluation.

Key findings from the model assessment of the impact of Arctic shipping emissions are the following:

- At the current (2010) level, Arctic shipping emissions contribute to less than 1% of ambient $O_3$ concentration over the eastern Arctic. This contribution is expected to increase to up to 5% in the 2030 business-as-usual (BAU) scenario with broader region of impact.

- In comparison, the impact of Arctic shipping emission on ambient $PM_{2.5}$ concentration is more confined to areas close to the shipping channels. Current (2010) levels of Arctic shipping contribute to $1 - 5\%$ of ambient $PM_{2.5}$ concentration over the shipping channels and $< 0.5\%$ over land. At the 2030 BAU level, the shipping contribution is expected to increase to $5 - 20\%$ over the shipping channels.

- For $NO_2$ and $SO_2$, both primary pollutants, Arctic shipping emissions make significant contributions to ambient concentrations over the eastern Arctic: $10 - 50\%$ for $NO_2$ and $20 - 100\%$ for $SO_2$, over shipping channels and coastal regions with close proximity to shipping routes, at current (2010) level. The shipping contribution to $NO_2$ concentrations is expected to increase to $> 50\%$ under 2030 BAU, while the increase in shipping contribution to $SO_2$ concentrations is more moderate due to the anticipated global cap on sulfur in ship fuel that is due to come into effect.

- Contrasting to the 2030 BAU, the 2030 ECA scenario, i.e., assuming the Canadian Arctic will be designated as an Emission Control Area (as is the case for the east and west coasts of North America), will see a significant reduction in Arctic shipping contribution to ambient concentrations of $SO_2$ and $PM_{2.5}$. Particularly, the Arctic shipping contributions to population-weighted concentration of $SO_2$ and $PM_{2.5}$ will be brought down to below the current level.

- Despite the significant contributions to the ambient concentrations of $SO_2$ and $NO_2$, Arctic shipping contribution to the deposition of total S and N to the Arctic ecosystem is small, $< 5\%$, at present (2010) level, due to the dominance of wet deposition. However, the contribution is expected to increase to up to 20% for S and 50% for N, under the 2030 BAU scenario.



- Based on existing estimates of critical loads for northern terrestrial ecosystems, the current S and N deposition to the three dominant land-cover types (tundra, lakes, and barren/desert) in the Canadian Arctic and northern region is well below the lowest critical loads for acidification and eutrophication. However, given the large uncertainty in the current critical load estimates for the Arctic ecosystem, the anticipated increase in atmospheric emissions and deposition of nitrogen globally, and the expected increase in Arctic shipping contribution to the deposition of N to the north, more careful assessment of potential ecosystem impacts from Arctic shipping emissions, particularly in the future context, is needed.

- The contribution to BC loadings from Canadian Arctic shipping emissions at the 2010 baseline level is limited and localized, generally below 0.1% on average and up to 2% over localized areas in the eastern Canadian Arctic. There is a considerable increase in the contribution to the BC loading from the Canadian Arctic shipping emissions in the 2030 BAU scenario, particularly over Baffin Bay, with up to 15% locally, in response to the projected increase in ship traffic there.

- The contribution to BC deposition from shipping in the Canadian Arctic at current (2010) levels is mostly between 0.1% and 0.5% over the shipping channels, and locally up to 5%. Similar to the case of BC column loading, there is an important increase in the shipping contribution to BC deposition in the 2030 BAU scenario over the east coast of Baffin Island. The shipping contribution to BC deposition averaged over the eastern Canadian high Arctic increases to 1.5%, exceeding 30% locally.

- In general, shipping over the Canadian Arctic waters makes a small contribution towards the total BC deposition on Arctic ice/snow (taking into account of the sea-ice cover during the Arctic shipping season). Proportionally, Arctic shipping makes a greater contribution to dry deposition than to wet deposition over the northern regions as the emissions are more likely to be trapped within the stable marine boundary layer and hence have greater impact on the near-surface atmospheric concentration. The analysis shows that shipping contributions to BC deposition fluxes to ice/snow are roughly double in the 2030 BAU scenario from present levels in response to the projected increase in Arctic shipping activities.

- It is indicative from this study that shipping induced changes in atmospheric composition and deposition are at regional to local scales (particularly in the Arctic). Climate feedbacks are consequently likely to act at these scales thus climate impact assessments will require modelling undertaken at much finer resolutions than those used in the existing radiative forcing and climate impact assessments.



## Appendix A. Model Evaluation Statistical Measures

The following statistical measures are considered for the model evaluation in this study, letting M the vector of model output and O the vector of observation (N-record both) with mean value $\bar{M}$ and $\bar{O}$, respectively:

Mean bias (MB)

$$MB = \frac{\sum_{i=1}^{N}(M_i - O_i)}{N}$$

5    Normalised mean bias (NMB)

$$NMB(\%) = 100 \times \frac{\sum_{i=1}^{N}(M_i - O_i)}{\sum_{i=1}^{N} O_i}$$

Root mean square error (RMSE)

$$RMSE = \sqrt{\frac{\sum_{i=1}^{N}(M_i - O_i)^2}{N}}$$

Normalised mean square error (NMSE)

$$NMSE(\%) = 100 \times \frac{N \sum_{i=1}^{N}(M_i - O_i)^2}{\sum_{i=1}^{N} M_i \sum_{i=1}^{N} O_i}$$

Pearson correlation coefficient (R)

$$R = \frac{\sum_{i=1}^{N} M_i O_i - N\bar{M}\bar{O}}{\sqrt{\sum_{i=1}^{N} M_i^2 - N\bar{M}} \sqrt{\sum_{i=1}^{N} O_i^2 - N\bar{O}}}$$





**Acknowledgement**

We would like to acknowledge ECCC's National Atmospheric Chemistry Database (NAtChem) and Analysis Facility for access to the North America air monitoring data used for model evaluation in this study. We are thankful to the agencies in Canada and U.S. for maintaining the networks, in particular the National Air Pollution Surveillance (NAPS) network in

5   Canada and the US EPA's Aerometric Information Retrieval Systems (AIRS) and Air Quality System (AQS) database. The World Data Centre for Greenhouse Gases (WDCGG) and NOAA/ESRL (Drs. Audra McClure-Begley and Irina Petropavlovskikh) are gratefully acknowledged for providing the $O_3$ monitoring data at Barrow, Alaska. We would also like to acknowledge the MACC-II project and Xiaobo Yang (ECMWF) for processing and providing the 2010 MACC-MOZART reanalysis data. Ms. Monica Hilborn of Environmental Protection Branch (EPB) at ECCC diligently reviewed and verified

10   the Canadian Marine inventory numbers, and Mr. Hui Peng of EPB/ECCC reviewed the manuscript. We are also grateful to the GEM-MACH development team at ECCC for technical support.



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



## Tables

**Table 1.** Number of trips and estimated emissions (in tonnes) of CAC pollutants from marine shipping activities over the Canadian Arctic waters for the year 2010 base case and for the projected 2030 year scenarios (BAU and ECA).

| Vessel Category/Class | # of trips | | NO$_x$ | | | SO$_x$ | | | CO | | VOC | | PM | | | NH$_3$ | |
|---|---|---|---|---|---|---|---|---|---|---|---|---|---|---|---|---|---|
| | 2010 | 2030 | 2010 | 2030 BAU | 2030 ECA | 2010 | 2030 BAU | 2030 ECA | 2010 | 2030[&] | 2010 | 2030[&] | 2010 | 2030 BAU | 2030 ECA | 2010 | 2030[&] |
| **Merchant Passenger** | | | 308 | 1,049 | 762 | 127 | 186 | 38 | 26 | 113 | 10 | 45 | 18 | 40 | 25 | 0 | 2 |
| Merchant Passenger | 63 | 271 | 308 | 1,049 | 762 | 127 | 186 | 38 | 26 | 113 | 10 | 45 | 18 | 40 | 25 | 0 | 2 |
| **Merchant Commercial** | | | 1,821 | 8,611 | 4,865 | 1,079 | 1,427 | 288 | 163 | 842 | 64 | 342 | 144 | 314 | 193 | 2 | 12 |
| Merchant Bulk | 39 | 191 | 431 | 4,926 | 2,266 | 206 | 798 | 160 | 38 | 488 | 15 | 198 | 28 | 176 | 108 | 0 | 7 |
| Merchant Other | 245 | 453 | 815 | 1,810 | 1,381 | 568 | 320 | 64 | 72 | 171 | 28 | 69 | 74 | 69 | 42 | 1 | 3 |
| Tanker | 169 | 247 | 575 | 1,875 | 1,218 | 305 | 310 | 63 | 53 | 183 | 21 | 75 | 42 | 69 | 43 | 1 | 3 |
| **Other** | | | 1,157 | 1,602 | 1,602 | 17 | 15 | 15 | 91 | 127 | 42 | 59 | 23 | 31 | 31 | 2 | 2 |
| Coast Guard | 20 | 25 | 613 | 844 | 844 | 10 | 13 | 13 | 51 | 70 | 22 | 31 | 13 | 17 | 17 | 1 | 1 |
| Tug Boat | 300 | 367 | 506 | 720 | 720 | 7 | 1 | 1 | 37 | 55 | 18 | 26 | 9 | 13 | 13 | 1 | 1 |
| Special Purpose | 7 | 6 | 38 | 38 | 38 | 1 | 1 | 1 | 3 | 3 | 1 | 1 | 1 | 1 | 1 | 0 | 0 |
| **Fishing** | | | 231 | 270 | 270 | 4 | 0 | 0 | 19 | 22 | 8 | 10 | 5 | 5 | 5 | 0 | 0 |
| Fishing | 134 | 156 | 231 | 270 | 270 | 4 | 0 | 0 | 19 | 22 | 8 | 10 | 5 | 5 | 5 | 0 | 0 |
| **Total** | 978[*] | 1,716 | 3,518 | 11,531 | 7,499 | 1,228 | 1,628 | 340 | 299 | 1,104 | 125 | 455 | 190 | 390 | 253 | 4 | 16 |

[*] Including one (1) trip for merchant container.
5   [&] No difference between BAU and ECA scenarios for these pollutants.





**Table 2.** 2010 marine emission estimates (in tonnes) for Canada's waters by activity mode and by regions.

| Air contaminant | Underway | | | Berthing | | | Anchoring | | | Total | | |
|---|---|---|---|---|---|---|---|---|---|---|---|---|
| | East | West | Arctic | East | West | Arctic | East | West | Arctic | East | West | Arctic |
| NO$_x$ | 112,301 | 70,980 | 3,257 | 6,722 | 2,966 | 252 | 1,571 | 1,188 | 9 | 120,594 | 75,134 | 3,518 |
| SO$_x$ | 55,978 | 38,600 | 1,068 | 5,393 | 2,638 | 152 | 1,448 | 1,390 | 8 | 62,819 | 42,628 | 1,228 |
| CO | 10,265 | 6,128 | 270 | 870 | 347 | 28 | 200 | 148 | 1 | 11,336 | 6,623 | 299 |
| VOC | 7,948 | 2,883 | 117 | 4,913 | 383 | 8 | 50 | 39 | 0 | 12,911 | 3,304 | 125 |
| PM | 7,998 | 5,403 | 171 | 617 | 303 | 18 | 161 | 151 | 1 | 8,775 | 5,858 | 190 |
| NH$_3$ | 140 | 92 | 4 | 2 | 1 | 0 | 0 | 0 | 0 | 142 | 94 | 4 |

5    **Table 3.** Stack parameters for different ship emissions inventories used in this study.

| Average values | Heavy Diesel | Diesel | Gasoline |
|---|---|---|---|
| Stack Height (m) | 41.82 | 40.23 | 24.52 |
| Stack Diameter (m) | 1 | 1 | 1 |
| Stack Velocity (m/s) | 20 | 20 | 20 |
| Stack Gas Exit Temperature (C) | 275 | 275 | 275 |




**Table 4.** Regional evaluation statistics for $O_3$, $PM_{2.5}$, $NO_2$, and $SO_2$.

| Geographical sector | # of sites | MB (ppbv, µg m$^{-3}$) | | | NMB (%) | | | RMSE (ppbv, µg m$^{-3}$) | | | NMSE (%) | | | r | | |
|---|---|---|---|---|---|---|---|---|---|---|---|---|---|---|---|---|
| | | Hly | Dly | Snl | Hly | Dly | Snl | Hly | Dly | Snl | Hly | Dly | Snl | Hly | Dly | Snl |
| $O_3$ (ppbv) | | | | | | | | | | | | | | | | |
| Northern | 15 | 3.0 (3.1)[*] | 3.5 | 3.2 | 15.5 (15.8) | 18.3 | 16.5 | 8.4 (4.4) | 6.5 | 4.3 | 14.5 (4.2) | 9.7 | 4.2 | 0.57 (0.73) | 0.56 | 0.66 |
| Southeastern | 69 | 5.7 (5.8) | 5.7 | 5.6 | 24.9 (25.1) | 24.6 | 24.1 | 12.1 (7.5) | 9.8 | 7.3 | 19.0 (7.8) | 13.2 | 8.0 | 0.66 (0.86) | 0.67 | 0.25 |
| Southwestern | 54 | 4.2 (4.2) | 4.3 | 4.1 | 20.6 (20.8) | 21.5 | 20.9 | 12.6 (5.8) | 9.9 | 7.5 | 27.8 (6.3) | 19.1 | 11.5 | 0.54 (0.87) | 0.43 | 0.30 |
| $PM_{2.5}$ (µg m$^{-3}$) | | | | | | | | | | | | | | | | |
| Northern | 9 | -0.7 (-0.6) | -0.7 | -0.8 | -14.4 (-12.0) | -14.3 | -16.4 | 6.5 (2.6) | 4.6 | 3.1 | 201 (34.6) | 98.7 | 53.2 | 0.08 (0.09) | 0.13 | -0.27 |
| Southeastern | 36 | -0.2 (-0.2) | -0.1 | -0.3 | -2.1 (-1.8) | -1.8 | -3.0 | 8.9 (5.5) | 7.0 | 3.0 | 69.7 (29.0) | 45.0 | 12.8 | 0.58 (0.79) | 0.70 | 0.28 |
| Southwestern | 9 | -3.3 (-3.2) | -3.3 | -3.6 | -34.3 (-34.1) | -34.0 | -37.0 | 19.4 (10.3) | 12.2 | 4.6 | 257 (102.9) | 104 | 35.7 | 0.37 (0.63) | 0.62 | -0.02 |
| $NO_2$ (ppbv) | | | | | | | | | | | | | | | | |
| Northern | 10 | 0.3 (0.4) | 0.3 | 0.3 | 8.3 (12.9) | 8.9 | 8.9 | 5.6 (2.0) | 3.9 | 2.8 | 104 (28.5) | 53.3 | 30.3 | 0.56 (0.52) | 0.76 | 0.86 |
| Southeastern | 30 | 2.6 (2.6) | 2.6 | 2.7 | 45.7 (47.0) | 46.6 | 47.6 | 10.3 (5.4) | 7.9 | 5.1 | 139 (54.9) | 90.1 | 42.3 | 0.45 (0.58) | 0.58 | 0.72 |
| Southwestern | 55 | 1.4 (1.7) | 1.4 | 1.4 | 20.2 (26.4) | 19.9 | 20.5 | 9.0 (2.7) | 7.0 | 6.0 | 90.4 (11.9) | 62.1 | 50.4 | 0.55 (0.82) | 0.66 | 0.70 |
| $SO_2$ (µg m$^{-3}$) | | | | | | | | | | | | | | | | |
| Northern | 18 | 9.9 (10.1) | 9.9 | 9.9 | 325. (334.) | 324 | 325 | 30.6 (14.0) | 21.7 | 16.1 | 1360 (537.0) | 609 | 357 | 0.09 (-0.08) | 0.30 | 0.90 |
| Southeastern | 17 | 6.2 (6.2) | 6.2 | 6.5 | 183 (187) | 183 | 197 | 17.7 (8.7) | 12.0 | 8.8 | 663 (235.0) | 323 | 194 | 0.10 (0.04) | 0.19 | 0.39 |
| Southwestern | 50 | -0.5 (-0.5) | -0.5 | -0.5 | -11.5 (-11.3) | -11.8 | -11.4 | 17.1 (3.0) | 10.1 | 5.8 | 1190 (49.9) | 489 | 181 | 0.10 (0.35) | 0.13 | 0.15 |

[*] Numbers in brackets are scores calculated based on modelled and observed hourly time series averaged over all sites within a given region as in Im et al (2015a,b).



**Table 5.** Division of geographical sectors over the Canadian Arctic and northern regions for the assessment

| Sector # | Latitude range | Longitude range |
|---|---|---|
| E1 | 50 – 60 N | 50 – 75 W |
| E2 | 50 – 60 N | 75 – 100 W |
| E3 | 60 – 70 N | 50 – 75 W |
| E4 | 60 – 70 N | 75 – 100 W |
| E5 | 70 – 80 N | 50 – 75 W |
| E6 | 70 – 80 N | 75 – 100 W |
| W1 | 50 – 60 N | 100 – 140 W |
| W2 | 60 – 70 N | 100 – 140 W |
| W3 | 70 – 80 N | 100 – 140 W |

**Table 6.** Percentage contribution from Arctic shipping to ambient concentrations of criteria pollutants, by geographical sectors (see Table 1), for July-August-September period.

| | sector # | $PM_{2.5}$ (%) | | | $O_3$ (%) | | | $NO_2$ (%) | | | $SO_2$ (%) | | |
|---|---|---|---|---|---|---|---|---|---|---|---|---|---|
| | | mean | med. | max. | mean | med. | max. | mean | med. | max. | mean | med. | max. |
| **2010** | E1 | 0.08 | 0.04 | 1.12 | 0.20 | 0.18 | 0.67 | 1.33 | 0.43 | 47.0 | 1.40 | 0.24 | 45.5 |
| | E2 | 0.04 | 0.04 | 0.66 | 0.21 | 0.18 | 0.51 | 2.53 | 0.66 | 56.2 | 1.51 | 0.14 | 43.6 |
| | E3 | 0.58 | 0.33 | 3.82 | 0.39 | 0.34 | 1.09 | 10.80 | 3.42 | 65.3 | 19.90 | 10.50 | 86.1 |
| | E4 | 0.22 | 0.19 | 2.98 | 0.33 | 0.29 | 0.86 | 7.98 | 5.08 | 63.9 | 19.30 | 15.70 | 94.0 |
| | E5 | 0.32 | 0.09 | 3.27 | 0.19 | 0.18 | 0.41 | 5.13 | 1.27 | 45.9 | 14.30 | 0.96 | 91.2 |
| | E6 | 0.53 | 0.35 | 3.48 | 0.21 | 0.21 | 0.54 | 14.60 | 8.86 | 78.5 | 47.80 | 44.40 | 97.8 |
| | W1 | 0.01 | 0.00 | 0.15 | 0.03 | 0.02 | 0.18 | 0.03 | 0.01 | 0.65 | 0.00 | 0.00 | 0.4 |
| | W2 | 0.05 | 0.03 | 0.74 | 0.14 | 0.14 | 0.59 | 2.16 | 0.49 | 85.9 | 2.11 | 0.14 | 62.5 |
| | W3 | 0.09 | 0.07 | 0.49 | 0.13 | 0.11 | 0.54 | 8.33 | 4.44 | 62.7 | 13.10 | 9.53 | 76.0 |
| **2030BAU** | E1 | 0.13 | 0.06 | 1.58 | 0.65 | 0.56 | 2.11 | 4.73 | 1.28 | 71.2 | 2.97 | 0.37 | 63.6 |
| | E2 | 0.05 | 0.04 | 0.40 | 0.60 | 0.54 | 1.12 | 4.62 | 1.80 | 50.3 | 1.13 | 0.17 | 16.9 |
| | E3 | 2.01 | 0.62 | 34.50 | 1.53 | 1.33 | 4.98 | 21.90 | 10.10 | 93.5 | 23.10 | 10.00 | 98.6 |
| | E4 | 0.39 | 0.27 | 7.16 | 0.90 | 0.90 | 1.72 | 13.60 | 10.80 | 90.5 | 21.60 | 15.80 | 96.1 |
| | E5 | 1.39 | 0.17 | 38.10 | 1.01 | 0.63 | 4.75 | 14.30 | 4.01 | 97.6 | 16.80 | 2.48 | 99.4 |
| | E6 | 1.58 | 0.66 | 28.00 | 0.96 | 0.72 | 4.92 | 33.00 | 26.20 | 97.1 | 57.30 | 61.30 | 99.6 |
| | W1 | 0.01 | 0.00 | 0.31 | 0.10 | 0.06 | 0.48 | 0.10 | 0.03 | 3.5 | -0.00 | -0.00 | 0.24 |
| | W2 | 0.08 | 0.05 | 0.88 | 0.44 | 0.43 | 1.25 | 6.29 | 1.52 | 83.5 | 3.82 | 0.24 | 64.3 |
| | W3 | 0.20 | 0.17 | 2.11 | 0.49 | 0.44 | 1.06 | 18.20 | 11.10 | 75.6 | 28.80 | 24.50 | 96.2 |
| **2030 ECA** | E1 | 0.05 | 0.03 | 0.65 | 0.47 | 0.40 | 1.52 | 3.52 | 0.85 | 64.9 | 0.62 | 0.06 | 22.6 |
| | E2 | 0.02 | 0.02 | 0.38 | 0.43 | 0.39 | 0.81 | 3.52 | 1.40 | 50.2 | 0.21 | 0.02 | 3.76 |
| | E3 | 0.70 | 0.20 | 16.80 | 1.05 | 0.91 | 3.18 | 17.10 | 6.80 | 89.4 | 9.31 | 2.13 | 92.8 |
| | E4 | 0.14 | 0.09 | 2.95 | 0.63 | 0.61 | 1.20 | 10.20 | 7.98 | 84.5 | 6.43 | 3.51 | 76.0 |
| | E5 | 0.49 | 0.05 | 20.30 | 0.67 | 0.42 | 3.48 | 11.00 | 2.60 | 96.0 | 8.48 | 0.47 | 96.9 |
| | E6 | 0.60 | 0.24 | 15.60 | 0.61 | 0.47 | 3.61 | 25.10 | 18.00 | 95.4 | 30.70 | 23.40 | 98.1 |
| | W1 | 0.01 | 0.00 | 0.14 | 0.07 | 0.04 | 0.34 | 0.07 | 0.02 | 3.39 | -0.01 | -0.01 | 0.02 |
| | W2 | 0.03 | 0.02 | 0.40 | 0.30 | 0.29 | 0.90 | 4.59 | 1.10 | 83.00 | 0.96 | 0.02 | 25.80 |
| | W3 | 0.08 | 0.06 | 0.98 | 0.31 | 0.28 | 0.71 | 13.20 | 8.19 | 66.20 | 9.65 | 5.98 | 83.00 |



**Table 7**. Arctic shipping contributions to population-weighted concentrations of criteria pollutants over eastern Canadian Arctic (North of 60 N, 60 – 100 W), for the July-August-September period.

| Scenario | PM$_{2.5}$ (%) | | | O$_3$ (%) | | | NO$_2$ (%) | | | SO$_2$ (%) | | |
|---|---|---|---|---|---|---|---|---|---|---|---|---|
| | mean | med. | max. | mean | med. | max. | mean | med. | max. | mean | med. | max. |
| 2010 | 0.75 | 0.63 | 3.23 | 0.37 | 0.37 | 0.83 | 7.73 | 6.95 | 15.23 | 53.93 | 62.35 | 83.35 |
| 2030 BAU | 1.28 | 1.12 | 5.07 | 1.36 | 1.36 | 3.08 | 25.57 | 23.48 | 61.24 | 60.58 | 65.76 | 90.36 |
| 2030 ECA | 0.58 | 0.48 | 2.54 | 0.98 | 0.96 | 2.47 | 23.39 | 20.27 | 60.78 | 28.78 | 28.54 | 65.15 |

5  **Table 8.** Percentage contribution from Arctic shipping to surface depositions of sulfur, nitrogen, and elemental carbon (EC) and column loading of EC, by geographical sectors (see Table 1), for the July-August-September period.

| | sector # | Total S deposition (%) | | | Total N deposition (%) | | | Total BC deposition (%) | | | BC column (%) | | |
|---|---|---|---|---|---|---|---|---|---|---|---|---|---|
| | | mean | med. | max. | mean | med. | max. | mean | med. | max. | mean | med. | max. |
| 2010 | E1 | 0.09 | 0.04 | 1.45 | 0.18 | 0.11 | 2.09 | 0.05 | 0.02 | 3.06 | 0.02 | 0.01 | 0.67 |
| | E2 | 0.07 | 0.05 | 2.39 | 0.07 | 0.05 | 0.74 | 0.02 | 0.02 | 2.17 | 0.01 | 0.01 | 1.01 |
| | E3 | 0.53 | 0.41 | 5.75 | 0.75 | 0.64 | 4.84 | 0.21 | 0.15 | 8.98 | 0.06 | 0.05 | 2.10 |
| | E4 | 0.51 | 0.42 | 6.99 | 0.61 | 0.51 | 4.40 | 0.14 | 0.09 | 2.98 | 0.05 | 0.05 | 0.54 |
| | E5 | 0.41 | 0.20 | 5.16 | 0.61 | 0.42 | 3.67 | 0.11 | 0.04 | 4.21 | 0.01 | 0.00 | 0.36 |
| | E6 | 0.61 | 0.49 | 5.08 | 0.94 | 0.85 | 4.37 | 0.20 | 0.13 | 3.34 | 0.05 | 0.04 | 0.99 |
| | W1 | 0.00 | 0.00 | 1.98 | 0.01 | 0.01 | 1.86 | -0.00 | 0.00 | 7.31 | 0.00 | 0.00 | 0.09 |
| | W2 | 0.07 | 0.04 | 2.73 | 0.32 | 0.14 | 8.41 | 0.03 | 0.01 | 5.81 | 0.01 | 0.01 | 0.75 |
| | W3 | 0.11 | 0.07 | 2.83 | 0.52 | 0.40 | 5.11 | 0.07 | 0.03 | 2.89 | 0.01 | 0.00 | 0.19 |
| 2030BAU | E1 | 0.12 | 0.06 | 2.69 | 0.71 | 0.42 | 5.82 | 0.85 | 0.62 | 5.89 | 0.04 | 0.02 | 0.45 |
| | E2 | 0.07 | 0.05 | 1.72 | 0.23 | 0.17 | 1.65 | 0.60 | 0.26 | 9.01 | 0.02 | 0.01 | 0.55 |
| | E3 | 1.54 | 0.50 | 33.20 | 5.41 | 2.60 | 57.50 | 0.70 | 0.47 | 19.60 | 0.09 | 0.07 | 4.11 |
| | E4 | 0.51 | 0.45 | 8.30 | 2.22 | 1.87 | 20.00 | 0.54 | 0.29 | 4.81 | 0.09 | 0.08 | 0.71 |
| | E5 | 1.61 | 0.51 | 34.30 | 5.01 | 2.07 | 59.00 | 0.61 | 0.35 | 9.15 | 0.04 | -0.01 | 2.15 |
| | E6 | 1.61 | 0.94 | 40.30 | 5.32 | 3.79 | 60.90 | 1.46 | 1.10 | 32.80 | 0.11 | 0.04 | 15.90 |
| | W1 | 0.01 | 0.00 | 3.52 | 0.04 | 0.03 | 2.10 | 0.32 | 0.13 | 7.36 | 0.00 | 0.00 | 0.19 |
| | W2 | 0.12 | 0.06 | 1.82 | 0.98 | 0.42 | 10.00 | 0.29 | 0.22 | 7.10 | 0.02 | 0.01 | 0.95 |
| | W3 | 0.20 | 0.15 | 2.55 | 1.41 | 1.14 | 7.85 | 0.30 | 0.17 | 6.18 | 0.01 | 0.01 | 0.49 |
| 2030 ECA | E1 | 0.04 | 0.03 | 1.12 | 0.53 | 0.32 | 4.08 | 0.09 | 0.05 | 2.93 | 0.04 | 0.03 | 0.33 |
| | E2 | 0.02 | 0.02 | 1.05 | 0.17 | 0.13 | 1.33 | 0.03 | 0.03 | 2.24 | 0.02 | 0.01 | 0.35 |
| | E3 | 0.36 | 0.13 | 10.10 | 3.48 | 1.74 | 44.60 | 0.31 | 0.19 | 13.70 | 0.08 | 0.07 | 2.81 |
| | E4 | 0.11 | 0.10 | 1.15 | 1.54 | 1.33 | 11.30 | 0.17 | 0.10 | 2.96 | 0.07 | 0.06 | 0.67 |
| | E5 | 0.37 | 0.12 | 10.60 | 3.35 | 1.41 | 48.50 | 0.21 | 0.06 | 5.59 | 0.03 | 0.00 | 1.36 |
| | E6 | 0.36 | 0.20 | 13.30 | 3.38 | 2.29 | 58.30 | 0.32 | 0.16 | 21.80 | 0.09 | 0.04 | 10.20 |
| | W1 | 0.00 | 0.00 | 2.11 | 0.03 | 0.02 | 3.13 | 0.00 | 0.00 | 9.06 | 0.00 | 0.00 | 0.19 |
| | W2 | 0.03 | 0.03 | 2.46 | 0.67 | 0.30 | 8.38 | 0.05 | 0.02 | 6.31 | 0.02 | 0.01 | 0.95 |
| | W3 | 0.05 | 0.04 | 1.03 | 0.89 | 0.73 | 5.25 | 0.09 | 0.04 | 3.68 | 0.01 | 0.01 | 0.31 |



**Table 9.** Land-cover weighted deposition of S and N for eastern Canadian Arctic (60 – 100 W, 60 – 90 N) over the July-August-September period and corresponding contributions from Arctic shipping.

|  | Land-cover type | Sulfur | | | | | | Nitrogen | | | | | |
|---|---|---|---|---|---|---|---|---|---|---|---|---|---|
|  |  | LC-weighted deposition (kg of S ha$^{-1}$) | | | Shipping contribution (%) | | | LC-weighted deposition (kg of N ha$^{-1}$) | | | Shipping contribution (%) | | |
|  |  | total | dry | wet | total | dry | wet | total | dry | wet | total | dry | wet |
| 2010 | lakes | 0.143 | 0.011 | 0.132 | 0.32 | 1.37 | 0.23 | 0.084 | 0.010 | 0.073 | 0.41 | 1.67 | 0.23 |
| 2010 | tundra | 0.116 | 0.008 | 0.109 | 0.41 | 1.30 | 0.35 | 0.068 | 0.011 | 0.058 | 0.60 | 1.97 | 0.35 |
| 2010 | barren | 0.073 | 0.005 | 0.067 | 0.53 | 1.19 | 0.47 | 0.036 | 0.005 | 0.031 | 0.81 | 2.42 | 0.58 |
| 2030 BAU | lakes | 0.143 | 0.011 | 0.132 | 0.34 | 1.35 | 0.25 | 0.085 | 0.011 | 0.074 | 1.50 | 5.41 | 0.92 |
| 2030 BAU | tundra | 0.116 | 0.008 | 0.109 | 0.52 | 1.80 | 0.43 | 0.070 | 0.011 | 0.059 | 2.44 | 7.11 | 1.54 |
| 2030 BAU | barren | 0.073 | 0.005 | 0.067 | 1.20 | 2.78 | 1.08 | 0.037 | 0.005 | 0.032 | 4.86 | 11.56 | 3.82 |
| 2030 ECA | lakes | 0.142 | 0.011 | 0.132 | 0.08 | 0.26 | 0.06 | 0.085 | 0.011 | 0.074 | 1.06 | 3.90 | 0.65 |
| 2030 ECA | tundra | 0.116 | 0.007 | 0.108 | 0.12 | 0.39 | 0.10 | 0.069 | 0.011 | 0.058 | 1.71 | 5.11 | 1.07 |
| 2030 ECA | barren | 0.072 | 0.005 | 0.067 | 0.26 | 0.61 | 0.23 | 0.037 | 0.005 | 0.032 | 3.01 | 7.36 | 2.35 |

5   **Table 10.** Averaged BC deposition on ice and snow over Canadian Arctic (50 – 140 W, 60 – 90 N), and contributions from shipping over the Canadian Arctic waters.

|  | Month | BC deposition to ice/snow (mg m$^{-2}$ mon$^{-1}$) | | | Arctic Shipping contribution (%) | | | Arctic Shipping contribution (µg m$^{-2}$ mon$^{-1}$) | | |
|---|---|---|---|---|---|---|---|---|---|---|
|  |  | total | dry | wet | total | dry | wet | total | dry | wet |
| 2010 | 7 | 0.560 | 0.051 | 0.509 | 0.03 | 0.09 | 0.02 | 0.16 | 0.04 | 0.12 |
| 2010 | 8 | 0.615 | 0.025 | 0.591 | 0.04 | 0.34 | 0.03 | 0.27 | 0.08 | 0.18 |
| 2010 | 9 | 0.163 | 0.004 | 0.159 | 0.14 | 0.77 | 0.12 | 0.22 | 0.03 | 0.19 |
| 2030 BAU | 7 | 0.561 | 0.051 | 0.510 | 0.06 | 0.27 | 0.04 | 0.34 | 0.14 | 0.20 |
| 2030 BAU | 8 | 0.617 | 0.025 | 0.593 | 0.09 | 0.67 | 0.06 | 0.54 | 0.17 | 0.37 |
| 2030 BAU | 9 | 0.163 | 0.004 | 0.159 | 0.27 | 1.32 | 0.24 | 0.44 | 0.06 | 0.39 |




**Figures**

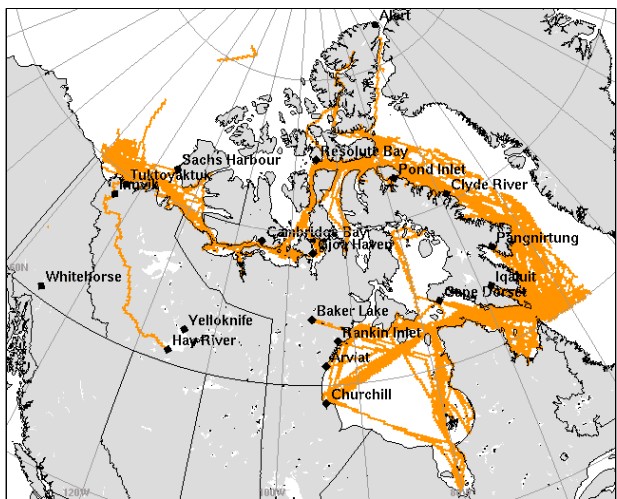

5    **Figure 1.** 2010 vessel movements in Canada's Arctic.

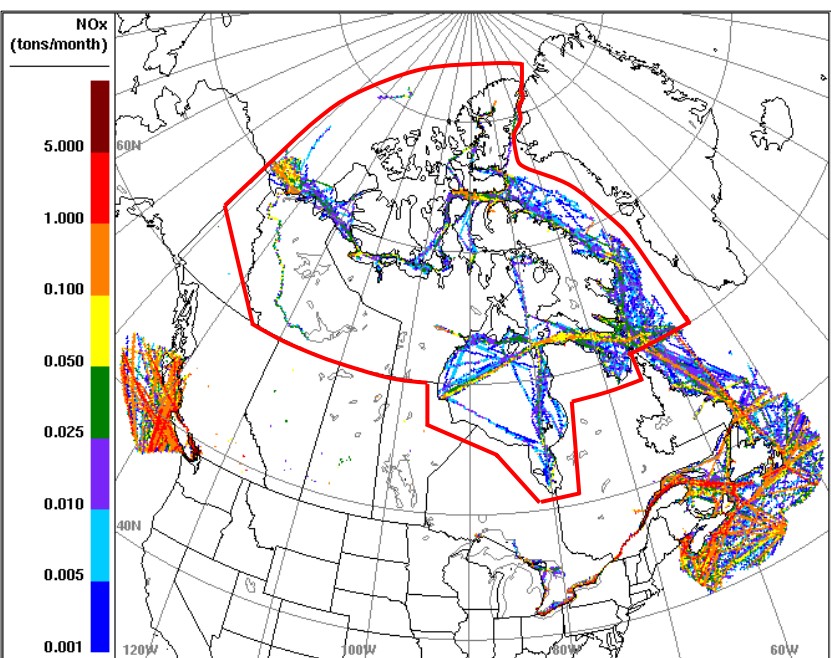

**Figure 2.** Processed model-ready $NO_x$ marine shipping emissions for August 2010 zoomed over Canadian waters, the red line outlining the Arctic region (including Hudson Bay) which is excluded from the current North American ECA 10   designation.





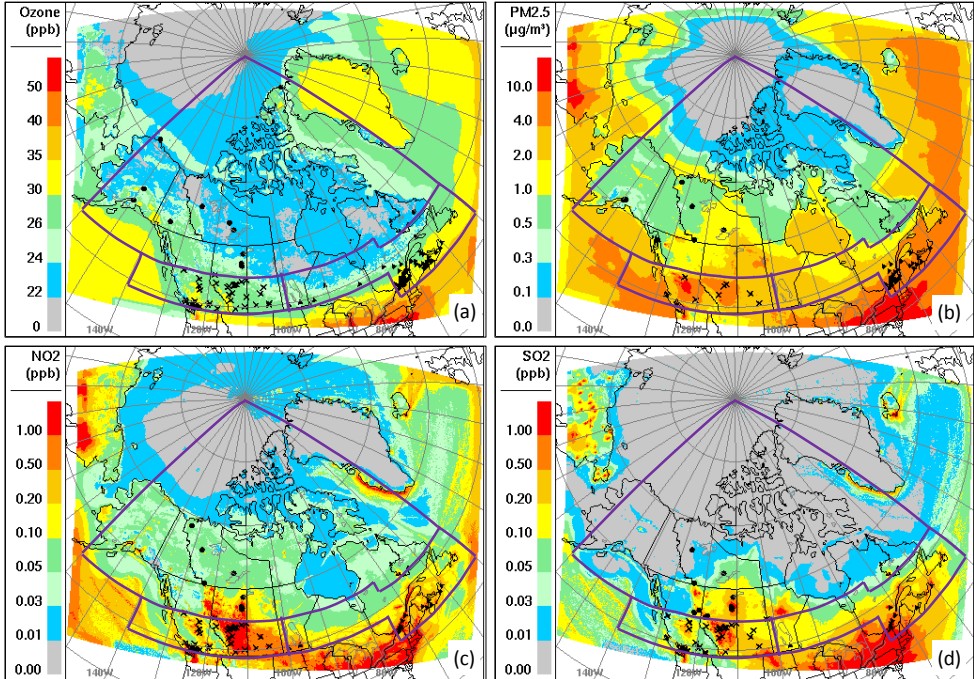

**Figure 3.** GEM-MACH Arctic modelling domain overlaid with monitoring sites: (a) $O_3$ monitoring sites shown on top of the modelled average ambient concentration over the July-September period; (b) same as (a) but for $PM_{2.5}$; (c) same as (a) but for $NO_2$; (d) same as (a) but for $SO_2$. (Subdivision of regions: crosses denoting the sites in "western region", filled triangles denoting the sites in "eastern region", and filled circles denoting sites in the "northern region").



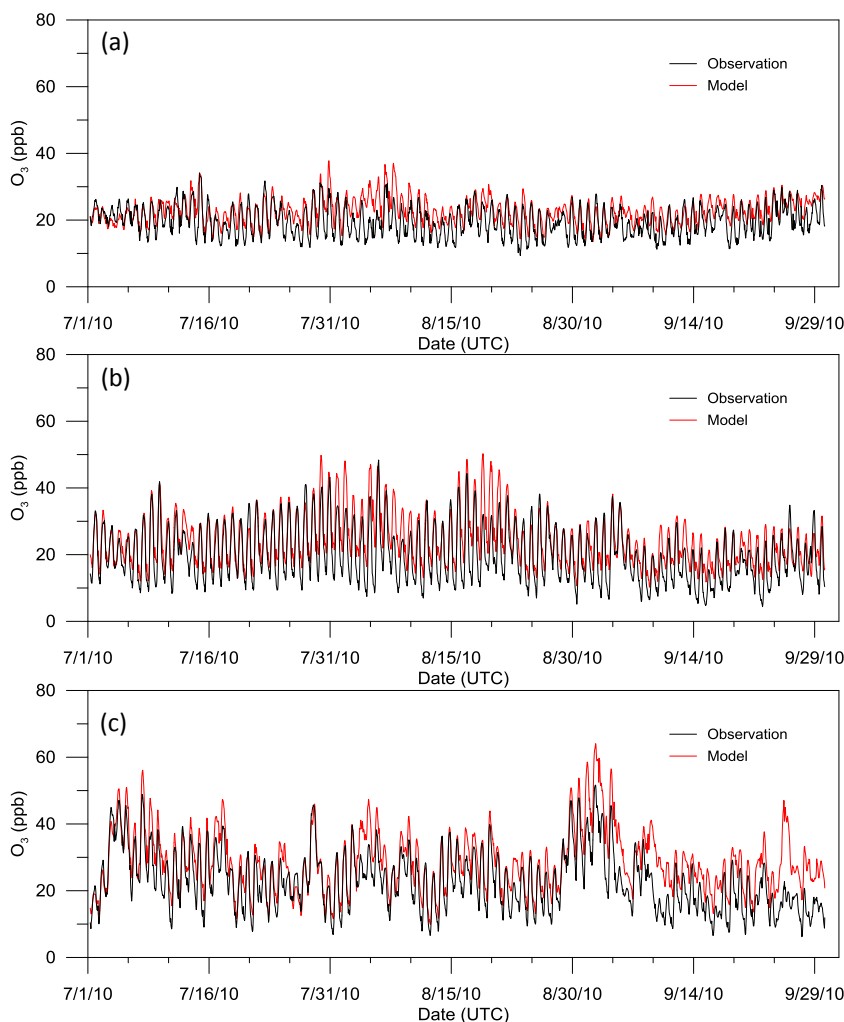

**Figure 4.** Regional-averaged $O_3$ time series, modelled and observed: (a) northern; (b) south-western; (c) south-eastern.





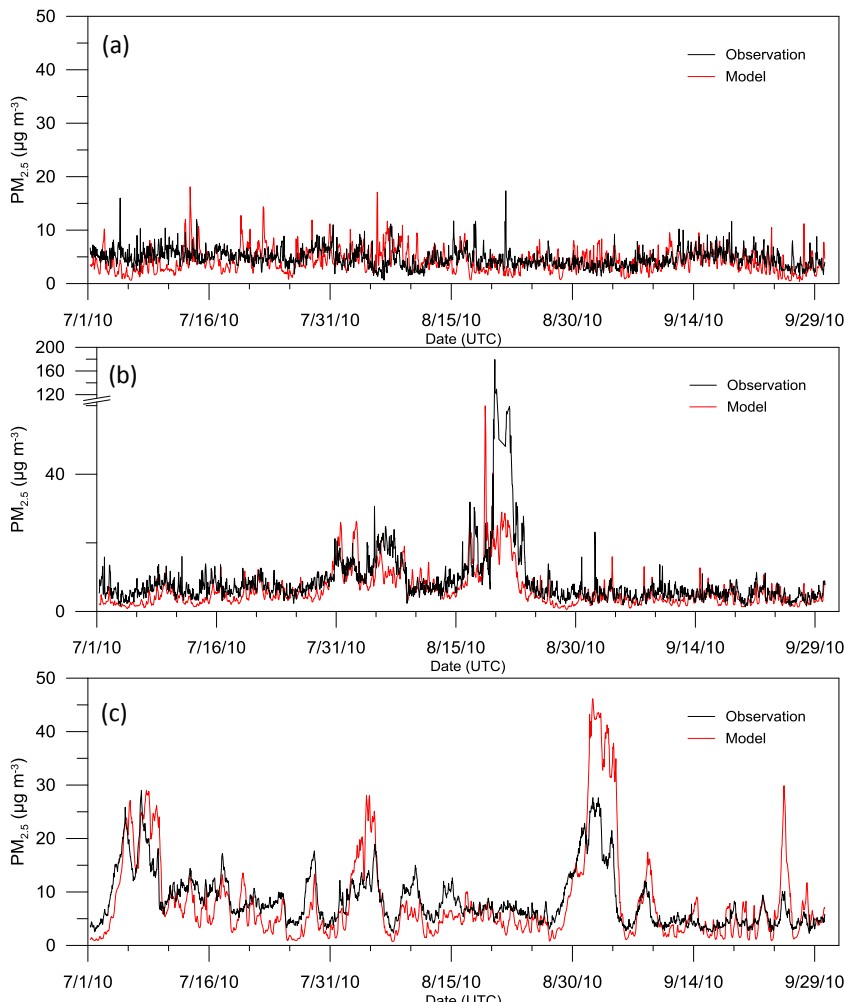

**Figure 5**. Regional-averaged PM$_{2.5}$ time series, modelled and observed: (a) northern; (b) south-western; (c) south-eastern.





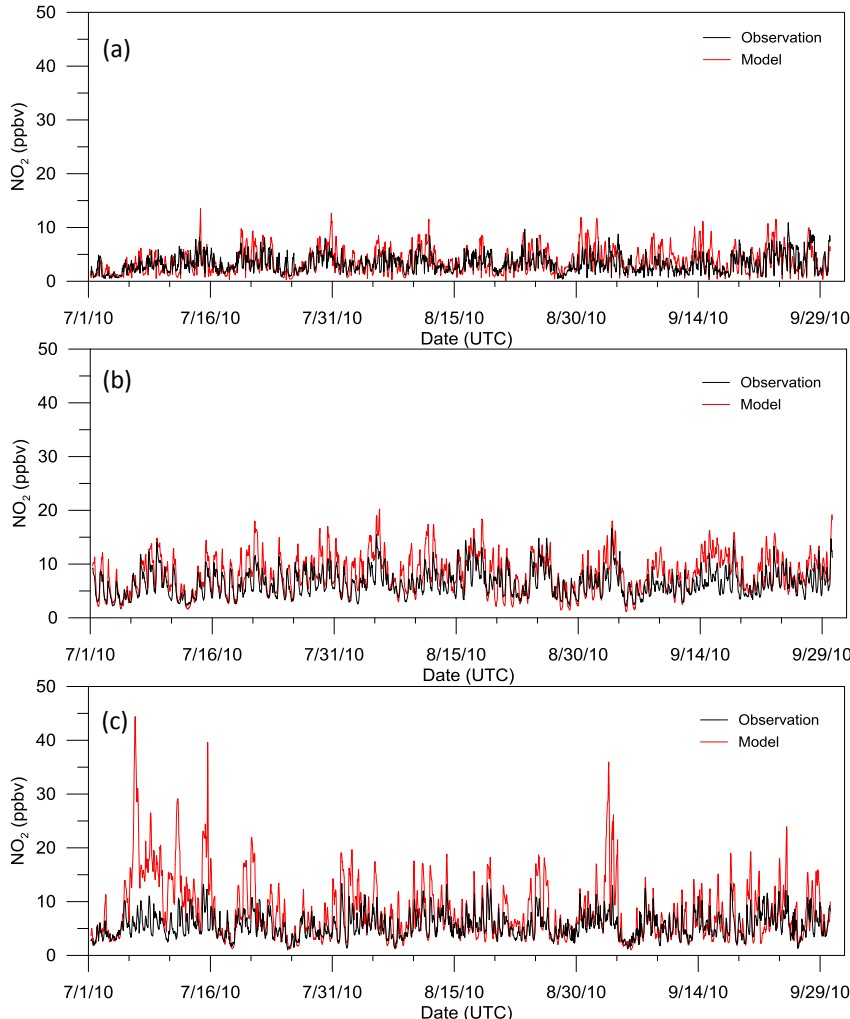

**Figure 6.** Regional-averaged NO$_2$ time series, modelled and observed: (a) northern; (b) south-western; (c) south-eastern.




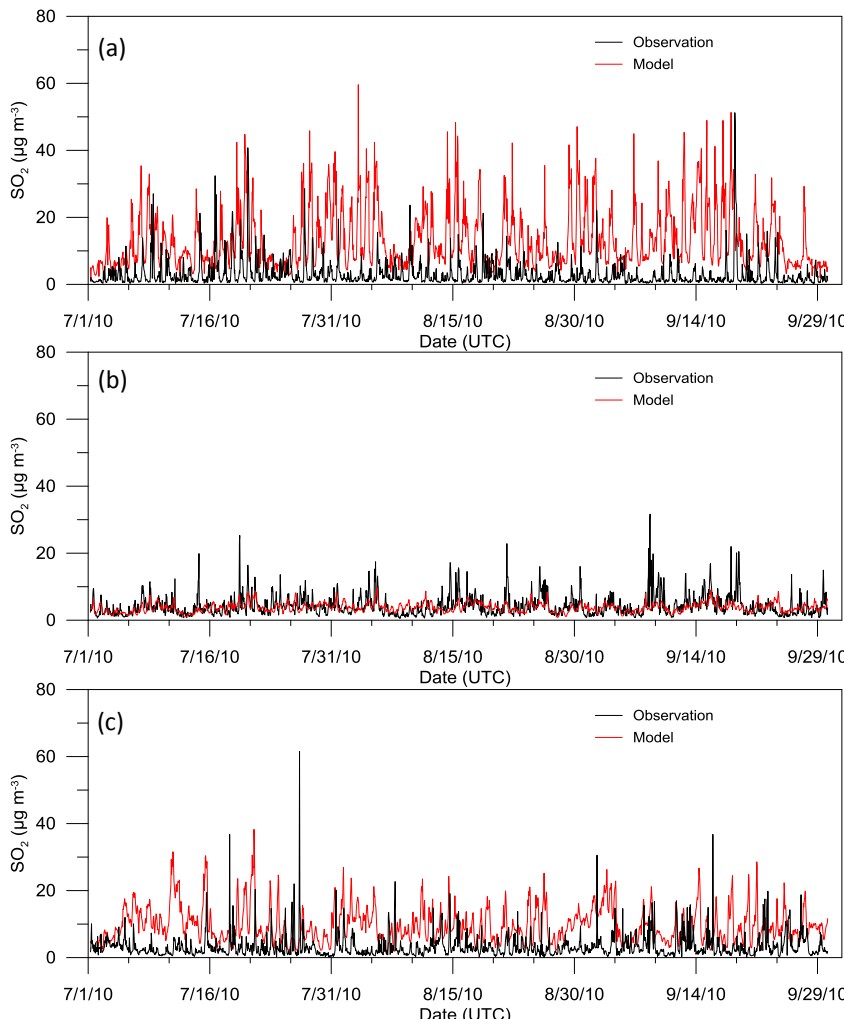

**Figure 7.** Regional-averaged SO₂ time series, modelled and observed: (a) northern; (b) south-western; (c) south-eastern.





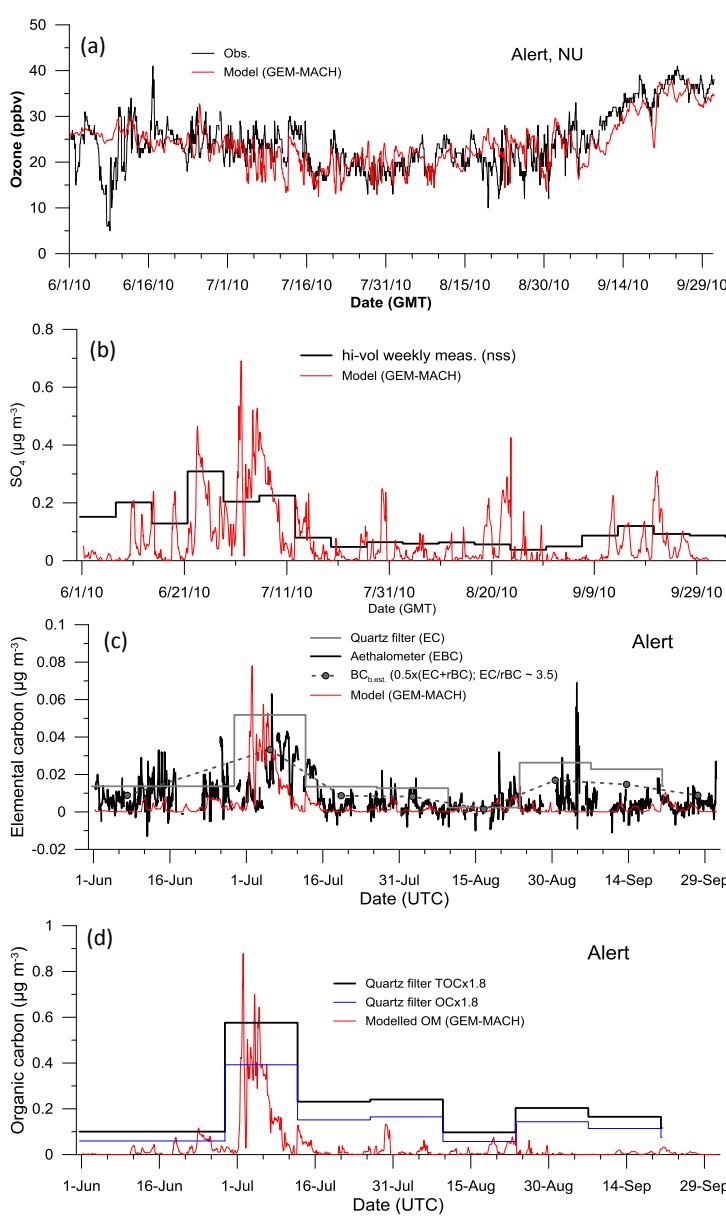

**Figure 8.** Comparison with observations at the Alert site for June-to-September, 2010: (a) O3, (b) sulphate, (c) EC, and (d) OC (OM).



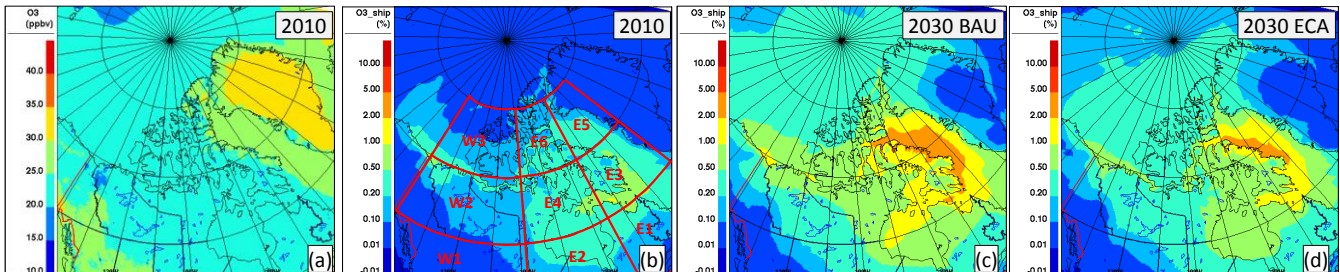

**Figure 9.** Modelled mean ambient $O_3$ concentrations for July-August-September (shipping season) for 2010 base year (a), and relative contribution from Canadian Arctic shipping emissions for the 2010 base year (b), 2030 BAU (c), and 2030 ECA (d). (The geographical subdivisions indicated on (b) are referred to in statistical assessment).

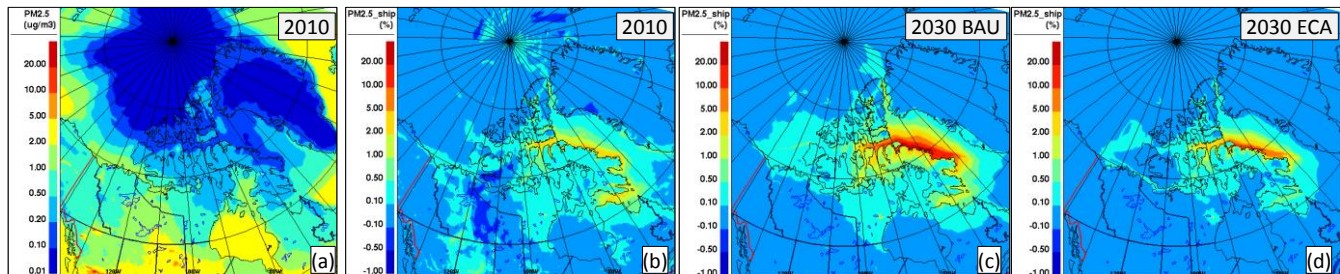

**Figure 10.** Modelled mean ambient $PM_{2.5}$ concentrations for July-August-September (shipping season): (a) 2010 base year, (c) 2030 BAU, and (e) 2030 ECA, and relative contribution from Canadian Arctic shipping emissions: (b) 2010 base year, (d) 2030 BAU, and (f) 2030 ECA.

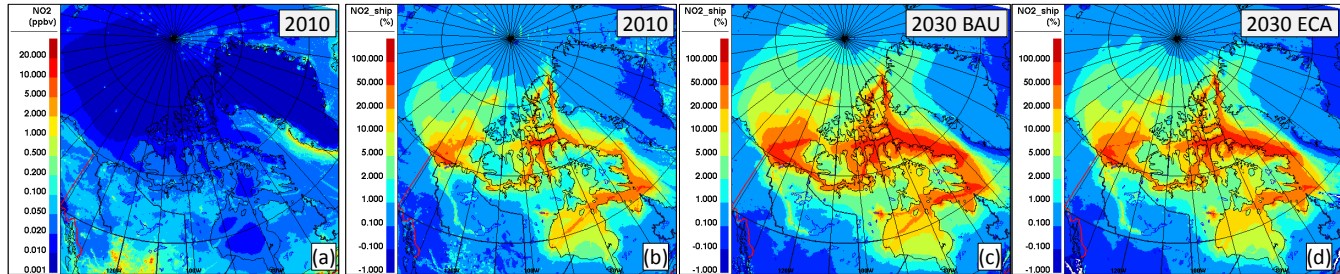

**Figure 11.** Modelled mean ambient $NO_2$ concentrations for July-August-September (shipping season): (a) 2010 base year, (c) 2030 BAU, and (e) 2030 ECA, and relative contribution from Canadian Arctic shipping emissions: (b) 2010 base year, (d) 2030 BAU, and (f) 2030 ECA.





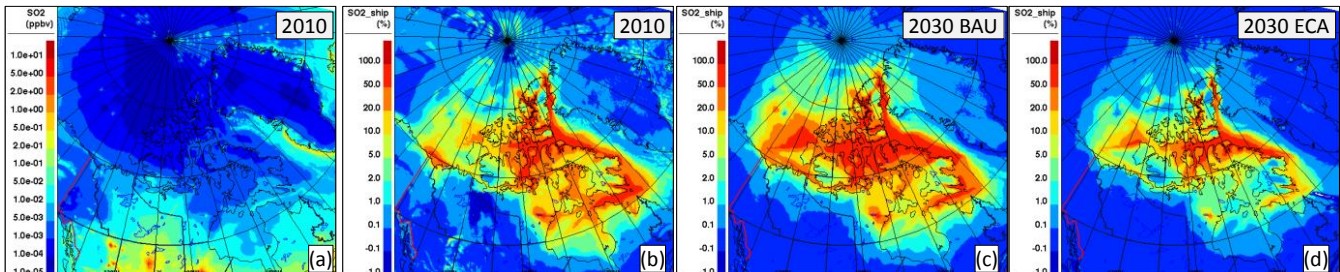

**Figure 12.** Modelled mean ambient SO$_2$ concentrations for July-August-September (shipping season): (a) 2010 base year, (c) 2030 BAU, and (e) 2030 ECA, and relative contribution from Canadian Arctic shipping emissions: (b) 2010 base year, (d) 2030 BAU, and (f) 2030 ECA.

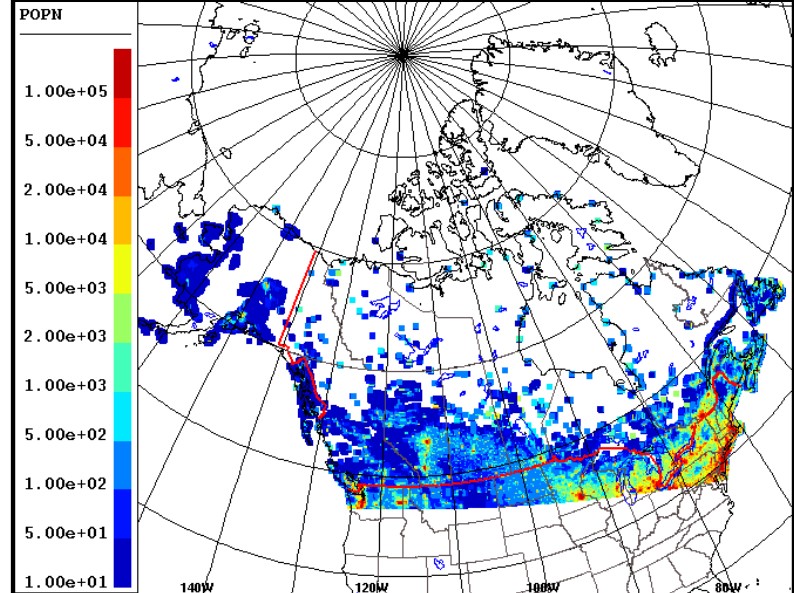

**Figure 13.** Gridded population based on 2010 US and 2011 Canadian census data.




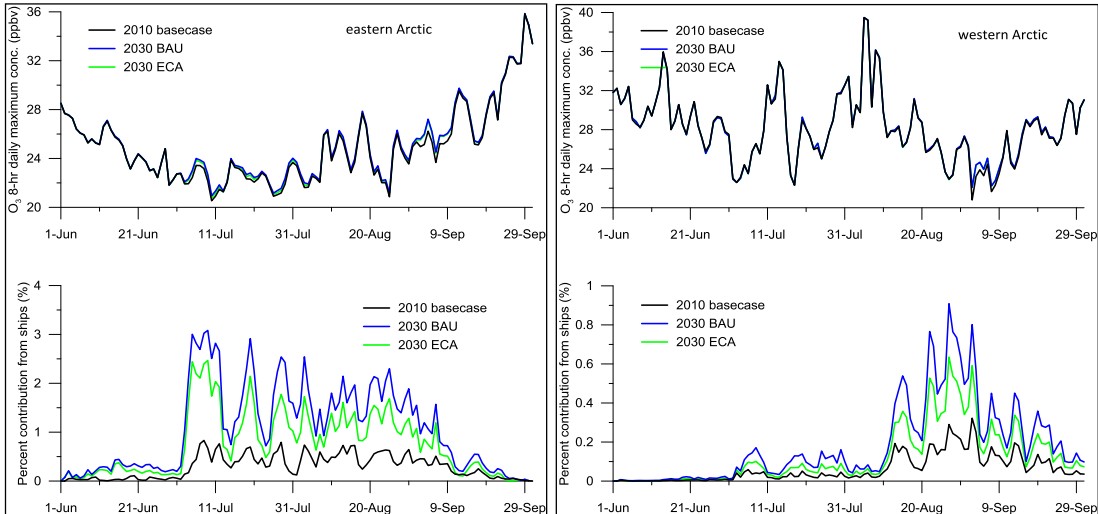

**Figure 14a.** Modelled population-weighted $O_3$ concentrations (8-hour daily maximum) over eastern and western Canadian Arctic (top panels), and contributions from Canadian Arctic shipping emissions (bottom panels).

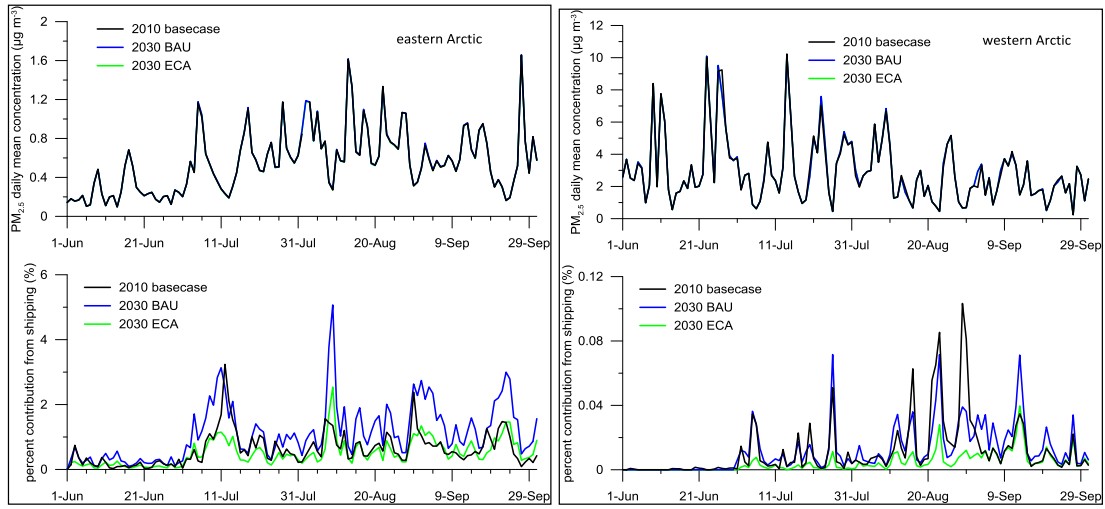

**Figure 14b.** Modelled population-weighted $PM_{2.5}$ concentrations over eastern and western Canadian Arctic (top panels), and contributions from Canadian Arctic shipping emissions (bottom panels).




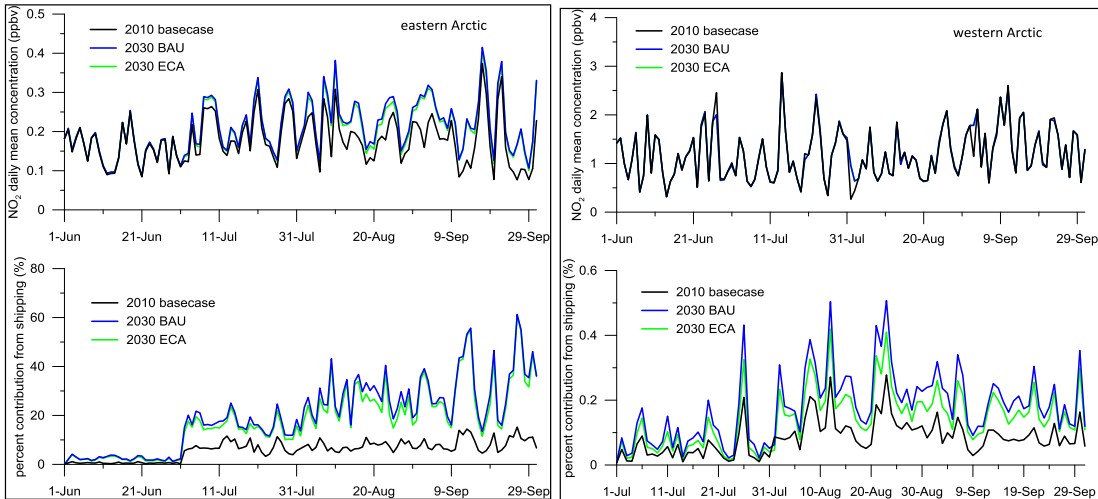

**Figure 14c.** Modelled population-weighted $NO_2$ concentrations over eastern and western Canadian Arctic (top panels), and contributions from Canadian Arctic shipping emissions (bottom panels).

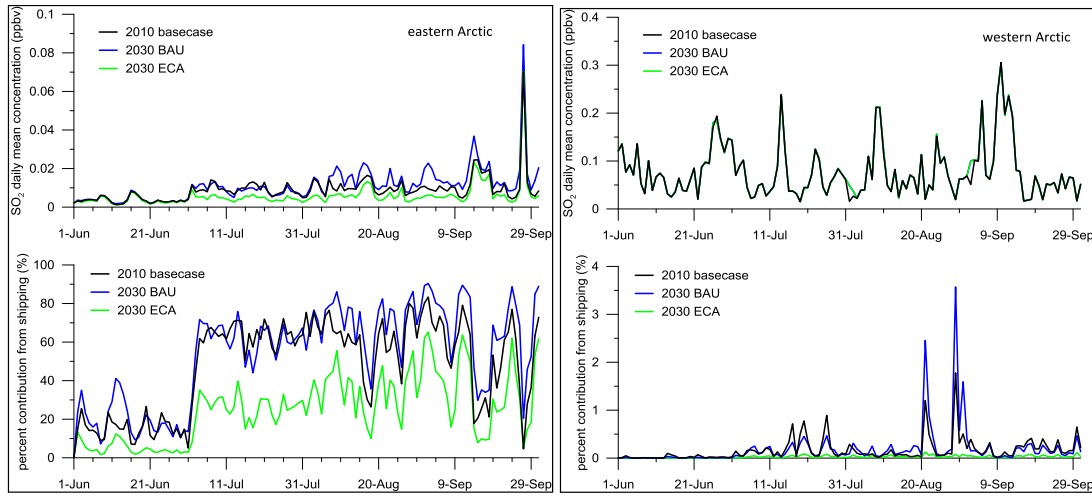

**Figure 14d.** Modelled population-weighted $SO_2$ concentration over eastern and western Canadian Arctic (top panels), and contributions from Canadian Arctic shipping emissions (bottom panels).




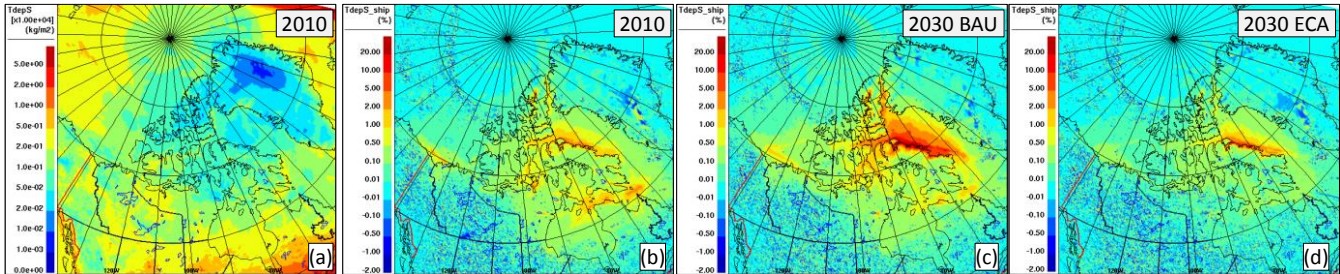

**Figure 15**. (a) Total sulphur deposition over the July-August-September period (accumulated) for the base case (2010); (b) Arctic shipping contribution at current (2010) level; (c) as in (b) for the 2030 BAU scenario; (d) as in (b) for the 2030 ECA scenario.

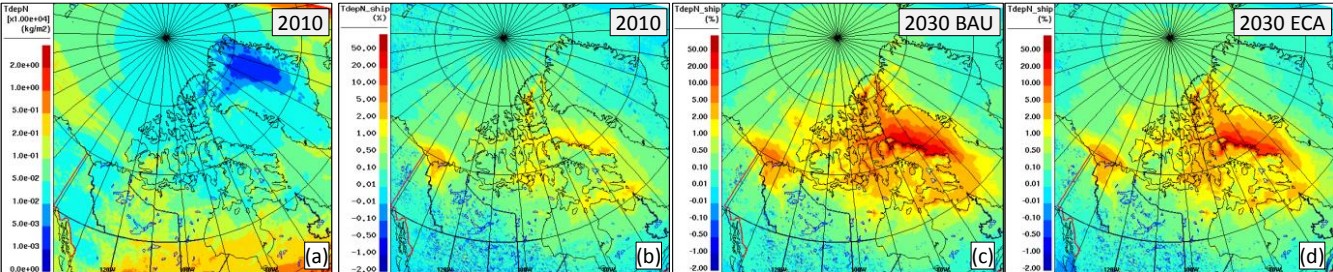

**Figure 16**. (a) Total nitrogen deposition over the July-August-September period (accumulated) for the base case (2010); (b) Arctic shipping contribution at current (2010) level; (c) as in (b) for the 2030 BAU scenario; (d) as in (b) for the 2030 ECA scenario.

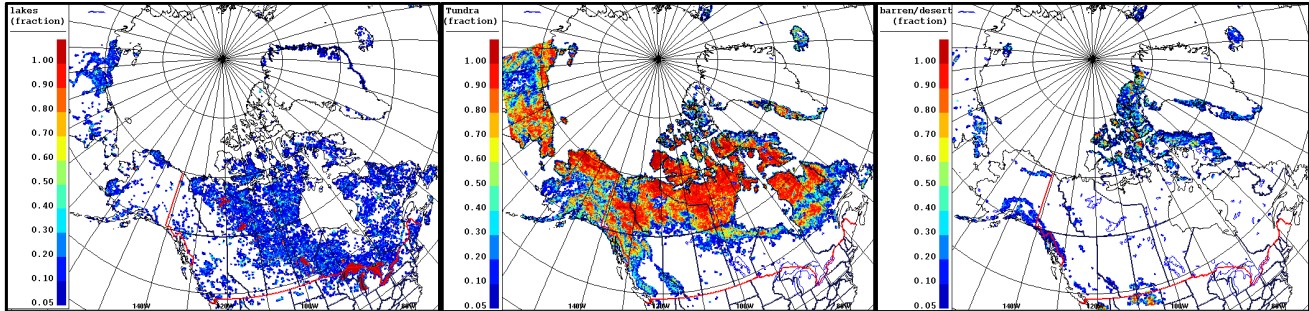

**Figure 17**. Gridded land-cover fractions for lakes, tundra, and barren/desert based on USGS v2.0 at 1-km resolution.



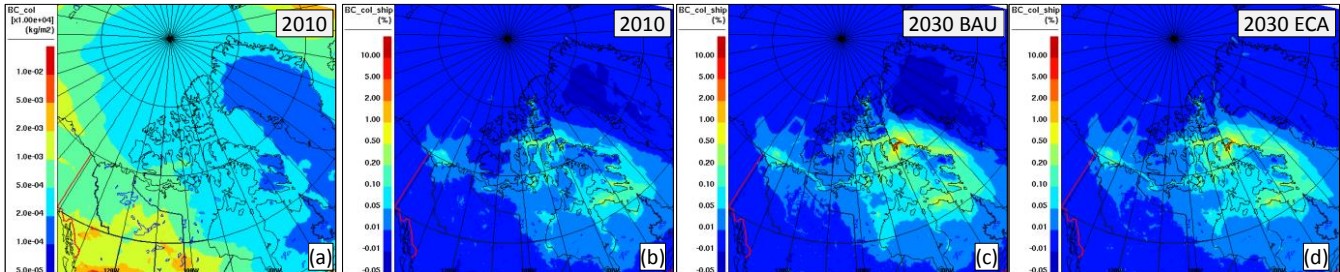

**Figure 18**. Modelled BC column loading (scaled up by $10^4$, in kg m$^{-2}$) averaged for over 2010 July-August-September period (a), and relative contributions from Canadian Arctic shipping emissions: (b) 2010 base year, (c) 2030 BAU, and (d) 2030 ECA.

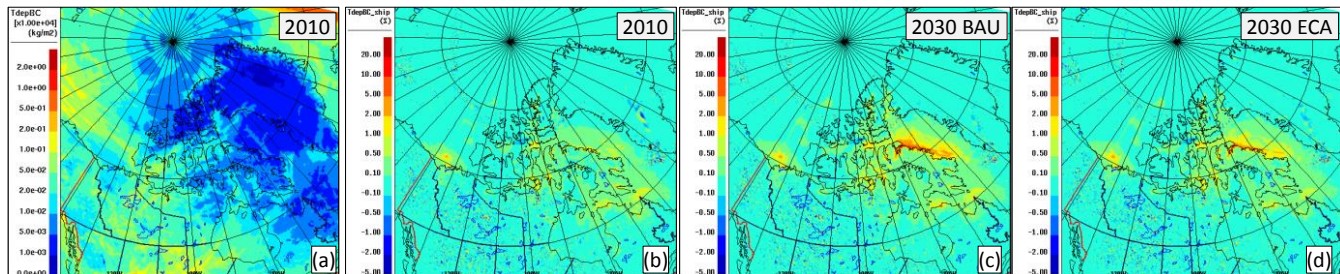

**Figure 19.** . Modelled total BC deposition flux (scaled up by $10^4$, in kg m$^{-2}$) accumulated over 2010 July-August-September period (a), and relative contributions from Canadian Arctic shipping emissions: (b) 2010 base year, (c) 2030 BAU, and (d) 2030 ECA.

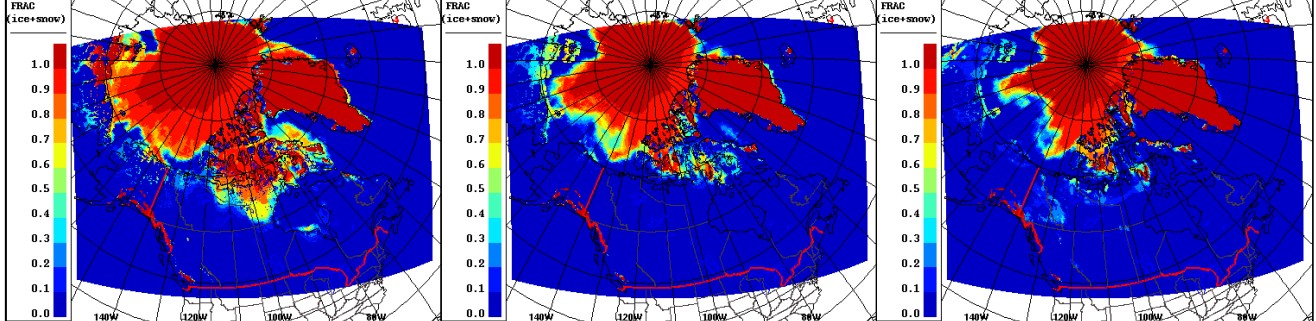

**Figure 20.** Monthly averaged ice/snow fraction for July, August, and September 2010.