# Peer review of "Assessing the impact of shipping emissions on air pollution in the Canadian Arctic and northern regions: current and future modelled scenarios"

_Atmospheric Chemistry and Physics, 2018_

## Referee Comment (RC1) · Anonymous Referee #1 · 15 May 2018

The paper presents a modelling study of the impact of shipping emissions in the Canadian Arctic. The work consist of two parts: Establishing of the national Canadian marine shipping inventory for 2010 and two projection for 2030, which was used as in put for the second part: atmospheric modelling of air quality by using the GEN-MACH model system in order to study the performance of the model system and the impacts of shipping emissions to the air quality.

The paper is generally well written and I recommend it to publish in ACP after revision.

General comments:

1. Major concern: The model runs have only been done for a short period March-October 2010 and only results for the shipping period (June –September) were presented. That is weakness of the paper, also because there are large seasonal variability in the arctic of air quality. It could be nice to see the model system performance a whole year, nice the see the overall contribution from shipping activities over a whole year, not only when shipping activities peaks.  Especially in the section about deposition it is a problem, because the results covers only 25% of a whole year and there are large seasonal variability of the deposition of N and S. In line 19-20 on page 20 there are a statement that these deposition levels are in general accordance with previous estimates e.g in Hole et al, 2009, which are a whole year estimate. It is a problem to extrapolate 3 month model results of deposition to a whole year deposition especially in the Northern part of Canada due the large seasonal variability of concentrations, surface conditions (snow-ice-forest-tundra) and the type and amount of precipitation.

2. Two kind of boundary conditions are used: The MACC-IFS for the arctic boundaries and the operational GEM-MACH forecast archives for the southern boundary, because the later should better represent transport from North America. It is little confusing to use to different boundary conditions. It could be nice to see how important the use GEM-MACH for the southern boundary are for the model performance is compared to use the global MACC-IFS 3-hour resolution input (is MACC-IFS so bad for the southern boundary?). It is actual mention in the text line 28-31 page 12 that some of the over prediction in the southern part of the model domain could be related to the boundary conditions.

Specific comments:

Page 2 line 27: large part of the particular matter is SO4 and is therefore a primary emission of SO4 in the shipping source area.

Page 4 line 1: the discussion of instant dilution of ship NOx emissions in global models. It is not only due the course spatial but also coarse temporal (monthly) because of the low number of ships in the arctic. Some models (.e.g. EMEP model) have special ship_NOX tracers which do not contribute to Ozone production only to HNO3.

Page 4 line 30: is the temporal emission from shipping really hourly so you can tracking the individual ships (see also my comment above)?

Page 5 line 26-page 6 line 18: I am missing figure (f.ex. of CO2 in order to avoid changes in emissions factors due to ECA) which shows the spatial distribution for 2030 which could be compared to 2010 of the ships emission and more information assumptions for the 2030 emissions inventory, e.g. the increase for the different ships sectors, emissions factors etc, so it is easier to compare the 2030 inventories with others.

Page 17 line 14: median and maximum percentage. Is it median and maximum of the 3 months average of the individual grid points inside the sectors or is it other spatial/temporal averaged concentrations?

---

## Referee Comment (RC2) · Anonymous Referee #2 · 25 Jun 2018

This study uses a regional air quality forecast model (GEM-MACH) coupled with a new shipping emission dataset based on real-time vessel movement data for 2010 to assess the impacts of shipping on air pollutant concentrations and deposition in the Canadian Arctic. Projections based on business as usual and controlled emission scenarios are used to estimate possible future changes in shipping impacts on ambient pollutant levels. The study provides a useful regional perspective on shipping impacts on Arctic pollution, and includes a comprehensive model evaluation for the Canadian Arctic region. I recommend publication in ACP once the following comments have been addressed by the authors.

[Figure]

General Comments

1. Although the paper is generally well written and clear, parts of the paper are quite long, and could be written more concisely. In particular, I would encourage the authors to consider whether a summary figure and more concise discussion of the model evaluation against observations could be included, with the minute detail of seasonal vs daily vs hourly comparisons and the information in Table 4 could be moved to supplementary information. Specific aspects of these evaluations of particular relevance for shipping impacts could be summarised in the main text.

2. An important aspect of how shipping may impact ambient pollution concentrations in coastal regions is related to the dispersion of pollutant plumes emitted by ships. This is partly related to the vertical boundary layer structure and stability. If possible, and data is available, it would be helpful to include somewhere some assessment of the model vertical BL structure (temperature profile, BL depth), or at least add a comment based on past evaluation of the model.

Specific Comments:

Page 3, line 7: Is all of the Arctic pristine? Depending on time of year, "background" PM concentrations may be very different in different areas? This may have implications for the impacts of shipping. How does the Canadian Arctic compare with e.g. N Siberia in terms of background (non-shipping) PM and ozone?

Page 4, lines 3-4: It would be useful to know more about what was assumed regarding the "limited number of transits of north-west passage". Which sea ice and climate projection scenario are these most consistent with? Which criteria went into this assumption?

Section 4.1: Discussion of model evaluation against observations. It would be helpful to compare model performance with other model studies focussed on similar regions where possible. e.g. the POLMIP models compared with ARCTAS aircraft data near

surface over Canada? (Emmons et al., 2015). Other global modelling studies? For the comparison of SO2 agains observations (but also relevant for other species) -it would be helpful to know how the regional averages and poor model performance are skewed by certain sites. e.g. can the comparisons with the oil sands sites be separated to show how the model compares away from this source (and specifically near to it)?

Section 5: The impacts of shipping on ambient pollutant concentrations in future may be closely tied climate system changes - particularly changes in sea ice. It would be interesting to consider how the conditions that may make shipping more favourable (reduced sea ice) may also contribute to a change in the impact of the shipping emissions. While additional model simulations are probably outside the scope of this study, could the authors comment on how reduced sea ice might be expected to impact ozone and PM in the summertime Arctic in context of their results?

Figures 4,5,6,7: How representative are the regional average concentrations, and is there much spread? e.g. it would be helpful to know how variable the concentrations are that make up each average. Can some spread be plotted in shading? This is also relevant to discussion of model evaluation above.

Editorial corrections:

Page 3, line 3: "theses" should be "these"

References:

Emmons, L. K., et al., The POLARCAT Model Intercomparison Project (POLMIP): overview and evaluation with observations, Atmos. Chem. Phys., 15, 6721-6744, doi:10.5194/acp-15-6721-2015, 2015.

---

## Author Comment (AC2) · 3 Aug 2018

Response to comments from Reviewer 2

(Reviewer's comments are copied *in italic* here; authors' responses are **in bold**)

*General Comments*

1. *Although the paper is generally well written and clear, parts of the paper are quite long, and could be written more concisely. In particular, I would encourage the authors to consider whether a summary figure and more concise discussion of the model evaluation against observations could be included, with the minute detail of seasonal vs daily vs hourly comparisons and the information in Table 4 could be moved to supplementary information. Specific aspects of these evaluations of particular relevance for shipping impacts could be summarised in the main text.*

**Response: This is a good suggestion. We have revised this section to be more concise. The revised Table 4 presents the hourly statistic scores only, while the extensive statistical scores (previously included in Table 4) are now presented in the supplementary material, and the discussions are more focused. The original manuscript did contain a paragraph at the end of this section summarising the main evaluation results in terms of model's ability of simulating ambient concentrations of criteria pollutants in northern Canada and the Arctic. We have strengthened the discussion in the revised version to emphasise the aspects of particular relevance to assessing shipping emission impacts.**

2. *An important aspect of how shipping may impact ambient pollution concentrations in coastal regions is related to the dispersion of pollutant plumes emitted by ships. This is partly related to the vertical boundary layer structure and stability. If possible, and data is available, it would be helpful to include somewhere some assessment of the model vertical BL structure (temperature profile, BL depth), or at least add a comment based on past evaluation of the model.*

**Response: Since the meteorological model (GEM, the hosting model of GEM-MACH) is the Environment and Climate Change Canada's operational numerical weather forecast model, the model evaluation in this study has been focused on atmospheric chemistry aspect. However, as the reviewer correctly pointed out, the model's ability to simulate the coastal marine boundary layer would have an important influence on assessing the shipping emission impact on ambient concentrations. Although GEM operational performance has continuously been evaluated against surface and upper air observations and compared against other NWP models of leading Operational Forecasting Centres in the world, the Arctic region alone had not been given significant attention in the past operational evaluation exercise. In order to address the reviewer's comment, we looked at the modelled vertical temperature profiles compared with upper air soundings at a number of coastal sites in the Arctic along the main shipping channels for the month of July in 2010. On average, the modelled vertical temperature profiles compare well with the observations (see Figure R2-1 below). We have also attempted to look into estimation of boundary layer heights. We compared the diagnosed boundary-layer heights from the modelled and observed profiles (determined from potential temperature profiles – the level of maximum**

vertical gradient, based on Stull 1988) at the upper air sites. There is an overall negative bias in model diagnosed BL heights compared to those diagnosed from the observation. However, it should be pointed out that under stable conditions (as the case of the Arctic marine BL) there is a large ambiguity in the definition of BL height – the diagnosed BL height can vary significantly depending on the particular method (or parameterization) used (e.g., Aliabadi et al., 2016, Atmosphere-Ocean 54 (1) 2016, 60–74). In the revised manuscript, we have added a brief discussion (at the end of section 4) on the model's ability in simulating the Arctic marine BL structure based on our simple assessment of modelled vertical temperature profiles. A more detailed examination of GEM's forecast capability in the Arctic is being carried out as part of the Year of Polar Prediction (YOPP) activities at Environment and Climate Change Canada.

[Figure]

Figure R2-1.  Comparison of modelled vertical temperature profiles with observations at 6 Arctic upper air sites for July 2010 (monthly means and standard deviations).

*Specific Comments:*

*Page 3, line 7: Is all of the Arctic pristine? Depending on time of year, "background" PM concentrations may be very different in different areas? This may have implications for the impacts of shipping. How does the Canadian Arctic compare with e.g. N Siberia in terms of background (non-shipping) PM and ozone?*

**Response: Here we are referring to eastern Arctic (or Canadian Arctic) during summer. It is true that the Arctic may not be as pristine as one would imagine in the wintertime due to long-range transport of pollutants from southern latitudes, and that parts of the Arctic are not as clean due to oil and gas activities (e.g., in northern Siberia and Alaska). We have revised the sentence to qualify this:**

**"Although Arctic marine shipping currently accounts for a small percentage of global shipping emissions, it makes a proportionally bigger impact on the environment than does shipping at lower latitudes due to the generally pristine Arctic background, particularly in the Canadian Arctic Archipelago."**

*Page 4, lines 3-4: It would be useful to know more about what was assumed regarding the "limited number of transits of north-west passage". Which sea ice and climate projection scenario are these most consistent with? Which criteria went into this assumption?*

**Response: The "limited number of transits via the Northwest Passage" was based on restricting the transit to bulk carrier vessels only. This is in consideration for the unpredictable ice conditions in the Canadian Arctic archipelago, even with the projected opening of the NWP (by mid-century based on available climate projections at the time), and economic viability (factoring in extra costs for vessel strengthening, loss of cargo carrying capacity, etc.). The projection for the 2030 NWP transit is based on a gradual (linear) increase from 2020 to a 2050 high-growth (or business-as-usual) scenario assuming bulk carriers would carry the 2050 Northern Europe-Asia bulk trade through the NWP. The 2050 bulk trade between Northern Europe and Asia was projected at an annual rate of increase based on historic trade data between 1975 and 2005. We have added a little more information on the 2030 projection in the revised manuscript but refer readers to the Innovation Maritime Report for more detailed information.**

*Section 4.1: Discussion of model evaluation against observations. It would be helpful to compare model performance with other model studies focussed on similar regions where possible. e.g. the POLMIP models compared with ARCTAS aircraft data near surface over Canada? (Emmons et al., 2015). Other global modelling studies? For the comparison of SO2 agains observations (but also relevant for other species) -it would be helpful to know how the regional averages and poor model performance are skewed by certain sites. e.g. can the comparisons with the oil sands sites be separated to show how the model compares away from this source (and specifically near to it)?*

**Response: We were not aware of the POLMIP project and are very glad that the reviewer brought the publication to our attention. Although we recognise that most of the models involved in the POLMIP project are global models at much coarser resolutions and that there may be inherent differences in**

comparing model with ground-level observations and aircraft measurements, we have made reference to the POLMIP model comparisons with the ARCTAS-B aircraft observations (O3, NO, NO2, and SO2) in the revised manuscript to provide a general comparison with other existing model studies in our region of interest. For the SO2 evaluation against observations, we have followed the reviewer's suggestion to examine the comparison excluding those sites strongly influenced by oil and gas production/processing activities (e.g., northern BC and Athabasca oil sands area). It clearly demonstrated that those sites were skewing the regional comparison: the large model positive biases were greatly reduced in this case. In fact the comparison shows that the model tends to under-predict SO2 (negative bias) at the more remote sites in the northern region (possibly due to the lack of representation of some local sources in the emission inventory, e.g., diesel generators, garbage burns). We have included this additional analysis in the revised manuscript.

*Section 5: The impacts of shipping on ambient pollutant concentrations in future may be closely tied climate system changes - particularly changes in sea ice. It would be interesting to consider how the conditions that may make shipping more favourable (reduced sea ice) may also contribute to a change in the impact of the shipping emissions. While additional model simulations are probably outside the scope of this study, could the authors comment on how reduced sea ice might be expected to impact ozone and PM in the summertime Arctic in context of their results?*

Response: Our study focused on assessing the contribution of Arctic shipping emissions to air pollution at current and projected future levels. We are isolating the changes to marine shipping emissions only and have not considered assessing the impact of shipping emissions under a changing climate. Indeed the impacts of shipping on ambient pollutant concentrations in the future will undoubtedly be linked to climate system changes. The reduced Arctic sea ice cover may have an impact on the atmospheric circulation and meteorological systems which will have an impact on the transport and transportation of atmospheric pollutants. The system is highly complex. To comment on how reduced sea ice would impact ozone and PM in future summertime Arctic in isolation without proper study would not be very meaningful in our opinion. We have added a statement in the "summary and conclusions" session to emphasise the focus (and limitation) of our assessment.

*Figures 4,5,6,7: How representative are the regional average concentrations, and is there much spread? e.g. it would be helpful to know how variable the concentrations are that make up each average. Can some spread be plotted in shading? This is also relevant to discussion of model evaluation above.*

Response: We have addressed the reviewer's question by adding shaded bands showing the lower to upper quartile range, highlighting a measure of spread and how well (or not) the mean represents the distrabution.

*Editorial corrections:*

*Page 3, line 3: "theses" should be "these"*

Response: Done. Thank you!

---

## Author Response (AR1)

Authors' responses to reviewers' comments – acp-2018-125

Gong et al.: "Assessing the impact of shipping emissions on air pollution in the Canadian Arctic and northern regions: current and future modelled scenarios"

Response to comments from Reviewer 1

(Reviewer's comments are copied *in italic* here; authors' responses and changes are **in bold**)

*General comments:*

1. *Major concern: The model runs have only been done for a short period March-October 2010 and only results for the shipping period (June –September) were presented. That is weakness of the paper, also because there are large seasonal variability in the arctic of air quality. It could be nice to see the model system performance a whole year, nice the see the overall contribution from shipping activities over a whole year, not only when shipping activities peaks. Especially in the section about deposition it is a problem, because the results covers only 25% of a whole year and there are large seasonal variability of the deposition of N and S. In line 19-20 on page 20 there are a statement that these deposition levels are in general accordance with previous estimates e.g in Hole et al, 2009, which are a whole year estimate. It is a problem to extrapolate 3 month model results of deposition to a whole year deposition especially in the Northern part of Canada due the large seasonal variability of concentrations, surface conditions (snow-ice-forest-tundra) and the type and amount of precipitation.*

   **Response: We understand the reviewer's concern on the length of the model runs. As our goal of the study is to assess the impact of marine shipping emissions over the Canadian Arctic waters, it made sense to us to focus on the Arctic shipping season. The following summary table (Table R1-1) shows that there is little shipping activity outside the period between July and October over the Canadian Arctic waters. The harsh ice and weather conditions, particularly over the Canadian Arctic Archipelago, make navigation through the Canadian Arctic waters extremely difficult outside the summer months. We decided to focus our analysis on the July-August-September period as it corresponds to the busiest shipping time in the Canadian Arctic and the Arctic boundary layer is the cleanest due to inefficient transport from mid latitudes to the Arctic during this time (e.g., Sharma et al., 2004, JGR, 109, D15203). We expect that the Arctic shipping would have the largest impact on air quality in the Arctic during this time period. The main basis for the reviewer's concern on the length of the model runs in our study is with regard to depositions of S and N which have large seasonal variability. When we compared our modelled deposition levels with the studies of Hole et al. (2009) and Vet et al. (2017), we scaled up the three-month (JAS) depositions to arrive at annual deposition estimates. For the base case (i.e., 2010 with Arctic marine shipping emissions) we did extend the model run for a full year. We have now computed the annual total deposition of S and N from the full-year simulation (shown in Figure R1-1 below). Our estimate of annual total depositions of S and N (based on the full-year model simulation) are 0.5-2 kg S ha$^{-1}$ and 0.2-1 kg N ha$^{-1}$, respectively, over the Canadian sub-Arctic, and 0.1-0.5 kg S ha$^{-1}$**

and 0.05-0.2 kg N ha$^{-1}$, respectively over the Canadian high-Arctic. These estimates can be directly compared with the estimates of Hole et al. (<20 to 70 mg m$^{-2}$, or <0.2 to 0.7 kg ha$^{-1}$, for both SO4 and NO3 over the eastern Arctic) and the estimates of Vet et al. (0.2-1 kg S or N ha$^{-1}$ over eastern sub-Arctic and 0-0.2 kg S or N ha$^{-1}$ over eastern high-Arctic). We have revised the discussions on S and N deposition to include the new annual results (see below under "Changes"). For assessing the shipping emission contributions to the S and N deposition, we have also computed the ship emission contributions to the July-to-October (or JASO, 4 months) accumulated deposition fluxes of S and N at the 2010 level (in consideration of significant shipping activities in October as shown in Table R1-1 below). Results (in terms of percentage contributions) are very similar to our existing results based on the JAS period (as seen in Figure 15(b) and 16(b) in the manuscript). Since the shipping activities in the Canadian Arctic waters are seasonal (summer), we feel that it is more meaningful to look at the shipping contributions during the shipping season.

Table R1-1. 2010 monthly Canadian Arctic marine shipping emissions

|      | Jan | Feb | Mar | Apr | May | Jun | Jul | Aug | Sept | Oct | Nov | Dec |
|------|-----|-----|-----|-----|-----|-----|-----|-----|------|-----|-----|-----|
| CO   | 1   | 0   | 0   | 0   | 0   | 5   | 44  | 95  | 86   | 54  | 11  | 2   |
| NH3  | 0   | 0   | 0   | 0   | 0   | 0   | 1   | 1   | 1    | 1   | 0   | 0   |
| NOx  | 12  | 0   | 0   | 5   | 0   | 62  | 508 | 1,120 | 1,022 | 634 | 128 | 26  |
| PM10 | 1   | 0   | 0   | 0   | 0   | 4   | 29  | 56  | 52   | 34  | 6   | 1   |
| PM25 | 1   | 0   | 0   | 0   | 0   | 4   | 26  | 52  | 48   | 31  | 6   | 0   |
| SO2  | 6   | 0   | 0   | 4   | 0   | 32  | 200 | 369 | 341  | 234 | 40  | 1   |
| VOC  | 0   | 0   | 0   | 0   | 0   | 2   | 18  | 40  | 36   | 22  | 5   | 1   |

[Figure]

Figure R1-1. Annual total deposition of S (left panel) and N (right panel) based on the full-year model simulation of the base case (i.e., 2010 with Arctic marine shipping emissions)

**Changes:**

- Revised 2[nd] paragraph in section 5.2 ("On deposition of S and N") to incorporate the annual deposition estimates based on the extended full-year simulation, and added a new figure (Figure R1-1 shown above) to supplementary materials (Figure S9).
- Revised 2[nd] last paragraph in section 5.2 ("On deposition of S and N") to incorporate land-cover-weighted annual deposition values from the extended full-year (2010) simulation in comparison with current critical load of acidity and nutrient N; added a new table (Table S3 in supplementary materials) showing the land-cover weighted deposition of S and N over eastern Canadian Arctic from the full-year simulation (2010) for three periods: July-to-September (JAS), July- October (JASO), and annual (Table S3 in supplementary materials).

2. *Two kind of boundary conditions are used: The MACC-IFS for the arctic boundaries and the operational GEM-MACH forecast archives for the southern boundary, because the later should better represent transport from North America. It is little confusing to use to different boundary conditions. It could be nice to see how important the use GEM-MACH for the southern boundary are for the model performance is compared to use the global MACC-IFS 3-hour resolution input (is MACC-IFS so bad for the southern boundary?). It is actual mention in the text line 28-31 page 12 that some of the over prediction in the southern part of the model domain could be related to the boundary conditions.*

Response: The daily chemical lateral boundary condition fields used in the final model simulations were constructed from blending MACC-IFS chemical reanalysis for 2010 (provided by ECMWF/MACC-II at 1.25 x 1.25° resolution, interpolated to the 15-km resolution model grids) with the GEM-MACH operational forecast archives (at collocated 15-km resolution model grids; the operational GEM-MACH forecast domain overlaps a portion of the Arctic domain – see Figure R1-2 below). The consideration was that, given the better (finer) resolution used by the operational GEM-MACH (15-km) forecast in comparison to MACC-IFS reanalysis at a resolution of 1.125°, we decided to make use of the GEM-MACH operational archive as much as possible to ensure a better capture of the regional transport from U.S. northeast into our model domain, which, in our opinion, would improve our model simulation. Some initial evaluations were carried out when we were testing various chemical boundary conditions (including climatology-based boundary conditions). However we did not conduct a formal sensitivity analysis on the importance of the southern boundary condition to model performance in the Canadian Arctic and northern regions. The analysis of model performance described in section 4 indicated that the southern boundary condition mainly influenced model results at sites close to the southern boundary while it had significantly less influence on model results over central and northern Canada. A similar blending approach was used for merging the North American regional emissions (processed for the 15 km resolution model grids) with the HTAP emissions used on the portion of the domain outside the North American continent for this study.

[Figure]

**Figure R1-2. The GEM-MACH Arctic domain (dark blue background) and the operational GEM-MACH forecast domain in 2010 (shown in lighter blue, foreground), both at 15-km resolution with collocating grids.**

*Specific comments:*

*Page 2 line 27: large part of the particular matter is SO4 and is therefore a primary emission of SO4 in the shipping source area.*

**Response: While it is true that a large part of particulate matters in ship plumes is particulate sulfate (SO4), most of the sulfate is formed from secondary oxidation of SO2 emitted from ship stacks. Although ship emissions do contain primary SO4, the sulfur emission from ship is mainly in the form of gaseous SO2. For completeness, in the revised manuscript, we have added particulate sulfate in the suite of ship-emitted gases and particles explicitly mentioned in the introduction.**

**Changes: Revised sentence on page 2, bottom paragraph: "Shipping is an important source of air pollutants. Emissions of exhaust gases and particles from ocean-going ships contain carbon dioxide (CO2), nitrogen oxides (NOx), carbon monoxide (CO), volatile organic compounds (VOC), sulfur dioxide (SO2), _particulate sulfate (SO4),_ black carbon (BC), and particulate organic matter (OM)."**

*Page 4 line 1: the discussion of instant dilution of ship NOx emissions in global models. It is not only due the course spatial but also coarse temporal (monthly) because of the low number of ships in the arctic. Some models (.e.g. EMEP model) have special ship_NOX tracers which do not contribute to Ozone production only to HNO3.*

**Response: We appreciate the reviewer's comment on other potential issues which may impact model assessment of ship emissions, in addition to the spatial resolution being discussed. Here we try to summarise the findings from existing studies suggesting that the non-linear effects associated with the unrealistic instant dilution of ship NOx emissions in global models run at coarse resolutions may affect model assessments of ship emission impact. This is to provide a context for our study and the approach we are taking in using a regional model at higher resolution. We recognise that insufficient temporal representation of ship emissions and other simplifications in some of the current models, such as the example given by the reviewer of special treatment for ship-NOx tracers in the EMEP model, may result in additional uncertainties. However, we are not aware of any existing studies**

addressing these aspects, and these aspects are not the focus of our current study. Nevertheless, the issues raised by the reviewer are important and should potentially be investigated in future studies.

*Page 4 line 30: is the temporal emission from shipping really hourly so you can tracking the individual ships (see also my comment above)?*

**Response: The base inventory is processed using the Marine Emission Inventory Tool (MEIT) based on ship movements of individual vessels tracked by the Canadian Coast Guard's tracking/logging systems, as explained in the manuscript (page 4 line 20 – 30), and the processed emissions are available at various time levels (e.g., monthly, daily, and hourly). However, as explained in the "Modelling system and simulation setup" section (under "Canadian marine shipping emissions"), for further processing to model-ready marine emissions, link-based monthly ship emissions by ship track, ship types, and fuel type were obtained from the MEIT database, along with ship route polygons and associated vessel activity information. The monthly emissions, aggregated into four ship classes, were mapped onto model grids, along ship tracks, in a form of aggregated point sources and further allocated to hourly emissions, by applying uniform temporal profiles for day-of-week and hour-of-day in the SMOKE emission processing system ([http://www.cmascenter.org/smoke/](http://www.cmascenter.org/smoke/)). (See page 8, line 9-22, in the original submitted manuscript, or page 8, line 16 – 32, in the revised manuscript).**

*Page 5 line 26-page 6 line 18: I am missing figure (f.ex. of CO2 in order to avoid changes in emissions factors due to ECA) which shows the spatial distribution for 2030 which could be compared to 2010 of the ships emission and more information assumptions for the 2030 emissions inventory, e.g. the increase for the different ships sectors, emissions factors etc, so it is easier to compare the 2030 inventories with others.*

**Response: We have added $CO_2$ numbers in Table 1 (both 2010 and projected 2030 BAU scenario, for different ship categories). The projected increase in shipping activities in 2030 is reflected in number of trips shown in Table 1 for different ship categories/sectors. We have also added an additional plot (Figure 2(b)) of the processed model-ready Canadian marine shipping emission of NOx projected for the August 2030 (BAU) to compare with the 2010 August NOx emission shown in Figure 2(a) (previously Figure 2). The emission factors are in accordance with fuel type usage (with considerations for compliance to the current and future IMO regulations). We have added further clarification on emission factors used for the 2030 projection in the revised manuscript.**

**Changes:**

- **Revised Table 1 to include emission estimates for CO2e per vessel category for 2010 and 2030 (BAU).**
- **Revised Figure 2 to include a plot showing the model-ready NOx marine shipping emissions for August 2030 (BAU scenario).**
- **Revised text in the last paragraph of Section 2 (on page 6) to clarify emission rates used for the projected 2030 Arctic marine shipping emission estimate: "In estimating emissions related to the projected shipping activities, _the emission rates were adjusted to reflect the regulatory (both domestic and international) and technological changes, such as fuel standards and fleet turnover._**

The MARPOL Annex VI global cap on the sulphur content of 0.5% for fuel oil used on board ships is assumed to be in place in the business-as-usual (BAU) scenario".

*Page 17 line 14: median and maximum percentage. Is it median and maximum of the 3 months average of the individual grid points inside the sectors or is it other spatial/temporal averaged concentrations?*

**Response:** **It is the former, i.e., the mean, median, maximum are based on shipping contributions to the 3-month mean concentrations evaluated at individual grid points within a given geographical sector. We have added a statement to clarify this in the revised version.**

**Changes:** **Revised text in the 1st paragraph of the sub-section "Statistical assessment by geographical sectors": "Table 6 summaries the mean, median, and maximum percentage contributions from Arctic shipping emissions to the JAS average ambient concentrations of criteria pollutants for each of the 9 sectors. The percentage contributions (as defined in (1)) were evaluated at individual grid points and statistics were then computed over all grid points within a given geographical sector."**

Response to comments from Reviewer 2

(Reviewer's comments are copied *in italic* here; authors' responses and changes are **in bold**)

*General Comments*

1. *Although the paper is generally well written and clear, parts of the paper are quite long, and could be written more concisely. In particular, I would encourage the authors to consider whether a summary figure and more concise discussion of the model evaluation against observations could be included, with the minute detail of seasonal vs daily vs hourly comparisons and the information in Table 4 could be moved to supplementary information. Specific aspects of these evaluations of particular relevance for shipping impacts could be summarised in the main text.*

   **Response: This is a good suggestion. We have revised this section to be more concise. The revised Table 4 presents the hourly statistic scores only, while the extensive statistical scores (previously included in Table 4) are now presented in the supplementary materials. We have removed some of the discussion on daily vs. hourly comparisons, and removed some of the measurement details to make the discussions more focused. There is a paragraph at the end of this section summarising the main evaluation results in terms of model's ability of simulating ambient concentrations of criteria pollutants in northern Canada and the Arctic as in the original manuscript.**

   **Changes:**

   - **Revised Table 4 to include a summary of hourly statistical scores only (while the full sets of statistical scores, hourly, daily, and seasonal, are presented in Table S1 in the Supplementary Materials); corresponding revised text in the opening paragraph in section 4.1 ("Statistical scores"); shortened discussion on O3 scores (removal of the discussion on daily and seasonal correlation coefficients).**
   - **Revised discussion on comparison with the measurements at Alert – removal of some details on measurement techniques (under "Canadian high Arctic site, Alert" in 4.2).**

2. *An important aspect of how shipping may impact ambient pollution concentrations in coastal regions is related to the dispersion of pollutant plumes emitted by ships. This is partly related to the vertical boundary layer structure and stability. If possible, and data is available, it would be helpful to include somewhere some assessment of the model vertical BL structure (temperature profile, BL depth), or at least add a comment based on past evaluation of the model.*

   **Response: Since the meteorological model (GEM, the hosting model of GEM-MACH) is the Environment and Climate Change Canada's operational numerical weather forecast model, the model evaluation in this study has been focused on atmospheric chemistry aspect. However, as the reviewer correctly pointed out, the model's ability to simulate the coastal marine boundary layer would have an important influence on assessing the shipping emission impact on ambient**

concentrations. Although GEM operational performance has continuously been evaluated against surface and upper air observations and compared against other NWP models of leading Operational Forecasting Centres in the world, the Arctic region alone had not been given significant attention in the past operational evaluation exercise. In order to address the reviewer's comment, we looked at the modelled vertical temperature profiles compared with upper air soundings at a number of coastal sites in the Arctic along the main shipping channels for the month of July in 2010. On average, the modelled vertical temperature profiles compare well with the observations (see Figure R2-1 below). We have also attempted to look into estimation of boundary layer heights. We compared the diagnosed boundary-layer heights from the modelled and observed profiles (based on bulk Richardson number) at the upper air sites. On average the model and observation diagnosed BL heights are within +/- 30 % of each other (see Figure R2-2 below). However, it should be pointed out that under stable conditions (as the case of the Arctic marine BL) there is a large ambiguity in the definition of BL height – the diagnosed BL height can vary significantly depending on the particular method (or parameterization) used (e.g., Aliabadi et al., 2016, Atmosphere-Ocean 54 (1) 2016, 60–74). In the revised manuscript, we have added a brief discussion (at the end of section 4) on the model's ability in simulating the Arctic marine BL structure based on our simple assessment of modelled vertical temperature profiles and diagnosed BL heights. A more detailed examination of GEM's forecast capability in the Arctic is being carried out as part of the Year of Polar Prediction (YOPP) activities at Environment and Climate Change Canada.

[Figure]

Figure R2-1. Comparison of modelled vertical temperature profiles with observations at 6 Arctic upper air sites for July 2010 (monthly means and standard deviations).

[Figure]

| site | $H_{obs}$ (m) | $H_{mod}$ (m) |
|------|---------------|---------------|
| Alert | 259.8 | 239.1 |
| Eureka | 282.8 | 223.8 |
| Hall Beach | 287.0 | 330.6 |
| Inuvik | 226.4 | 315.4 |
| Resolute | 267.4 | 308.8 |
| Barrow | 286.3 | 326.3 |

Figure R2-2. Diagnosed PBL heights from modelled (pink) and observed (cyan) profiles at selected Arctic sites for the month of July 2010. The monthly averaged PBL heights are shown in the table insert.

**Changes:**

- **a new paragraph added at the end of section 4 to briefly discuss model's ability of simulating the Arctic marine boundary layer:**

"**The model evaluation conducted in this study is mainly focused on the atmospheric chemistry aspect. However, the model's ability to simulate the vertical structure and stability of the coastal marine boundary layer has an important influence on assessing the shipping emission impact on ambient concentrations. Although the operational performance of the meteorological model GEM (the hosting model for GEM-MACH) has continuously been evaluated against surface and upper air observations and compared against other NWP models of leading Operational Forecasting**

Centres in the world, the Arctic region alone had not been given significant attention in the past operational evaluation exercise. To evaluate the GEM-MACH performance in simulating the Arctic marine boundary layer, we compared the modelled vertical temperature profiles with upper air soundings at a number of coastal sites in the Arctic along the main shipping channels for the month of July in 2010. On average, the modelled vertical temperature profiles compare well with the observations (see Supplementary Material, Figure S2a). We also attempted to diagnose boundary-layer (BL) heights based on bulk Richardson number, following Mahrt (1981) and Aliabadi et al. (2016a), from both modelled and observed profiles at these selected Arctic sites. On average, the model and observation diagnosed BL heights are within ± 30% of each other (see Figure S2b). Particularly, for Resolute site, the model and observation diagnosed BL heights for July, averaged at 315.4 m and 267.4 m, respectively, are comparable to the estimated BL heights, 274±164 m, over the same area during a recent field campaign in July 2014 (Aliabadi et al., 2016b). It should be pointed out, however, there is a large ambiguity in the definition of BL height under stable conditions (such as the case of the Arctic marine BL) and the diagnosed BL height can vary considerably depending on the particular method (or parameterization ) used (e.g., Aliabadi et al., 2016a). A more detailed examination of GEM's forecast capability in the Arctic is being pursued under the Year of Polar Prediction (YOPP) initiative (https://public.wmo.int/en/projects/polar-prediction)."

- **New figures added to Supplementary Materials (Figure S2a and S2b) to accompany the discussion above.**

*Specific Comments:*

*Page 3, line 7: Is all of the Arctic pristine? Depending on time of year, "background" PM concentrations may be very different in different areas? This may have implications for the impacts of shipping. How does the Canadian Arctic compare with e.g. N Siberia in terms of background (non-shipping) PM and ozone?*

**Response:** Here we are referring to eastern Arctic (or Canadian Arctic) during summer. It is true that the Arctic may not be as pristine as one would imagine in the wintertime due to long-range transport of pollutants from southern latitudes, and that parts of the Arctic are not as clean due to oil and gas activities (e.g., in northern Siberia and Alaska). We have revised the sentence to qualify this.

**Changes:** Revised sentence on page 3, line 5-7:

"Although Arctic marine shipping currently accounts for a small percentage of global shipping emissions, it makes a proportionally bigger impact on the environment than does shipping at lower latitudes due to the generally pristine Arctic background, *particularly in the Canadian Arctic Archipelago*."

*Page 4, lines 3-4: It would be useful to know more about what was assumed regarding the "limited number of transits of north-west passage". Which sea ice and climate projection scenario are these most consistent with? Which criteria went into this assumption?*

**Response:** **The "limited number of transits via the Northwest Passage" was based on restricting the transit to bulk carrier vessels only. This is in consideration for the unpredictable ice conditions in the Canadian Arctic archipelago, even with the projected opening of the NWP (by mid-century based on available climate projections at the time), and economic viability (factoring in extra costs for vessel strengthening, loss of cargo carrying capacity, etc.). The projection for the 2030 NWP transit is based on a gradual (linear) increase from 2020 to a 2050 high-growth (or business-as-usual) scenario assuming bulk carriers would carry the 2050 Northern Europe-Asia bulk trade through the NWP. The 2050 bulk trade between Northern Europe and Asia was projected at an annual rate of increase based on historic trade data between 1975 and 2005. We have added a little more information on the 2030 projection in the revised manuscript but refer readers to the Innovation Maritime Report for more detailed information.**

**Changes:** **revised description on projected NWP transits (page 6):** "**In predicting future shipping traffic, a limited number of transits via the Northwest Passage were assumed***, based on restricting the transit to bulk carrier vessels only and economic viability[1]***"**, and a footnote:** "*[1] The projection for the 2030 NWP transit is based on a gradual (linear) increase from 2020 to a 2050 high-growth (or business-as-usual) scenario assuming that bulk carriers would carry the 2050 Northern Europe-Asia bulk trade through the NWP. The 2050 bulk trade between Northern Europe and Asia was projected at an annual rate of increase based on historic trade data between 1975 and 2005 (see Innovation Maritime and SNC-Lavalin Environment, 2013)*"**.

*Section 4.1: Discussion of model evaluation against observations. It would be helpful to compare model performance with other model studies focussed on similar regions where possible. e.g. the POLMIP models compared with ARCTAS aircraft data near surface over Canada? (Emmons et al., 2015). Other global modelling studies? For the comparison of SO2 agains observations (but also relevant for other species) -it would be helpful to know how the regional averages and poor model performance are skewed by certain sites. e.g. can the comparisons with the oil sands sites be separated to show how the model compares away from this source (and specifically near to it)?*

**Response:** **We were not aware of the POLMIP project and are very glad that the reviewer brought the publication to our attention.  Although we recognise that most of the models involved in the POLMIP project are global models at much coarser resolutions and that there may be inherent differences in comparing model with ground-level observations and aircraft measurements, we have made reference to the POLMIP model comparisons with the ARCTAS-B aircraft observations (O3, NO2, and SO2) over northern Canada and into the Arctic in 2008 in the revised version. We have also now made reference to Shindell et al. (2008) where simulations using several global models were compared with long-term measurements at a few Arctic sites (including Alert and Barrow). These discussions are included in a new paragraph towards the end of section 4 to provide a general comparison with other existing model studies in our region of interest.  For the SO2 evaluation against observations, we have**

followed the reviewer's suggestion to examine the comparison excluding those sites strongly influenced by oil and gas production/processing activities (e.g., northern BC and Athabasca oil sands area). It clearly demonstrated that those sites were skewing the regional averages. The model and observations are in much better agreement when these sites are removed.  We have included this additional analysis in the revised manuscript.

**Changes:**

- A new paragraph (second last paragraph in section 4) to compare with other existing modelling studies with a focus on the Arctic:

  "While there has not been many regional modelling studies focussed on the Arctic and northern regions, there are some existing studies mostly using global models with a focus on the Arctic. For example, Emmons et al. (2015) reported a multi-model intercomparison project where model simulations performed using a number of models (9 global and two regional) were compared with observations conducted during the 2008 International Polar Year in the Arctic. In particular, comparisons were made with aircraft measurements conducted over northern Canada and into the Arctic over a 12-day period during late June to early July. They found that models generally under-predicted O3 and SO2 in mid troposphere and over-predicted NO2 in the boundary layer during this summer period. A direct comparison in terms of model performance to the current study is difficult to make as the model evaluation in the current study is based on surface observations and over a longer time period. Shindell et al. (2008) also compared global model simulations, conducted under the Task Force on Hemispheric Transport of Air Pollution (HTAP), against long-term observations at selected Arctic sites including Alert and Barrow. They found that the models generally under-predict O3 at Barrow during summer by as much as 10 ppb, and that models performed poorly in predicting sulfate and BC at Alert. In comparison, the model evaluation from the current study demonstrates much better model skills in predicting the ambient concentrations of these pollutants in the Arctic (e.g., comparisons shown in Figure 8)."

- Revised Figure 7a to include the regional averaged SO2 time series (both observation and model) when the sites near oil and gas facilities in Athabasca oil sands and northeastern BC are excluded, and corresponding revised discussion on the comparison of regional averaged SO2 time series (page 13):

  "As a reflection of the $SO_2$ regional statistical scores discussed above, the comparison of regional averaged time series of the observed and modelled $SO_2$ for the northern region is strongly influenced by the sites located near oil and gas facilities.  *Also shown in Figure 7a are the regional averaged time series excluding the sites in the Athabasca oil sands and northeastern BC oil and gas industry areas (in dashed lines). It is evident that these sites are skewing the regional averages. The large discrepancies between the model simulation and observations at these sites are indicative of the possible deficiency in the existing emission*

*inventory and the emission processing for these facilities. The model and observations are in much better agreement at the northern sites away from the oil and gas facilities.*"

*Section 5: The impacts of shipping on ambient pollutant concentrations in future may be closely tied climate system changes - particularly changes in sea ice. It would be interesting to consider how the conditions that may make shipping more favourable (reduced sea ice) may also contribute to a change in the impact of the shipping emissions. While additional model simulations are probably outside the scope of this study, could the authors comment on how reduced sea ice might be expected to impact ozone and PM in the summertime Arctic in context of their results?*

**Response: Our study focused on assessing the contribution of Arctic shipping emissions to air pollution at current and projected future levels. We are isolating the changes to marine shipping emissions only (as stated at the beginning of Section 5) and have not considered assessing the impact of shipping emissions under a changing climate. Indeed the impacts of shipping on ambient pollutant concentrations in the future will undoubtedly be linked to climate system changes. The reduced Arctic sea ice cover may have an impact on the atmospheric circulation and meteorological systems which will have an impact on the transport and transformation of atmospheric pollutants. The system is highly complex. To comment on how reduced sea ice would impact ozone and PM in future summertime Arctic in isolation without a proper study would not be very meaningful in our opinion.**

*Figures 4,5,6,7: How representative are the regional average concentrations, and is there much spread? e.g. it would be helpful to know how variable the concentrations are that make up each average. Can some spread be plotted in shading? This is also relevant to discussion of model evaluation above.*

**Response: We have addressed the reviewer's question by revising Figure 4-7 (comparison of the modelled and observed time series of regional averaged concentrations of O3, PM2.5, NO2, and SO2) to include the 1$^{st}$ to 3$^{rd}$ quartile range (shown in shaded bands), highlighting a measure of spread and how well (or not) the mean represents the distribution. The time series are shown in 24-hour running mean to focus on regional/synoptic events in the revised manuscript.**

**Changes:**

- **Revised Figure 4-7 to show the modelled and observed time series of regional averaged time series of O3, PM2.5, NO2, and SO2 (in 24-hour running mean – change from previously hourly time series in the original manuscript) and spreads (represented by the range between the 1$^{st}$ and 3$^{rd}$ quartiles, both model and observations);**
- **corresponding wording changes in section 4.2 on page 12 and 13 adjusting to the revised figures.**

*Editorial corrections:*

*Page 3, line 3: "theses" should be "these"*

**Response: Done. Thank you!**

[revised manuscript text omitted]

---

## Author Response (AR2)

Gong et al.: "Assessing the impact of shipping emissions on air pollution in the Canadian Arctic and northern regions: current and future modelled scenarios"

5 Authors' general acknowledgement: We are grateful for the efforts of the co-editor and two anonymous reviewers for going through this rather lengthy manuscript and providing valuable and constructive comments/suggestions that, we believe, had resulted in a much improved version of the manuscript.

In the following, we address the two comments from the co-editor. The comments from the co-editor are copied in *italic*, and authors' response and changes are in **bold**.

- 10 In the description of the model experiments: "The eight-month simulation was conducted by a series of staggered 30-hour runs with a 6-hour (meteorology-only) "jump-back", starting at 00 Z daily, to allow meteorological spin-up; the meteorology is initialized at 00 Z using the Canadian Meteorological Centre's regional objective analyses while chemistry (delayed for 6-10 hours from run start time) continues from the preceding run".
- 15 This is explaining the set-up of the model runs using quite detailed/technical jargon and was wondering if you could state this in a manner that is also easily to apprehend by readers not too familiar with such technical details. What is 00 Z?

Response: "00 Z" denotes 00 UTC. "Z" stands for Zulu time, the same as UTC or GMT. It is commonly used in numerical weather prediction community. We have revised the sentence to the following:

20 "The eight-month simulation was conducted by a series of staggered 30-hour runs with a 6-hour (meteorology-only) overlap, starting at 00 UTC daily, to allow meteorological spin-up from initialization; the meteorology is thus initialized at the beginning of every 30-hour run using the Canadian Meteorological Centre's regional objective analyses while chemistry is continuous."

*Pp 12-13; You show in figure 4-7 a comparison of the long-term time series of the simulated and observed tracer*

- 25 concentrations. First of all, although it is pretty clear it would be good to state here that these are surface observations. Then it is also good to know the typical reference height of these observations given that the reference height of the model simulated concentration is generally ~10m (given surface layer depths of 20m in low level terrain). And then in your discussion of what might explain the differences between the simulated and observed O3, I wonder also to what extent this might be due to issues on BL mixing and dry deposition as well as
- 30 the imposed boundary conditions (also in terms of high altitude ozone). Since you use daily mean value, these might be strongly effected by nocturnal biases associated with the representation of the nocturnal inversion layer and then the large sensitivity to small differences in dry deposition. You mentioned in the model introduction that you have adjusted the snow-ice deposition rates of O3 but also having this biases in

summertime there might be issues on the representation of deposition to the bare soil/sparse vegetation/water surfaces

Response: The data used for the model evaluation are indeed from ground-based observations. Unlike surface meteorological observations (e.g., 10 m for wind and 1.5 m for air temperature and humidity), there

- 5 isn't a standard height for the air chemistry measurements at the monitoring sites. Some of the monitors may be mounted on rooftops, for example. However, the probes are generally placed between 2 and 15 m above ground (e.g., according to Canadian National Air Pollution Surveillance network guideline; US networks have similar guidelines and some of the networks specify a height clearance of 5 – 10 feet for both ground and rooftop located monitors). For comparing with these "surface" observations, model results from the lowest
- 10 model level are used (~ 20 m above ground in our case, with a vertical coordinate that is terrain-following close to the surface). So overall there is not a significant difference between model and observation particularly in a well-mixed boundary layer. We have now added a statement in the opening paragraph of Section 4 (page 9) to clarify this:

"For comparing with these ground-based monitoring observations, model results were extracted from the

15 lowest model level (~20m above local surface) at given observational locations (nearest grid points). In contrast to surface meteorological observations, there is no standard height for the air chemistry measurements from the monitoring networks. However, the sampling probes are generally located between 2 and 15 m above local surface based on network guidelines."

Regarding the discussion on possible factors contributing to the difference between observed and simulated

- O3, we do mention that for the southwestern region the overall positive bias was largely a result of the overprediction of the O3 nighttime minima. This was better indicated from the hourly time series plots included in the previous version; the plots were replaced with the 24-hour running mean time series in the revised version, in order to show spread (e.g., 1st and 3rd quartiles) for addressing one of the reviewer's comments. The night-time over-prediction, as the co-editor alluded to (re BL mixing), can be a result of the model not
- 25 being able to simulate (or resolve) the nighttime stable boundary layer. The lowest model layer (at ~20m above surface) may be located in the residue layer rather than in the surface layer and the vertical gradient in O3 concentration close to the surface due to the loss from dry deposition (though generally smaller during nighttime) is thus not resolved by the model. This is a very valid point. We have now added the following sentence:
- 30 "The nighttime model bias can be a result of the model's difficulty in simulating (or resolving) the stable nocturnal boundary layer: the lowest model level may reside in residual layer rather than in the surface layer (where surface O3 monitors are located)."

Regarding the impact of imposed boundary condition, particularly whether the daily average from 3-hourly MACC-IFS reanalysis may be influenced by the possible nighttime bias, we have not looked into this (i.e.,

whether the MACC-IFS O3 reanalysis has a nighttime bias), but we cannot rule out possible bias in the O3 boundary condition due to the relatively coarse resolution (~ 80 km) of the reanalysis. As for high altitude O3, Inness et al. (2013) showed that the MACC-IFS O3 reanalysis compared well with ozonesonde measurements particularly for sites north of 30°N. The influence of high altitude (UTLS) O3 on ground level O3 is mostly

- 5 through stratosphere intrusion events which are more frequent in spring in mid-latitudes. Previous studies (e.g., Pendlebury et al., 2018; Bourqui and Trépanier, 2010) have shown that the meteorological model GEM (hosting model of GEM-MACH) is capable of capturing well the dynamics of these events. Regarding O3 dry deposition, we introduced the consideration of sea ice representation for our study and revised the O3 dry deposition velocity over ice and snow based on Helmig et al. (2007). The impact from this change is mostly
- 10 over ice covered ocean and northern coastal region during spring into early summer (in our simulation). It is possible that there may be issues with the current dry deposition parameters used for some of the land cover types in the north (e.g., tundra, bare soil, etc.). We did not explore this aspect in our study. There are indeed other factors that may contribute to the model biases. For example, a recent study found that accounting for forest canopy shading and canopy induced turbulence mixing can significantly reduce the high O3 bias
- 15 commonly found in regional air quality forecast models (including GEM-MACH, which do not include these processes), particularly in forested areas (Makar et al, 2017). This is potentially relevant to the northern boreal regions. We decided not to include these discussions in consideration of the length of the manuscript.

1Science and Technology Branch, Environment and Climate Change Canada, Toronto, Ontario, M3H 5T4, Canada 2Meteorological Service of Canada, Environment and Climate Change Canada, Montreal, Quebec, H9P 1J3, Canada 3Science and Technology Branch, Environment and Climate Change Canada, Ottawa, Ontario, K1V 1C7, Canada

4Environmental Protection Branch, Environment and Climate Change Canada, Gatineau, Quebec, K1A 0H3, Canada 5Environmental Protection Branch, Environment and Climate Change Canada, Toronto, Ontario, M3H 5T4, Canada 6Environmental Protection Branch, Environment and Climate Change Canada, Vancouver, British Columbia, V6C 3S5, Canada

Correspondence to: Wanmin Gong (Wanmin.gong@canada.ca)

[revised manuscript text omitted]
                                   |       | со                    |                                                                         | v     | ос   |                       | PM   |                       | NH₃ |   | CO 2 e |          |                      |
|-----------------------|------------------|-------|--------------------------------------------------|-----------------|-------|---------------------------------------------------|-------|-----------------------|-------------------------------------------------------------------------|-------|------|-----------------------|------|-----------------------|-----|---|-------------------|----------|----------------------|
| vessel Category/Class | 2010             | 2030  | 030 2010 2030 BAU 2030 ECA |                 | 2010  | 2010 2030 BAU 2030 ECA 2010 |       | 2030 & | 2010 2030 & 2010 2030 BAU 2030 ECA |       | 2010 | 2030 & | 2010 | 2030 & |     |   |                   |          |                      |
| Merchant Passenger    |                  |       | 308                                              | 1,049           | 762   | 127                                               | 186   | 38                    | 26                                                                      | 113   | 10   | 45                    | 18   | 40                    | 25  | 0 | 2                 | 13,814   | 54,826               |
| Merchant Passenger    | 63               | 271   | 308                                              | 1,049           | 762   | 127                                               | 186   | 38                    | 26                                                                      | 113   | 10   | 45                    | 18   | 40                    | 25  | 0 | 2                 | 13,814   | 54,826               |
| Merchant Commercial   |                  |       | 1,821                                            | 8,611           | 4,865 | 1,079                                             | 1,427 | 288                   | 163                                                                     | 842   | 64   | 342                   | 144  | 314                   | 193 | 2 | 12                | 90,502   | 418,727              |
| Merchant Bulk         | 39               | 191   | 431                                              | 4,926           | 2,266 | 206                                               | 798   | 160                   | 38                                                                      | 488   | 15   | 198                   | 28   | 176                   | 108 | 0 | 7                 | 17,389   | 224,260              |
| Merchant Other        | 245              | 453   | 815                                              | 1,810           | 1,381 | 568                                               | 320   | 64                    | 72                                                                      | 171   | 28   | 69                    | 74   | 69                    | 42  | 1 | 3                 | 40,700   | 97,454               |
| Tanker                | 169              | 247   | 575                                              | 1,875           | 1,218 | 305                                               | 310   | 63                    | 53                                                                      | 183   | 21   | 75                    | 42   | 69                    | 43  | 1 | 3                 | 32,413   | 97,013               |
| Other                 |                  |       | 1,157                                            | 1,602           | 1,602 | 17                                                | 15    | 15                    | 91                                                                      | 127   | 42   | 59                    | 23   | 31                    | 31  | 2 | 2                 | 57,223   | 74,914               |
| Coast Guard           | 20               | 25    | 613                                              | 844             | 844   | 10                                                | 13    | 13                    | 51                                                                      | 70    | 22   | 31                    | 13   | 17                    | 17  | 1 | 1                 | 31,861   | 41,050               |
| Tug Boat              | 300              | 367   | 506                                              | 720             | 720   | 7                                                 | 1     | 1                     | 37                                                                      | 55    | 18   | 26                    | 9    | 13                    | 13  | 1 | 1                 | 23,404   | 31,983               |
| Special Purpose       | 7                | 6     | 38                                               | 38              | 38    | 1                                                 | 1     | 1                     | 3                                                                       | 3     | 1    | 1                     | 1    | 1                     | 1   | 0 | 0                 | 1,958    | 1,881                |
| Fishing               |                  |       | 231                                              | 270             | 270   | 4                                                 | 0     | 0                     | 19                                                                      | 22    | 8    | 10                    | 5    | 5                     | 5   | 0 | 0                 | 11,195   | 12,770               |
| Fishing               | 134              | 156   | 231                                              | 270             | 270   | 4                                                 | 0     | 0                     | 19                                                                      | 22    | 8    | 10                    | 5    | 5                     | 5   | 0 | 0                 | 11,195   | 12,770               |
| Total                 | 978 * | 1,716 | 3,518                                            | 11,531          | 7,499 | 1,228                                             | 1,628 | 340                   | 299                                                                     | 1,104 | 125  | 455                   | 190  | 390                   | 253 | 4 | 16                | 172,740* | 561,243 * |

[revised manuscript text omitted]

J

**Table 7**. Arctic shipping contributions to population-weighted concentrations of criteria pollutants over eastern Canadian Arctic (North of 60 N, 60 - 100 W), for the July-August-September period.

| Scenario | PM 2.5 (%) |      |      | O 3 (%) |      |      | NO2 (%) |       |       | SO₂ (%) |       |       |
|----------|-----------------------|------|------|--------------------|------|------|---------|-------|-------|---------|-------|-------|
|          | mean                  | med. | max. | mean               | med. | max. | mean    | med.  | max.  | mean    | med.  | max.  |
| 2010     | 0.75                  | 0.63 | 3.23 | 0.37               | 0.37 | 0.83 | 7.73    | 6.95  | 15.23 | 53.93   | 62.35 | 83.35 |
| 2030 BAU | 1.28                  | 1.12 | 5.07 | 1.36               | 1.36 | 3.08 | 25.57   | 23.48 | 61.24 | 60.58   | 65.76 | 90.36 |
| 2030 ECA | 0.58                  | 0.48 | 2.54 | 0.98               | 0.96 | 2.47 | 23.39   | 20.27 | 60.78 | 28.78   | 28.54 | 65.15 |

**Table 8.** Percentage contribution from Arctic shipping to surface depositions of sulfur, nitrogen, and elemental carbon (EC) and column loading of EC, by geographical sectors (see Table 1), for the July-August-September period.

|     | sector | Total | S depositi | ion (%) | Total | N depositi | ion (%) | Total | Total BC deposition (%) |       |      | BC column (%) |       |  |
|-----|--------|-------|------------|---------|-------|------------|---------|-------|-------------------------|-------|------|---------------|-------|--|
|     | #      | mean  | med.       | max.    | mean  | med.       | max.    | mean  | med.                    | max.  | mean | med.          | max.  |  |
|     | E1     | 0.09  | 0.04       | 1.45    | 0.18  | 0.11       | 2.09    | 0.05  | 0.02                    | 3.06  | 0.02 | 0.01          | 0.67  |  |
|     | E2     | 0.07  | 0.05       | 2.39    | 0.07  | 0.05       | 0.74    | 0.02  | 0.02                    | 2.17  | 0.01 | 0.01          | 1.01  |  |
|     | E3     | 0.53  | 0.41       | 5.75    | 0.75  | 0.64       | 4.84    | 0.21  | 0.15                    | 8.98  | 0.06 | 0.05          | 2.10  |  |
|     | E4     | 0.51  | 0.42       | 6.99    | 0.61  | 0.51       | 4.40    | 0.14  | 0.09                    | 2.98  | 0.05 | 0.05          | 0.54  |  |
| 010 | E5     | 0.41  | 0.20       | 5.16    | 0.61  | 0.42       | 3.67    | 0.11  | 0.04                    | 4.21  | 0.01 | 0.00          | 0.36  |  |
| ~   | E6     | 0.61  | 0.49       | 5.08    | 0.94  | 0.85       | 4.37    | 0.20  | 0.13                    | 3.34  | 0.05 | 0.04          | 0.99  |  |
|     | W1     | 0.00  | 0.00       | 1.98    | 0.01  | 0.01       | 1.86    | -0.00 | 0.00                    | 7.31  | 0.00 | 0.00          | 0.09  |  |
|     | W2     | 0.07  | 0.04       | 2.73    | 0.32  | 0.14       | 8.41    | 0.03  | 0.01                    | 5.81  | 0.01 | 0.01          | 0.75  |  |
|     | W3     | 0.11  | 0.07       | 2.83    | 0.52  | 0.40       | 5.11    | 0.07  | 0.03                    | 2.89  | 0.01 | 0.00          | 0.19  |  |
|     | E1     | 0.12  | 0.06       | 2.69    | 0.71  | 0.42       | 5.82    | 0.85  | 0.62                    | 5.89  | 0.04 | 0.02          | 0.45  |  |
|     | E2     | 0.07  | 0.05       | 1.72    | 0.23  | 0.17       | 1.65    | 0.60  | 0.26                    | 9.01  | 0.02 | 0.01          | 0.55  |  |
|     | E3     | 1.54  | 0.50       | 33.20   | 5.41  | 2.60       | 57.50   | 0.70  | 0.47                    | 19.60 | 0.09 | 0.07          | 4.11  |  |
| AU  | E4     | 0.51  | 0.45       | 8.30    | 2.22  | 1.87       | 20.00   | 0.54  | 0.29                    | 4.81  | 0.09 | 0.08          | 0.71  |  |
| 30B | E5     | 1.61  | 0.51       | 34.30   | 5.01  | 2.07       | 59.00   | 0.61  | 0.35                    | 9.15  | 0.04 | -0.01         | 2.15  |  |
| 200 | E6     | 1.61  | 0.94       | 40.30   | 5.32  | 3.79       | 60.90   | 1.46  | 1.10                    | 32.80 | 0.11 | 0.04          | 15.90 |  |
|     | W1     | 0.01  | 0.00       | 3.52    | 0.04  | 0.03       | 2.10    | 0.32  | 0.13                    | 7.36  | 0.00 | 0.00          | 0.19  |  |
|     | W2     | 0.12  | 0.06       | 1.82    | 0.98  | 0.42       | 10.00   | 0.29  | 0.22                    | 7.10  | 0.02 | 0.01          | 0.95  |  |
|     | W3     | 0.20  | 0.15       | 2.55    | 1.41  | 1.14       | 7.85    | 0.30  | 0.17                    | 6.18  | 0.01 | 0.01          | 0.49  |  |
|     | E1     | 0.04  | 0.03       | 1.12    | 0.53  | 0.32       | 4.08    | 0.09  | 0.05                    | 2.93  | 0.04 | 0.03          | 0.33  |  |
|     | E2     | 0.02  | 0.02       | 1.05    | 0.17  | 0.13       | 1.33    | 0.03  | 0.03                    | 2.24  | 0.02 | 0.01          | 0.35  |  |
|     | E3     | 0.36  | 0.13       | 10.10   | 3.48  | 1.74       | 44.60   | 0.31  | 0.19                    | 13.70 | 0.08 | 0.07          | 2.81  |  |
| S   | E4     | 0.11  | 0.10       | 1.15    | 1.54  | 1.33       | 11.30   | 0.17  | 0.10                    | 2.96  | 0.07 | 0.06          | 0.67  |  |
| 0   | E5     | 0.37  | 0.12       | 10.60   | 3.35  | 1.41       | 48.50   | 0.21  | 0.06                    | 5.59  | 0.03 | 0.00          | 1.36  |  |
| 205 | E6     | 0.36  | 0.20       | 13.30   | 3.38  | 2.29       | 58.30   | 0.32  | 0.16                    | 21.80 | 0.09 | 0.04          | 10.20 |  |
|     | W1     | 0.00  | 0.00       | 2.11    | 0.03  | 0.02       | 3.13    | 0.00  | 0.00                    | 9.06  | 0.00 | 0.00          | 0.19  |  |
|     | W2     | 0.03  | 0.03       | 2.46    | 0.67  | 0.30       | 8.38    | 0.05  | 0.02                    | 6.31  | 0.02 | 0.01          | 0.95  |  |
|     | W3     | 0.05  | 0.04       | 1.03    | 0.89  | 0.73       | 5.25    | 0.09  | 0.04                    | 3.68  | 0.01 | 0.01          | 0.31  |  |

**Table 9.** Land-cover weighted deposition of S and N for eastern Canadian Arctic (60 - 100 W, 60 - 90 N) over the July-August-September period and corresponding contributions from Arctic shipping.

|       | Land                                                  | Sulfur |       |                              |       |      |       |                          | Nitrogen                  |                              |       |       |      |  |  |
|-------|-------------------------------------------------------|--------|-------|------------------------------|-------|------|-------|--------------------------|---------------------------|------------------------------|-------|-------|------|--|--|
| cover | LC-weighted deposition
(kg of S ha -1 ) |        |       | Shipping contribution
(%) |       |      | LC-we | ighted dep
kg of N ha | osition
1 ) | Shipping contribution
(%) |       |       |      |  |  |
|       | type                                                  | total  | dry   | wet                          | total | dry  | wet   | total                    | dry                       | wet                          | total | dry   | wet  |  |  |
| •     | lakes                                                 | 0.143  | 0.011 | 0.132                        | 0.32  | 1.37 | 0.23  | 0.084                    | 0.010                     | 0.073                        | 0.41  | 1.67  | 0.23 |  |  |
| 010   | tundra                                                | 0.116  | 0.008 | 0.109                        | 0.41  | 1.30 | 0.35  | 0.068                    | 0.011                     | 0.058                        | 0.60  | 1.97  | 0.35 |  |  |
| 2     | barren                                                | 0.073  | 0.005 | 0.067                        | 0.53  | 1.19 | 0.47  | 0.036                    | 0.005                     | 0.031                        | 0.81  | 2.42  | 0.58 |  |  |
| 0 -   | lakes                                                 | 0.143  | 0.011 | 0.132                        | 0.34  | 1.35 | 0.25  | 0.085                    | 0.011                     | 0.074                        | 1.50  | 5.41  | 0.92 |  |  |
| 3AL   | tundra                                                | 0.116  | 0.008 | 0.109                        | 0.52  | 1.80 | 0.43  | 0.070                    | 0.011                     | 0.059                        | 2.44  | 7.11  | 1.54 |  |  |
| 2     | barren                                                | 0.073  | 0.005 | 0.067                        | 1.20  | 2.78 | 1.08  | 0.037                    | 0.005                     | 0.032                        | 4.86  | 11.56 | 3.82 |  |  |
| 0 -   | lakes                                                 | 0.142  | 0.011 | 0.132                        | 0.08  | 0.26 | 0.06  | 0.085                    | 0.011                     | 0.074                        | 1.06  | 3.90  | 0.65 |  |  |
| 030   | tundra                                                | 0.116  | 0.007 | 0.108                        | 0.12  | 0.39 | 0.10  | 0.069                    | 0.011                     | 0.058                        | 1.71  | 5.11  | 1.07 |  |  |
| ~ -   | barren                                                | 0.072  | 0.005 | 0.067                        | 0.26  | 0.61 | 0.23  | 0.037                    | 0.005                     | 0.032                        | 3.01  | 7.36  | 2.35 |  |  |

**Table 10.** Averaged BC deposition on ice and snow over Canadian Arctic (50 – 140 W, 60 – 90 N), and contributions from shipping over the Canadian Arctic waters.

|            | Month | BC depositio | n to ice/snow (r | ng m -2 mon -1 ) | Arctic Sh | ipping contribu | Arctic Shipping contribution
(μg m -2 mon -1 ) |       |      |      |
|------------|-------|--------------|------------------|----------------------------------------|-----------|-----------------|-------------------------------------------------------------------------|-------|------|------|
|            |       | total        | dry              | wet                                    | total     | dry             | wet                                                                     | total | dry  | wet  |
| •          | 7     | 0.560        | 0.051            | 0.509                                  | 0.03      | 0.09            | 0.02                                                                    | 0.16  | 0.04 | 0.12 |
| 010        | 8     | 0.615        | 0.025            | 0.591                                  | 0.04      | 0.34            | 0.03                                                                    | 0.27  | 0.08 | 0.18 |
| 2          | 9     | 0.163        | 0.004            | 0.159                                  | 0.14      | 0.77            | 0.12                                                                    | 0.22  | 0.03 | 0.19 |
| 0 7        | 7     | 0.561        | 0.051            | 0.510                                  | 0.06      | 0.27            | 0.04                                                                    | 0.34  | 0.14 | 0.20 |
| 030
3AU | 8     | 0.617        | 0.025            | 0. 593                                 | 0.09      | 0.67            | 0.06                                                                    | 0.54  | 0.17 | 0.37 |
| 2 1        | 9     | 0. 163       | 0.004            | 0. 159                                 | 0.27      | 1.32            | 0.24                                                                    | 0.44  | 0.06 | 0.39 |

**Figures**

---

## Author Response (AR3)

Authors' response to co-editor's comments – acp-2018-125 (22 Oct 2018)

Gong et al.: "Assessing the impact of shipping emissions on air pollution in the Canadian Arctic and northern regions: current and future modelled scenarios"

*Co-Editor's comments to the Author:*

5   *Dear author, co-authors, thanks for addressing these last list of minor comments on the revised version of your ms submitted for publication in ACP. I read your response and this appears to handle well these comments accept the added statement on the nocturnal O3 bias and the role of the SBL.*

*Your proposed revision on this does not come out well according to me:*

*"The nighttime model bias can be a result of the model's difficulty in simulating (or resolving) the stable*
10   *nocturnal boundary layer: the lowest model level may reside in residual layer rather than in the surface layer (where surface O3 monitors are located).",*

*My point is that the model's lowest layer represents the surface layer and its processes and that the layer(s) above might represent the lowest part of the residual layer actually present in the atmosphere if indeed the SBL would be only about 20m deep (which I doubt, over snow/ice cover it could be indeed only some meters deep but*
15   *generally more ~50-150m). I would suggest to phrase this sentence to something like:*

*The nighttime model bias can be a result of the model's difficulty in simulating (or resolving) the stable nocturnal boundary layer and where small differences in the actual O3 sources and/or sinks, like O3 dry deposition can have a large impact on O3 concentration gradients that might also be reflected in significant differences in observed and simulated nocturnal O3 for different reference heights"*

20   **Authors' Response: We agree with the Co-Editor that the nighttime model bias may be linked to the model's difficulty in simulating (or resolving) the stable boundary layer in two folds: the model's inability to capture the vertical gradient well due to small differences in O3 sources and/or sinks, such as O3 deposition, and the difference in reference height between the model and observations. We have revised the sentence as suggested by the Co-Editor as follows:**

[revised manuscript text omitted]